# On the Mode-Seeking Properties of Langevin Dynamics

## Abstract

The Langevin Dynamics framework, which aims to generate samples from the score function of a probability distribution, is widely used for analyzing and interpreting score-based generative modeling. While the convergence behavior of Langevin Dynamics under unimodal distributions has been extensively studied in the literature, in practice the data distribution could consist of multiple distinct modes. In this work, we investigate Langevin Dynamics in producing samples from multimodal distributions and theoretically study its mode-seeking properties. We prove that under a variety of sub-Gaussian mixtures, Langevin Dynamics is unlikely to find all mixture components within a sub-exponential number of steps in the data dimension. To reduce the mode-seeking tendencies of Langevin Dynamics, we propose *Chained Langevin Dynamics*, which divides the data vector into patches of constant size and generates every patch sequentially conditioned on the previous patches. We perform a theoretical analysis of Chained Langevin Dynamics by reducing it to sampling from a constant-dimensional distribution. We present the results of several numerical experiments on synthetic and real image datasets, supporting our theoretical results on the iteration complexities of sample generation from mixture distributions using the chained and vanilla Langevin Dynamics.

## 1 Introduction

A central task in unsupervised learning involves learning the underlying probability distribution of training data and efficiently generating new samples from the distribution. Score-based generative modeling (SGM) (Song et al., 2020c) has achieved state-of-the-art performance in various learning tasks including image generation (Song and Ermon, 2019, 2020; Ho et al., 2020; Song et al., 2020a; Ramesh et al., 2022; Rombach et al., 2022), audio synthesis (Chen et al., 2020; Kong et al., 2020), and video generation (Ho et al., 2022; Blattmann et al., 2023). In addition to the successful empirical results, the convergence analysis of SGM has attracted significant attention in the recent literature (Lee et al., 2022, 2023; Chen et al., 2023; Li et al., 2023, 2024).

Stochastic gradient Langevin dynamics (SGLD) (Welling and Teh, 2011), as a fundamental methodology to implement and interpret SGM, can produce samples from the (Stein) score function of a probability density, i.e., the gradient of the log probability density function with respect to data. It has been widely recognized that a pitfall of SGLD is its slow mixing rate (Wooddard et al., 2009; Raginsky et al., 2017; Lee et al., 2018). Specifically, Song and Ermon (2019) shows that under a multi-modal data distribution, the samples from Langevin dynamics may have an incorrect relative density across the modes. Based on this finding, Song and Ermon (2019) proposes *anneal Langevin dynamics*, which injects different levels of Gaussian noise into the data distribution and samples with SGLD on the perturbed distribution. While outputting the correct relative density across modes can be challenging for SGLD, a natural question is whether SGLD would be able to find all the modes of a multi-modal distribution.

Submitted to 38th Conference on Neural Information Processing Systems (NeurIPS 2024). Do not distribute.

In this work, we study this question by analyzing the mode-seeking properties of SGLD. The notion of mode-seekingness (Bishop, 2006; Ke et al., 2021; Li and Farnia, 2023) refers to the property that a generative model captures only a subset of the modes of a multi-modal distribution. We note that a similar problem, known as metastability, has been studied in the context of Langevin diffusion, a continuous-time version of SGLD described by stochastic differential equation (SDE) (Bovier et al., 2002, 2004; Gayrard et al., 2005). Specifically, Bovier et al. (2002) gave a sharp bound on the mean hitting time of Langevin diffusion and proved that it may require exponential (in the space dimensionality $d$) time for transition between modes. Regarding discrete SGLD, Lee et al. (2018) constructed a probability distribution whose density is close to a mixture of two well-separated isotropic Gaussians, and proved that SGLD could not find one of the two modes within an exponential number of steps. However, further exploration of mode-seeking tendencies of SGLD and its variants such as annealed Langevin dynamics for general distributions is still lacking in the literature.

In this work, we theoretically formulate and demonstrate the potential mode-seeking tendency of SGLD. We begin by analyzing the convergence under a variety of Gaussian mixture probability distributions, under which SGLD could fail to visit all the mixture components within sub-exponential steps (in the data dimension). Subsequently, we generalize this result to mixture distributions with sub-Gaussian modes. This generalization extends our earlier result on Gaussian mixtures to a significantly larger family of mixture models, as the sub-Gaussian family includes any distribution over an $\ell_2$-norm-bounded support set. Furthermore, we extend our theoretical results to anneal Langevin dynamics with bounded noise scales.

To reduce SGLD's large iteration complexity shown under a high-dimensional input vector, we propose *Chained Langevin Dynamics (Chained-LD)*. Since SGLD could suffer from the curse of dimensionality, we decompose the sample $\mathbf{x} \in \mathbb{R}^d$ into $d/Q$ patches $\mathbf{x}^{(1)}, \cdots, \mathbf{x}^{(d/Q)}$, each of constant size $Q$, and sequentially generate every patch $\mathbf{x}^{(q)}$ for all $q \in [d/Q]$ statistically conditioned on previous patches, i.e., $P(\mathbf{x}^{(q)} \mid \mathbf{x}^{(0)}, \cdots \mathbf{x}^{(q-1)})$. The combination of all patches generated from the conditional distribution faithfully follows the probability density $P(\mathbf{x})$, while learning each patch requires less cost due to the reduced dimension. We also provide a theoretical analysis of Chained-LD by reducing the convergence of a $d$-dimensional sample to the convergence of each patch.

Finally, we present the results of several numerical experiments to validate our theoretical findings. For synthetic experiments, we consider moderately high-dimensional Gaussian mixture models, where the vanilla and annealed Langevin dynamics could not find all the components within a million steps, while Chained-LD could capture all the components with correct frequencies in $\mathcal{O}(10^4)$ steps. For experiments on real image datasets, we consider a mixture of two modes by using the original images from MNIST/Fashion-MNIST training dataset (black background and white digits/objects) as the first mode and constructing the second mode by i.i.d. flipping the images (white background and black digits/objects) with probability 0.5. Following from Song and Ermon (2019), we trained a Noise Conditional Score Network (NCSN) to estimate the score function. Our numerical results indicate that vanilla Langevin dynamics can fail to capture the two modes, as also observed by Song and Ermon (2019). On the other hand, Chained-LD was capable of finding both modes regardless of initialization. We summarize the contributions of this work as follows:

- Theoretically studying the mode-seeking properties of vanilla and annealed Langevin dynamics,
- Proposing Chained Langevin Dynamics (Chained-LD), which decomposes the sample into patches and sequentially generates each patch conditioned on previous patches,
- Providing a theoretical analysis of the convergence behavior of Chained-LD,
- Numerically comparing the mode-seeking properties of vanilla, annealed, and chained Langevin dynamics.

**Notations:** We use $[n]$ to denote the set $\{1, 2, \cdots, n\}$. Also, in the paper, $\|\cdot\|$ refers to the $\ell_2$ norm. We use $\mathbf{0}_n$ and $\mathbf{1}_n$ to denote a 0-vector and 1-vector of length $n$. We use $\boldsymbol{I}_n$ to denote the identity matrix of size $n \times n$. In the text, TV stands for the total variation distance.

## 2 Related Works

**Langevin Dynamics:** The convergence guarantees for Langevin diffusion, a continuous version of Langevin dynamics, are classical results extensively studied in the literature (Bhattacharya, 1978;

Roberts and Tweedie, 1996; Bakry and Émery, 1983; Bakry et al., 2008). Langevin dynamics, also known as Langevin Monte Carlo, is a discretization of Langevin diffusion typically modeled as a Markov Chain Monte Carlo (Welling and Teh, 2011). For unimodal distributions, e.g., the probability density function that is log-concave or satisfies log-Sobolev inequality, the convergence of Langevin dynamics is provably fast (Dalalyan, 2017; Durmus and Moulines, 2017; Vempala and Wibisono, 2019). However, for multimodal distributions, the non-asymptotic convergence analysis is much more challenging (Cheng et al., 2018). Raginsky et al. (2017) gave an upper bound on the convergence time of Langevin dynamics for arbitrary non-log-concave distributions with certain regularity assumptions, which, however, could be exponentially large without imposing more restrictive assumptions. Lee et al. (2018) studied the special case of a mixture of Gaussians of equal variance and provided heuristic analysis of sampling from general non-log-concave distributions.

**Mode-Seekingness of Langevin Dynamics:**  The investigation of the mode-seekingness of generative models starts with different generative adversarial network (GAN) (Goodfellow et al., 2014) model formulations and divergence measures, from both the practical (Goodfellow, 2016; Poole et al., 2016) and theoretical (Shannon et al., 2020; Li and Farnia, 2023) perspectives. In the context of Langevin dynamics, mode-seekingness is closely related to a lower bound on the transition time between two modes, e.g., two local maximums. Bovier et al. (2002, 2004); Gayrard et al. (2005) studied the mean hitting time of the continuous Langevin diffusion. Lee et al. (2018) proved the existence of a mixture of two Gaussian distributions whose covariance matrices differ by a constant factor, Langevin dynamics cannot find both modes in polynomial time.

**Score-based Generative Modeling:**  Since Song et al. (2020b) proposed sliced score matching which can train deep models to learn the score functions of implicit probability distributions on high-dimensional data, score-based generative modeling (SGM) has been going through a spurt of growth. Annealed Langevin dynamics (Song and Ermon, 2019) estimates the noise score of the probability density perturbed by Gaussian noise and utilizes stochastic gradient Langevin dynamics to generate samples from a sequence of decreasing noise scales. Song and Ermon (2020) conducted a heuristic analysis of the effect of noise levels on the performance of annealed Langevin dynamics. Denoising diffusion probabilistic model (DDPM) (Ho et al., 2020) incorporates a step-by-step introduction of random noise into data, followed by learning to reverse this diffusion process in order to generate desired data samples from the noise. Song et al. (2020c) unified anneal Langevin dynamics and DDPM via a stochastic differential equation. A recent line of work focuses on the non-asymptotic convergence guarantees for SGM with an imperfect score estimation under various assumptions on the data distribution (Block et al., 2020; De Bortoli et al., 2021; Lee et al., 2022; Chen et al., 2023; Benton et al., 2023; Li et al., 2023, 2024).

# 3 Preliminaries

## 3.1 Langevin Dynamics

Generative modeling aims to produce samples such that their distribution is close to the underlying true distribution $P$. For a continuously differentiable probability density $P(\mathbf{x})$ on $\mathbb{R}^d$, its score function is defined as the gradient of the log probability density function (PDF) $\nabla_{\mathbf{x}} \log P(\mathbf{x})$. Langevin diffusion is a stochastic process defined by the stochastic differential equation (SDE)

$$\mathrm{d}\mathbf{x}_t = -\nabla_{\mathbf{x}} \log P(\mathbf{x}_t) \, \mathrm{d}t + \sqrt{2} \, \mathrm{d}\mathbf{w}_t,$$

where $\mathbf{w}_t$ is the Wiener process on $\mathbb{R}^d$. To generate samples from Langevin diffusion, Welling and Teh (2011) proposed stochastic gradient Langevin dynamics (SGLD), a discretization of the SDE for $T$ iterations. Each iteration of SGLD is defined as

$$\mathbf{x}_t = \mathbf{x}_{t-1} + \frac{\delta_t}{2} \nabla_{\mathbf{x}} \log P(\mathbf{x}_{t-1}) + \sqrt{\delta_t} \boldsymbol{\epsilon}_t, \tag{1}$$

where $\delta_t$ is the step size and $\boldsymbol{\epsilon}_t \sim \mathcal{N}(\mathbf{0}_d, \boldsymbol{I}_d)$ is Gaussian noise. It has been widely recognized that Langevin diffusion could take exponential time to mix without additional assumptions on the probability density (Bovier et al., 2002, 2004; Gayrard et al., 2005; Raginsky et al., 2017; Lee et al., 2018). To combat the slow mixing, Song and Ermon (2019) proposed annealed Langevin dynamics by perturbing the probability density with Gaussian noise of variance $\sigma^2$, i.e.,

$$P_\sigma(\mathbf{x}) := \int P(\mathbf{z}) \mathcal{N}(\mathbf{x} \mid \mathbf{z}, \sigma^2 \boldsymbol{I}_d) \, \mathrm{d}\mathbf{z}, \tag{2}$$

and running SGLD on the perturbed data distribution $P_{\sigma_t}(\mathbf{x})$ with gradually decreasing noise levels $\{\sigma_t\}_{t \in [T]}$, i.e.,

$$\mathbf{x}_t = \mathbf{x}_{t-1} + \frac{\delta_t}{2} \nabla_{\mathbf{x}} \log P_{\sigma_t}(\mathbf{x}_{t-1}) + \sqrt{\delta_t} \boldsymbol{\epsilon}_t, \tag{3}$$

where $\delta_t$ is the step size and $\boldsymbol{\epsilon}_t \sim \mathcal{N}(\mathbf{0}_d, \boldsymbol{I}_d)$ is Gaussian noise. When the noise level $\sigma$ is vanishingly small, the perturbed distribution is close to the true distribution, i.e., $P_\sigma(\mathbf{x}) \approx P(\mathbf{x})$. Since we do not have direct access to the (perturbed) score function, Song and Ermon (2019) proposed the Noise Conditional Score Network (NCSN) $\mathbf{s}_{\boldsymbol{\theta}}(\mathbf{x}, \sigma)$ to jointly estimate the scores of all perturbed data distributions, i.e.,

$$\forall \sigma \in \{\sigma_t\}_{t \in [T]}, \ \mathbf{s}_{\boldsymbol{\theta}}(\mathbf{x}, \sigma) \approx \nabla_{\mathbf{x}} \log P_\sigma(\mathbf{x}).$$

To train the NCSN, Song and Ermon (2019) adopted denoising score matching, which minimizes the following loss

$$\mathcal{L}\left(\boldsymbol{\theta}; \{\sigma_t\}_{t \in [T]}\right) := \frac{1}{2T} \sum_{t \in [T]} \sigma_t^2 \mathbb{E}_{\mathbf{x} \sim P} \mathbb{E}_{\tilde{\mathbf{x}} \sim \mathcal{N}(\mathbf{x}, \sigma_t^2 \boldsymbol{I}_d)} \left[ \left\| \mathbf{s}_{\boldsymbol{\theta}}(\tilde{\mathbf{x}}, \sigma_t) - \frac{\tilde{\mathbf{x}} - \mathbf{x}}{\sigma_t^2} \right\|^2 \right].$$

Assuming the NCSN has enough capacity, $\mathbf{s}_{\boldsymbol{\theta}^*}(\mathbf{x}, \sigma)$ minimizes the loss $\mathcal{L}\left(\boldsymbol{\theta}; \{\sigma_t\}_{t \in [T]}\right)$ if and only if $\mathbf{s}_{\boldsymbol{\theta}^*}(\mathbf{x}, \sigma_t) = \nabla_{\mathbf{x}} \log P_{\sigma_t}(\mathbf{x})$ almost surely for all $t \in [T]$.

## 3.2 Multi-Modal Distributions

Our work focuses on multi-modal distributions. We use $P = \sum_{i \in [k]} w_i P^{(i)}$ to represent a mixture of $k$ modes, where each mode $P^{(i)}$ is a probability density with frequency $w_i$ such that $w_i > 0$ for all $i \in [k]$ and $\sum_{i \in [k]} w_i = 1$. In our theoretical analysis, we consider Gaussian mixtures and sub-Gaussian mixtures, i.e., every component $P^{(i)}$ is a Gaussian or sub-Gaussian distribution. A probability distribution $p(\mathbf{z})$ of dimension $d$ is defined as a sub-Gaussian distribution with parameter $\nu^2$ if, given the mean vector $\boldsymbol{\mu} := \mathbb{E}_{\mathbf{z} \sim p}[\mathbf{z}]$, the moment generating function (MGF) of $p$ satisfies the following inequality for every vector $\boldsymbol{\alpha} \in \mathbb{R}^d$:

$$\mathbb{E}_{\mathbf{z} \sim p} \left[ \exp\left( \boldsymbol{\alpha}^T (\mathbf{z} - \boldsymbol{\mu}) \right) \right] \leq \exp\left( \frac{\nu^2 \|\boldsymbol{\alpha}\|_2^2}{2} \right). \tag{4}$$

We remark that sub-Gaussian distributions include a wide variety of distributions such as Gaussian distributions and any distribution within a bounded $\ell_2$-norm distance from the mean $\boldsymbol{\mu}$. From equation 2 we note that the perturbed distribution is the convolution of the original distribution and a Gaussian random variable, i.e., for random variables $\mathbf{z} \sim p$ and $\mathbf{t} \sim \mathcal{N}(\mathbf{0}_d, \boldsymbol{I}_d)$, their sum $\mathbf{z} + \mathbf{t} \sim p_\sigma$ follows the perturbed distribution with noise level $\sigma$. Therefore, a perturbed (sub)Gaussian distribution remains (sub)Gaussian. We formalize this property in Proposition 1 and defer the proof to Appendix A for completeness.

**Proposition 1.** *Suppose the perturbed distribution of a $d$-dimensional probability distribution $p$ with noise level $\sigma$ is $p_\sigma$, then the mean of the perturbed distribution is the same as the original distribution, i.e., $\mathbb{E}_{\mathbf{z} \sim p_\sigma}[\mathbf{z}] = \mathbb{E}_{\mathbf{z} \sim p}[\mathbf{z}]$. If $p = \mathcal{N}(\boldsymbol{\mu}, \boldsymbol{\Sigma})$ is a Gaussian distribution, $p_\sigma = \mathcal{N}(\boldsymbol{\mu}, \boldsymbol{\Sigma} + \sigma^2 \boldsymbol{I}_d)$ is also a Gaussian distribution. If $p$ is a sub-Gaussian distribution with parameter $\nu^2$, $p_\sigma$ is a sub-Gaussian distribution with parameter $(\nu^2 + \sigma^2)$.*

# 4 Theoretical Analysis of the Mode-Seeking Properties of Langevin Dynamics

In this section, we theoretically investigate the mode-seeking properties of vanilla and annealed Langevin dynamics. We begin with analyzing Langevin dynamics in Gaussian mixtures.

## 4.1 Langevin Dynamics in Gaussian Mixtures

**Assumption 1.** *Consider a data distribution $P := \sum_{i=0}^{k} w_i P^{(i)}$ as a mixture of Gaussian distributions, where $1 \leq k = o(d)$ and $w_i > 0$ is a positive constant such that $\sum_{i=0}^{k} w_i = 1$. Suppose that $P^{(i)} = \mathcal{N}(\boldsymbol{\mu}_i, \nu_i^2 \boldsymbol{I}_d)$ is a Gaussian distribution over $\mathbb{R}^d$ for all $i \in \{0\} \cup [k]$ such that for all $i \in [k]$, $\nu_i < \nu_0$ and $\|\boldsymbol{\mu}_i - \boldsymbol{\mu}_0\|^2 \leq \frac{\nu_0^2 - \nu_i^2}{2} \left( \log\left( \frac{\nu_i^2}{\nu_0^2} \right) - \frac{\nu_i^2}{2\nu_0^2} + \frac{\nu_0^2}{2\nu_i^2} \right) d$. Denote $\nu_{\max} := \max_{i \in [k]} \nu_i$.*

177 Regarding the first requirement $\nu_i < \nu_0$, we first note that the probability density $p(\mathbf{z})$ of a Gaussian

178 distribution $\mathcal{N}(\boldsymbol{\mu}, \nu^2 \boldsymbol{I}_d)$ decays exponentially in terms of $\frac{\|\mathbf{z}-\boldsymbol{\mu}\|^2}{\nu^2}$. When a state $\mathbf{z}$ is sufficiently far

179 from all modes (i.e., $\|\mathbf{z}\| \gg \|\boldsymbol{\mu}_i\|$), the Gaussian distribution with the largest variance (i.e., $P^{(0)}$ in

180 Assumption 1) dominates all other modes because $\frac{\|\mathbf{z}-\boldsymbol{\mu}_0\|^2}{\nu_0^2} \approx \frac{\|\mathbf{z}\|^2}{\nu_0^2} \gg \frac{\|\mathbf{z}\|^2}{\nu_i^2} \approx \frac{\|\mathbf{z}-\boldsymbol{\mu}_i\|^2}{\nu_i^2}$. We call

181 such mode $P^{(0)}$ the *universal mode*. Therefore, if $\mathbf{z}$ is initialized far from all modes, it can only

182 converge to the universal mode because the gradient information of other modes is masked. Once

183 $\mathbf{z}$ enters the universal mode $P^{(0)}$, if the step size $\delta_t$ of Langevin dynamics is small (i.e., $\delta_t \leq \nu_0^2$),

184 it would take exponential steps to escape the local mode $P^{(0)}$; while if the step size is large (i.e.,

185 $\delta_t > \nu_0^2$), the state $\mathbf{z}$ would again be far from all modes and thus the universal mode $P^{(0)}$ dominates

186 all other modes. Hence, $\mathbf{z}$ can only visit the universal mode unless the stochastic noise $\boldsymbol{\epsilon}_t$ miraculously

187 leads it to the region of another mode. In addition, it can be verified that $\log\left(\frac{\nu_i^2}{\nu_0^2}\right) - \frac{\nu_i^2}{2\nu_0^2} + \frac{\nu_0^2}{2\nu_i^2}$ is a

188 positive constant for $\nu_i < \nu_0$, thus the second requirement of Assumption 1 essentially represents

189 $\|\boldsymbol{\mu}_i - \boldsymbol{\mu}_0\|^2 \leq \mathcal{O}(d)$. We formalize the intuition in Theorem 1 and defer the proof to Appendix A.1.

190 **Theorem 1.** *Consider a data distribution $P$ satisfying Assumption 1. We follow Langevin dynamics*

191 *for $T = \exp(\mathcal{O}(d))$ steps. Suppose the sample is initialized in $P^{(0)}$, then with probability at least*

192 $1 - T \cdot \exp(-\Omega(d))$, *we have $\|\mathbf{x}_t - \boldsymbol{\mu}_i\|^2 > \frac{\nu_0^2 + \nu_{\max}^2}{2} d$ for all $t \in \{0\} \cup [T]$ and $i \in [k]$.*

193 We note that $\|\mathbf{x}_t - \boldsymbol{\mu}_i\|^2 > \frac{\nu_0^2 + \nu_{\max}^2}{2} d$ is a strong notion of mode-seekingness, since the probability

194 density of mode $P^{(i)} = \mathcal{N}(\boldsymbol{\mu}_i, \nu_i^2 \boldsymbol{I}_d)$ concentrates around the $\ell_2$-norm ball $\left\{ \mathbf{z} : \|\mathbf{z} - \boldsymbol{\mu}_i\|^2 \leq \nu_i^2 d \right\}$.

195 This notion can also easily be translated into a lower bound in terms of other distance measures such

196 as total variation distance and Wasserstein 2-distance. Moreover, in Theorem 2 we extend the result

197 to annealed Langevin dynamics with bounded noise level, and the proof is deferred to Appendix A.2.

198 **Theorem 2.** *Consider a data distribution $P$ satisfying Assumption 1. We follow annealed Langevin*

199 *dynamics for $T = \exp(\mathcal{O}(d))$ steps with noise levels $c_\sigma \geq \sigma_0 \geq \cdots \geq \sigma_T \geq 0$ for constant $c_\sigma > 0$.*

200 *In addition, assume for all $i \in [k]$, $\|\boldsymbol{\mu}_i - \boldsymbol{\mu}_0\|^2 \leq \frac{\nu_0^2 - \nu_i^2}{2} \left( \log\left(\frac{\nu_i^2 + c_\sigma^2}{\nu_0^2 + c_\sigma^2}\right) - \frac{\nu_i^2 + c_\sigma^2}{2\nu_0^2 + c_\sigma^2} + \frac{\nu_0^2 + c_\sigma^2}{2\nu_i^2 + c_\sigma^2} \right) d$.*

201 *Suppose that the sample is initialized in $P_{\sigma_0}^{(0)}$, then with probability at least $1 - T \cdot \exp(-\Omega(d))$, we*

202 *have $\|\mathbf{x}_t - \boldsymbol{\mu}_i\|^2 > \frac{\nu_0^2 + \nu_{\max}^2 + 2\sigma_t^2}{2} d$ for all $t \in \{0\} \cup [T]$ and $i \in [k]$.*

## 4.2 Langevin Dynamics in Sub-Gaussian Mixtures

204 We further generalize our results to sub-Gaussian mixtures. We impose the following assumptions on

205 the mixture. It is worth noting that these assumptions automatically hold for Gaussian mixtures.

206 **Assumption 2.** *Consider a data distribution $P := \sum_{i=0}^{k} w_i P^{(i)}$ as a mixture of sub-Gaussian*

207 *distributions, where $1 \leq k = o(d)$ and $w_i > 0$ is a positive constant such that $\sum_{i=0}^{k} w_i = 1$.*

208 *Suppose that $P^{(0)} = \mathcal{N}(\boldsymbol{\mu}_0, \nu_0^2 \boldsymbol{I}_d)$ is Gaussian and for all $i \in [k]$, $P^{(i)}$ satisfies*

209   *i. $P^{(i)}$ is a sub-Gaussian distribution of mean $\boldsymbol{\mu}_i$ with parameter $\nu_i^2$,*

210   *ii. $P^{(i)}$ is differentiable and $\nabla P^{(i)}(\boldsymbol{\mu}_i) = \mathbf{0}_d$,*

211   *iii. the score function of $P^{(i)}$ is $L_i$-Lipschitz such that $L_i \leq \frac{c_L}{\nu_i^2}$ for some constant $c_L > 0$,*

212   *iv. $\nu_0^2 > \max\left\{ 1, \frac{4(c_L^2 + c_\nu c_L)}{c_\nu (1 - c_\nu)} \right\} \frac{\nu_{\max}^2}{1 - c_\nu}$ for constant $c_\nu \in (0, 1)$, where $\nu_{\max} := \max_{i \in [k]} \nu_i$,*

213   *v. $\|\boldsymbol{\mu}_i - \boldsymbol{\mu}_0\|^2 \leq \frac{(1 - c_\nu)\nu_0^2 - \nu_i^2}{2(1 - c_\nu)} \left( \log \frac{c_\nu \nu_i^2}{(c_L^2 + c_\nu c_L)\nu_0^2} - \frac{\nu_i^2}{2(1 - c_\nu)\nu_0^2} + \frac{(1 - c_\nu)\nu_0^2}{2\nu_i^2} \right) d$.*

214 We validate the feasibility of Assumption 2.v. in Lemma 9 in the Appendix. With Assumption 2, we

215 show the mode-seeking tendency of Langevin dynamics under sub-Gaussian distributions in Theorem

216 3 and defer the proof to Appendix A.3.

---

**Algorithm 1** Chained Langevin Dynamics (Chained-LD)

---

**Require:** Patch size $Q$, dimension $d$, conditional score function estimator $\mathbf{s_\theta}$, number of iterations $T$, noise levels $\{\sigma_t\}_{t\in[TQ/d]}$, step size $\{\delta_t\}_{t\in[TQ/d]}$.

1:  Initialize $\mathbf{x}_0$, and divide $\mathbf{x}_0$ into $d/Q$ patches $\mathbf{x}_0^{(1)}, \cdots \mathbf{x}_0^{(d/Q)}$ of equal size $Q$

2:  **for** $q \leftarrow 1$ to $d/Q$ **do**

3:     **for** $t \leftarrow 1$ to $TQ/d$ **do**

4:         $\mathbf{x}_t^{(q)} \leftarrow \mathbf{x}_{t-1}^{(q)} + \frac{\delta_t}{2}\mathbf{s_\theta}\left(\mathbf{x}_t^{(q)} \mid \sigma_t, \mathbf{x}_t^{(1)}, \cdots, \mathbf{x}_t^{(q-1)}\right) + \sqrt{\delta_t}\boldsymbol{\epsilon}_t$, where $\boldsymbol{\epsilon}_t \sim \mathcal{N}(\mathbf{0}_Q, \boldsymbol{I}_Q)$

5:     **end for**

6:     $\mathbf{x}_0^{(q)} \leftarrow \mathbf{x}_{TQ/d}^{(q)}$

7:  **end for**

8:  **return** $\mathbf{x}_{TQ/d}$

---

**Theorem 3.** *Consider a data distribution $P$ satisfying Assumption 2. We follow Langevin dynamics for $T = \exp(\mathcal{O}(d))$ steps. Suppose the sample is initialized in $P^{(0)}$, then with probability at least $1 - T \cdot \exp(-\mathcal{O}(d))$, we have $\|\mathbf{x}_t - \boldsymbol{\mu}_i\|^2 > \left(\frac{\nu_0^2}{2} + \frac{\nu_{\max}^2}{2(1-c_\nu)}\right) d$ for all $t \in \{0\} \cup [T]$ and $i \in [k]$.*

Finally, we slightly modify Assumption 2 and extend our results to annealed Langevin dynamics under sub-Gaussian mixtures in Theorem 4. The details of Assumption 3 and the proof of Theorem 4 are deferred to Appendix A.4.

**Theorem 4.** *Consider a data distribution $P$ satisfying Assumption 3. We follow annealed Langevin dynamics for $T = \exp(\mathcal{O}(d))$ steps with noise levels $c_\sigma \geq \sigma_0 \geq \cdots \geq \sigma_T \geq 0$. Suppose the sample is initialized in $P_{\sigma_0}^{(0)}$, then with probability at least $1 - T \cdot \exp(-\mathcal{O}(d))$, we have $\|\mathbf{x}_t - \boldsymbol{\mu}_i\|^2 > \left(\frac{\nu_0^2 + \sigma_t^2}{2} + \frac{\nu_{\max}^2 + \sigma_t^2}{2(1-c_\nu)}\right) d$ for all $t \in \{0\} \cup [T]$ and $i \in [k]$.*

## 5 Chained Langevin Dynamics

To reduce the mode-seeking tendencies of vanilla and annealed Langevin dynamics, we propose Chained Langevin Dynamics (Chained-LD) in Algorithm 1. While vanilla and annealed Langevin dynamics apply gradient updates to all coordinates of the sample in every step, we decompose the sample into patches of constant size and generate each patch sequentially to alleviate the exponential dependency on the dimensionality. More precisely, we divide a sample $\mathbf{x}$ into $d/Q$ patches $\mathbf{x}^{(1)}, \cdots \mathbf{x}^{(d/Q)}$ of some constant size $Q$, and apply annealed Langevin dynamics to sample each patch $\mathbf{x}^{(q)}$ (for $q \in [d/Q]$) from the conditional distribution $P(\mathbf{x}^{(q)} \mid \mathbf{x}^{(1)}, \cdots \mathbf{x}^{(q-1)})$.

An ideal conditional score function estimator $\mathbf{s_\theta}$ could jointly estimate the scores of all perturbed conditional patch distribution, i.e., $\forall \sigma \in \{\sigma_t\}_{t\in[TQ/d]}, q \in [d/Q]$,

$$\mathbf{s_\theta}\left(\mathbf{x}^{(q)} \mid \sigma, \mathbf{x}^{(1)}, \cdots, \mathbf{x}^{(q-1)}\right) \approx \nabla_{\mathbf{x}^{(q)}} \log P_\sigma(\mathbf{x}^{(q)} \mid \mathbf{x}^{(1)}, \cdots \mathbf{x}^{(q-1)}).$$

Following from Song and Ermon (2019), we use the denoising score matching to train the estimator. For a given $\sigma$, the denoising score matching objective is

$$\ell(\boldsymbol{\theta}; \sigma) := \frac{1}{2}\mathbb{E}_{\mathbf{x}\sim P}\mathbb{E}_{\tilde{\mathbf{x}}\sim\mathcal{N}(\mathbf{x},\sigma^2\boldsymbol{I}_d)} \sum_{q\in[d/Q]} \left[\left\|\mathbf{s_\theta}\left(\mathbf{x}^{(q)} \mid \sigma, \mathbf{x}^{(1)}, \cdots, \mathbf{x}^{(q-1)}\right) - \frac{\tilde{\mathbf{x}}^{(q)} - \mathbf{x}^{(q)}}{\sigma^2}\right\|^2\right].$$

Then, combining the objectives gives the following loss

$$\mathcal{L}\left(\boldsymbol{\theta}; \{\sigma_t\}_{t\in[TQ/d]}\right) := \frac{d}{TQ} \sum_{t\in[TQ/d]} \sigma_t^2 \ell(\boldsymbol{\theta}; \sigma_t).$$

As shown in Vincent (2011), an estimator $\mathbf{s_\theta}$ with enough capacity minimizes the loss $\mathcal{L}$ if and only if $\mathbf{s_\theta}$ outputs the scores of all perturbed conditional patch distribution almost surely. Ideally, if a sampler

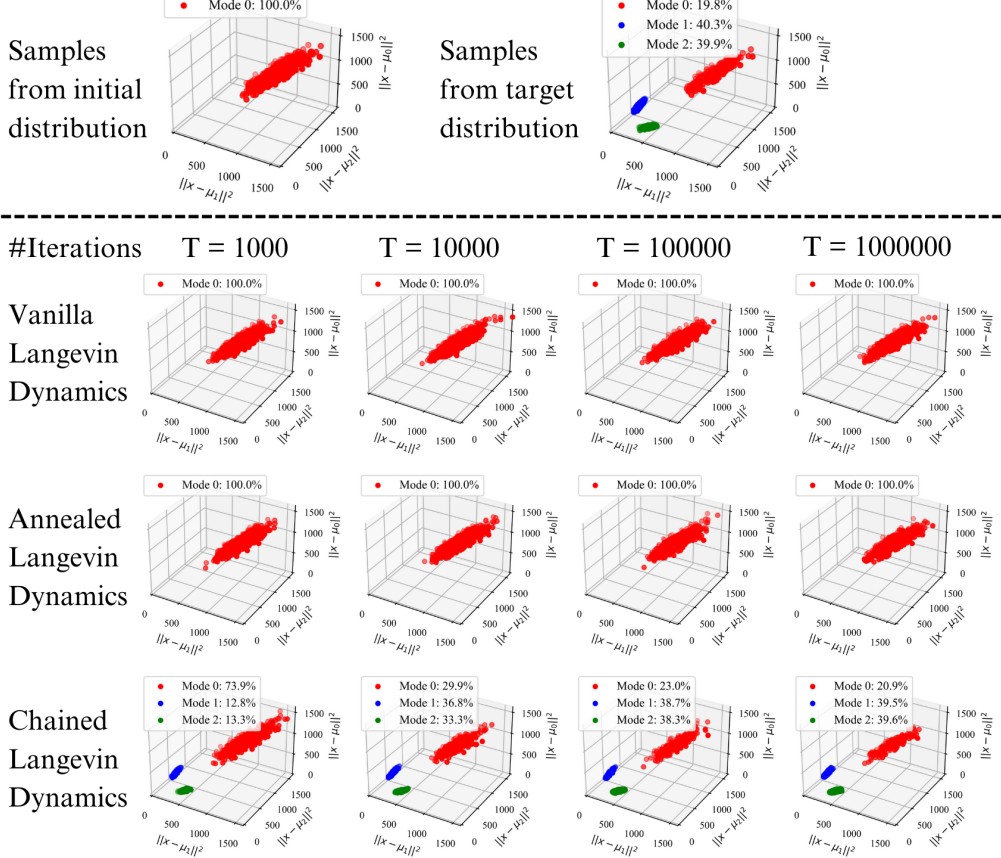

Figure 1: Samples from a mixture of three Gaussian modes generated by vanilla, annealed, and chained Langevin dynamics. Three axes are $\ell_2$ distance from samples to the mean of the three modes. The samples are initialized in mode 0.

perfectly generates every patch, combining all patches gives a sample from the original distribution since $P(\mathbf{x}) = \prod_{q \in [d/Q]} P(\mathbf{x}^{(q)} \mid \mathbf{x}^{(1)}, \cdots \mathbf{x}^{(q-1)})$. In Theorem 5 we give a linear reduction from producing samples of dimension $d$ using Chained-LD to learning the distribution of a $Q$-dimensional variable for constant $Q$. The proof of Theorem 5 is deferred to Appendix A.5.

**Theorem 5.** *Consider a sampler algorithm taking the first $q - 1$ patches $\mathbf{x}^{(1)}, \cdots, \mathbf{x}^{(q-1)}$ as input and outputing a sample of the next patch $\mathbf{x}^{(q)}$ with probability $\hat{P}\left(\mathbf{x}^{(q)} \mid \mathbf{x}^{(1)}, \cdots, \mathbf{x}^{(q-1)}\right)$ for all $q \in [d/Q]$. Suppose that for every $q \in [d/Q]$ and any given previous patches $\mathbf{x}^{(1)}, \cdots, \mathbf{x}^{(q-1)}$, the sampler algorithm can achieve*

$$TV\left(\hat{P}\left(\mathbf{x}^{(q)} \mid \mathbf{x}^{(1)}, \cdots, \mathbf{x}^{(q-1)}\right), P\left(\mathbf{x}^{(q)} \mid \mathbf{x}^{(1)}, \cdots, \mathbf{x}^{(q-1)}\right)\right) \le \varepsilon \cdot \frac{Q}{d}$$

*in $\tau(\varepsilon, d)$ iterations for some $\varepsilon > 0$. Then, equipped with the sampler algorithm, the Chained-LD algorithm in $\frac{d}{Q} \cdot \tau(\varepsilon, d)$ iterations can achieve*

$$TV\left(\hat{P}(\mathbf{x}), P(\mathbf{x})\right) \le \varepsilon.$$

## 6 Numerical Results

In this section, we empirically evaluated the mode-seeking tendencies of vanilla, annealed, and chained Langevin dynamics. We performed numerical experiments on synthetic Gaussian mixture models and real image datasets including MNIST (LeCun, 1998) and Fashion-MNIST (Xiao et al., 2017). Details on the experiment setup are deferred to Appendix B.

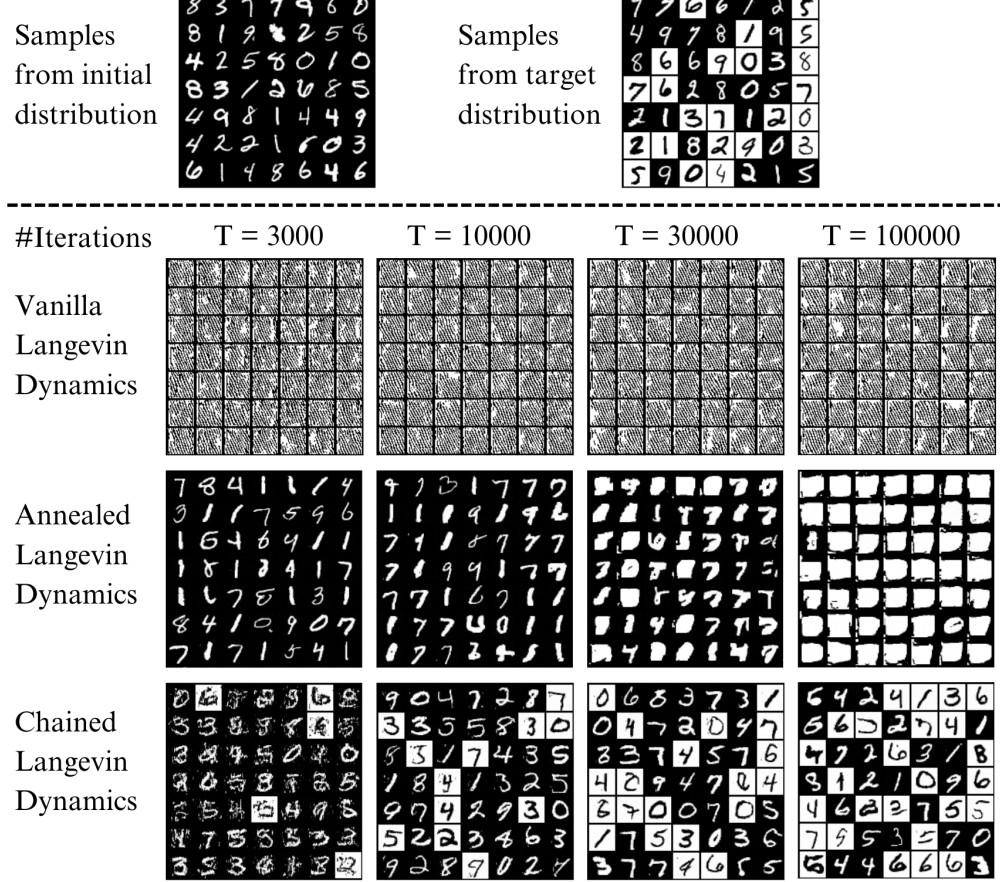

Figure 2: Samples from a mixture distribution of the original and flipped images from the MNIST dataset generated by vanilla, annealed, and chained Langevin dynamics. The samples are initialized as original images from MNIST.

**Synthetic Gaussian mixture model:** We define the data distribution $P$ as a mixture of three Gaussian components in dimension $d = 100$, where mode 0 defined as $P^{(0)} = \mathcal{N}(\mathbf{0}_d, 3\boldsymbol{I}_d)$ is the universal mode with the largest variance, and mode 1 and mode 2 are respectively defined as $P^{(1)} = \mathcal{N}(\mathbf{1}_d, \boldsymbol{I}_d)$ and $P^{(2)} = \mathcal{N}(-\mathbf{1}_d, \boldsymbol{I}_d)$. The frequencies of the three modes are 0.2, 0.4 and 0.4, i.e.,

$$P = 0.2P^{(0)} + 0.4P^{(1)} + 0.4P^{(2)} = 0.2\mathcal{N}(\mathbf{0}_d, 3\boldsymbol{I}_d) + 0.4\mathcal{N}(\mathbf{1}_d, \boldsymbol{I}_d) + 0.4\mathcal{N}(-\mathbf{1}_d, \boldsymbol{I}_d).$$

As shown in Figure 1, vanilla and annealed Langevin dynamics cannot find mode 1 or 2 within $10^6$ iterations if the sample is initialized in mode 0, while chained Langevin dynamics can find the other two modes in 1000 steps and correctly recover their frequencies as gradually increasing the number of iterations. In Appendix B.1 we present additional experiments on samples initialized in mode 1 or 2, which also verify the mode-seeking tendencies of vanilla and annealed Langevin dynamics.

**Image datasets:** We construct the distribution as a mixture of two modes by using the original images from MNIST/Fashion-MNIST training dataset (black background and white digits/objects) as the first mode and constructing the second mode by i.i.d. randomly flipping an image (white background and black digits/objects) with probability 0.5. Regarding the neural network architecture of the score function estimator, for vanilla and annealed Langevin dynamics we use U-Net (Ronneberger et al., 2015) following from Song and Ermon (2019). For chained Langevin dynamics, we proposed to use Recurrent Neural Network (RNN) architectures. We note that for a sequence of inputs, the output of RNN from the previous step is fed as input to the current step. Therefore, in the scenario of chained Langevin dynamics, the hidden state of RNN contains information about the previous patches and allows the network to estimate the conditional score function $\nabla_{\mathbf{x}^{(q)}} \log P(\mathbf{x}^{(q)} \mid \mathbf{x}^{(1)}, \cdots \mathbf{x}^{(q-1)})$. More implementation details are deferred to Appendix B.2.

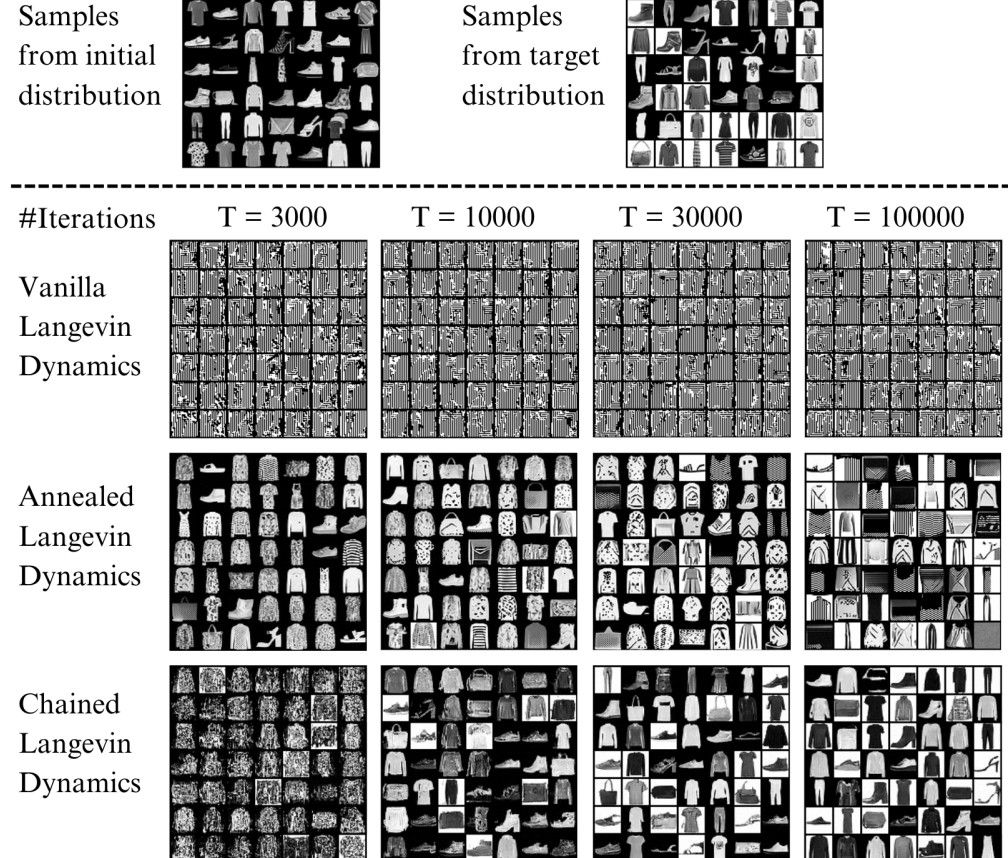

Figure 3: Samples from a mixture distribution of the original and flipped images from the Fashion-MNIST dataset generated by vanilla, annealed, and chained Langevin dynamics. The samples are initialized as original images from Fashion-MNIST.

The numerical results on image datasets are shown in Figures 2 and 3. Vanilla Langevin dynamics fails to generate reasonable samples, as also observed in Song and Ermon (2019). When the sample is initialized as original images from the datasets, annealed Langevin dynamics tends to generate samples from the same mode, while chained Langevin dynamics can generate samples from both modes. Additional experiments are deferred to Appendix B.2.

# 7 Conclusion

In this work, we theoretically and numerically studied the mode-seeking properties of vanilla and annealed Langevin dynamics sampling methods under a multi-modal distribution. We characterized Gaussian and sub-Gaussian mixture models under which Langevin dynamics are unlikely to find all the components within a sub-exponential number of iterations. To reduce the mode-seeking tendency of vanilla Langevin dynamics, we proposed Chained Langevin Dynamics (Chained-LD) and analyzed its convergence behavior. Studying the connections between Chained-LD and denoising diffusion models will be an interesting topic for future exploration.

**Limitations**

Our RNN-based implementation of Chained-LD is currently limited to image data generation tasks. An interesting future direction is to extend the application of Chained-LD to other domains such as audio and text data. Another future direction could be to study the convergence of Chained-LD under an imperfect score estimation which we did not address in our analysis.

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

## A  Theoretical Analysis on the Mode-Seeking Tendency of Langevin Dynamics

We begin by introducing some well-established lemmas used in our proof. We first provide the proof of Proposition 1 for completeness:

*Proof of Proposition 1.* By the definition in equation 2, we have

$$p_\sigma(\mathbf{z}) = \int p(\mathbf{t})\mathcal{N}(\mathbf{z} \mid \mathbf{t}, \sigma^2 \mathbf{I}_d) \, \mathrm{d}\mathbf{t} = \int p(\mathbf{t})\mathcal{N}(\mathbf{z} - \mathbf{t} \mid \mathbf{0}_d, \sigma^2 \mathbf{I}_d) \, \mathrm{d}\mathbf{t}.$$

For random variables $\mathbf{t} \sim p$ and $\mathbf{y} \sim \mathcal{N}(\mathbf{0}_d, \mathbf{I}_d)$, their sum $\mathbf{z} = \mathbf{t} + \mathbf{y} \sim p_\sigma$ follows the perturbed distribution with noise level $\sigma$. Therefore,

$$\mathbb{E}_{\mathbf{z} \sim p_\sigma}[\mathbf{z}] = \mathbb{E}_{(\mathbf{t}+\mathbf{y}) \sim p_\sigma}[\mathbf{t} + \mathbf{y}] = \mathbb{E}_{\mathbf{t} \sim p}[\mathbf{t}] + \mathbb{E}_{\mathbf{y} \sim \mathcal{N}(\mathbf{0}_d, \mathbf{I}_d)}[\mathbf{y}] = \mathbb{E}_{\mathbf{t} \sim p}[\mathbf{t}].$$

If $\mathbf{t} \sim p = \mathcal{N}(\boldsymbol{\mu}, \boldsymbol{\Sigma})$ follows a Gaussian distribution, we have $\mathbf{z} = \mathbf{t} + \mathbf{y} \sim p_\sigma = \mathcal{N}(\boldsymbol{\mu}, \boldsymbol{\Sigma} + \sigma^2 \mathbf{I}_d)$. If $p$ is a sub-Gaussian distribution with parameter $\nu^2$, we have $\mathbf{z} = \mathbf{t} + \mathbf{y} \sim p_\sigma$ is a sub-Gaussian distribution with parameter $(\nu^2 + \sigma^2)$. Hence we obtain Proposition 1. □

We use the following lemma on the tail bound for multivariate Gaussian random variables.

**Lemma 1** (Lemma 1, Laurent and Massart (2000)). *Suppose that a random variable $\mathbf{z} \sim \mathcal{N}(\mathbf{0}_d, \mathbf{I}_d)$. Then for any $\lambda > 0$,*

$$\mathbb{P}\left( \|\mathbf{z}\|^2 \geq d + 2\sqrt{d\lambda} + 2\lambda \right) \leq \exp(-\lambda),$$

$$\mathbb{P}\left( \|\mathbf{z}\|^2 \leq d - 2\sqrt{d\lambda} \right) \leq \exp(-\lambda).$$

We also use a tail bound for one-dimensional Gaussian random variables and provide the proof here for completeness.

**Lemma 2.** *Suppose a random variable $Z \sim \mathcal{N}(0, 1)$. Then for any $t > 0$,*

$$\mathbb{P}(Z \geq t) = \mathbb{P}(Z \leq -t) \leq \frac{\exp(-t^2/2)}{\sqrt{2\pi}t}.$$

*Proof of Lemma 2.* Since $\frac{z}{t} \geq 1$ for all $z \in [t, \infty)$, we have

$$\mathbb{P}(Z \geq t) = \frac{1}{\sqrt{2\pi}} \int_t^\infty \exp\left( -\frac{z^2}{2} \right) \mathrm{d}z \leq \frac{1}{\sqrt{2\pi}} \int_t^\infty \frac{z}{t} \exp\left( -\frac{z^2}{2} \right) \mathrm{d}z = \frac{\exp(-t^2/2)}{\sqrt{2\pi}t}.$$

Since the Gaussian distribution is symmetric, we have $\mathbb{P}(Z \geq t) = \mathbb{P}(Z \leq -t)$. Hence we obtain the desired bound. □

### A.1  Proof of Theorem 1: Langevin Dynamics under Gaussian Mixtures

Without loss of generality, we assume that $\boldsymbol{\mu}_0 = \mathbf{0}_d$ for simplicity. Let $r$ and $n$ respectively denote the rank and nullity of the vector space $\{\boldsymbol{\mu}_i\}_{i \in [k]}$, then we have $r + n = d$ and $0 \leq r \leq k = o(d)$. Denote $\mathbf{R} \in \mathbb{R}^{d \times r}$ an orthonormal basis of the vector space $\{\boldsymbol{\mu}_i\}_{i \in [k]}$, and denote $\mathbf{N} \in \mathbb{R}^{d \times n}$ an orthonormal basis of the null space of $\{\boldsymbol{\mu}_i\}_{i \in [k]}$. Now consider decomposing the sample $\mathbf{x}_t$ by

$$\mathbf{r}_t := \mathbf{R}^T \mathbf{x}_t, \text{ and } \mathbf{n}_t := \mathbf{N}^T \mathbf{x}_t,$$

where $\mathbf{r}_t \in \mathbb{R}^r$, $\mathbf{n}_t \in \mathbb{R}^n$. Then we have

$$\mathbf{x}_t = \mathbf{R}\mathbf{r}_t + \mathbf{N}\mathbf{n}_t.$$

Similarly, we decompose the noise $\boldsymbol{\epsilon}_t$ into

$$\boldsymbol{\epsilon}_t^{(\mathbf{r})} := \mathbf{R}^T \boldsymbol{\epsilon}_t, \text{ and } \boldsymbol{\epsilon}_t^{(\mathbf{n})} := \mathbf{N}^T \boldsymbol{\epsilon}_t,$$

where $\boldsymbol{\epsilon}_t^{(\mathbf{r})} \in \mathbb{R}^r$, $\boldsymbol{\epsilon}_t^{(\mathbf{n})} \in \mathbb{R}^n$. Then we have

$$\boldsymbol{\epsilon}_t = \mathbf{R}\boldsymbol{\epsilon}_t^{(\mathbf{r})} + \mathbf{N}\boldsymbol{\epsilon}_t^{(\mathbf{n})}.$$

439 Since a linear combination of a Gaussian random variable still follows Gaussian distribution, by
440 $\boldsymbol{\epsilon}_t \sim \mathcal{N}(\mathbf{0}_d, \boldsymbol{I}_d)$, $\mathbf{R}^T\mathbf{R} = \boldsymbol{I}_r$, and $\mathbf{N}^T\mathbf{N} = \boldsymbol{I}_n$ we obtain

$$\boldsymbol{\epsilon}_t^{(\mathbf{r})} \sim \mathcal{N}(\mathbf{0}_r, \boldsymbol{I}_r), \text{ and } \boldsymbol{\epsilon}_t^{(\mathbf{n})} \sim \mathcal{N}(\mathbf{0}_n, \boldsymbol{I}_n).$$

441 By the definition of Langevin dynamics in equation 1, the two components of $\mathbf{x}_t$ follow from the
442 update rule:

$$\mathbf{n}_t = \mathbf{n}_{t-1} + \frac{\delta_t}{2}\mathbf{N}^T\nabla_\mathbf{x} \log P(\mathbf{x}_{t-1}) + \sqrt{\delta_t}\boldsymbol{\epsilon}_t^{(\mathbf{n})}, \tag{5}$$

$$\mathbf{r}_t = \mathbf{r}_{t-1} + \frac{\delta_t}{2}\mathbf{R}^T\nabla_\mathbf{x} \log P(\mathbf{x}_{t-1}) + \sqrt{\delta_t}\boldsymbol{\epsilon}_t^{(\mathbf{r})}.$$

443 It is worth noting that since $\mathbf{N}^T\boldsymbol{\mu}_i = \mathbf{0}_n$. To show $\|\mathbf{x}_t - \boldsymbol{\mu}_i\|^2 > \frac{\nu_0^2+\nu_{\max}^2}{2}d$, it suffices to prove

$$\|\mathbf{n}_t\|^2 > \frac{\nu_0^2 + \nu_{\max}^2}{2}d.$$

444 We start by proving that the initialization of the state $\mathbf{x}_0$ has a large norm on the null space with high
445 probability in the following proposition.

446 **Proposition 2.** *Suppose that a sample $\mathbf{x}_0$ is initialized in the distribution $P^{(0)}$, i.e., $\mathbf{x}_0 \sim P^{(0)}$, then*
447 *for any constant $\nu_{\max} < \nu_0$, with probability at least $1 - \exp(-\Omega(d))$, we have $\|\mathbf{n}_0\|^2 \geq \frac{3\nu_0^2+\nu_{\max}^2}{4}d$.*

448 *Proof of Proposition 2.* Since $\mathbf{x}_0 \sim P^{(0)} = \mathcal{N}(\mathbf{0}_d, \nu_0^2\boldsymbol{I}_d)$ and $\mathbf{N}^T\mathbf{N} = \boldsymbol{I}_n$, we know $\mathbf{n}_0 = \mathbf{N}^T\mathbf{x}_0 \sim$
449 $\mathcal{N}(\mathbf{0}_n, \nu_0^2\boldsymbol{I}_n)$. Therefore, by Lemma 1 we can bound

$$\mathbb{P}\left(\|\mathbf{n}_0\|^2 \leq \frac{3\nu_0^2 + \nu_{\max}^2}{4}d\right) = \mathbb{P}\left(\frac{\|\mathbf{n}_0\|^2}{\nu_0^2} \leq d - 2\sqrt{d \cdot \left(\frac{\nu_0^2 - \nu_{\max}^2}{8\nu_0^2}\right)^2 d}\right)$$

$$\leq \mathbb{P}\left(\frac{\|\mathbf{n}_0\|^2}{\nu_0^2} \leq n - 2\sqrt{n\left(\frac{\nu_0^2 - \nu_{\max}^2}{8\nu_0^2}\right)^2 \frac{d}{2}}\right)$$

$$\leq \exp\left(-\left(\frac{\nu_0^2 - \nu_{\max}^2}{8\nu_0^2}\right)^2 \frac{d}{2}\right),$$

450 where the second last step follows from the assumption $d - n = r = o(d)$. Hence we complete the
451 proof of Proposition 2. $\qquad\square$

452 Then, with the assumption that the initialization satisfies $\|\mathbf{n}_0\|^2 \geq \frac{3\nu_0^2+\nu_{\max}^2}{4}d$, the following proposi-
453 tion shows that $\|\mathbf{n}_t\|$ remains large with high probability.

454 **Proposition 3.** *Consider a data distribution $P$ satisfies the constraints specified in Theorem 1.*
455 *We follow the Langevin dynamics for $T = \exp(\mathcal{O}(d))$ steps. Suppose that the initial sample*
456 *satisfies $\|\mathbf{n}_0\|^2 \geq \frac{3\nu_0^2+\nu_{\max}^2}{4}d$, then with probability at least $1 - T \cdot \exp(-\Omega(d))$, we have that*
457 *$\|\mathbf{n}_t\|^2 > \frac{\nu_0^2+\nu_{\max}^2}{2}d$ for all $t \in \{0\} \cup [T]$.*

458 *Proof of Proposition 3.* To establish a lower bound on $\|\mathbf{n}_t\|$, we consider different cases of the step
459 size $\delta_t$. Intuitively, when $\delta_t$ is large enough, $\mathbf{n}_t$ will be too noisy due to the introduction of random
460 noise $\sqrt{\delta_t}\boldsymbol{\epsilon}_t^{(\mathbf{n})}$ in equation 5. While for small $\delta_t$, the update of $\mathbf{n}_t$ is bounded and thus we can
461 iteratively analyze $\mathbf{n}_t$. We first handle the case of large $\delta_t$ in the following lemma.

462 **Lemma 3.** *If $\delta_t > \nu_0^2$, with probability at least $1 - \exp(-\Omega(d))$, for $\mathbf{n}_t$ satisfying equation 5, we*
463 *have $\|\mathbf{n}_t\|^2 \geq \frac{3\nu_0^2+\nu_{\max}^2}{4}d$ regardless of the previous state $\mathbf{x}_{t-1}$.*

464 *Proof of Lemma 3.* Denote $\mathbf{v} := \mathbf{n}_{t-1} + \frac{\delta_t}{2}\mathbf{N}^T\nabla_\mathbf{x} \log P(\mathbf{x}_{t-1})$ for simplicity. Note that $\mathbf{v}$ is fixed
465 for any given $\mathbf{x}_{t-1}$. We decompose $\boldsymbol{\epsilon}_t^{(\mathbf{n})}$ into a vector aligning with $\mathbf{v}$ and another vector orthogonal

466   to $\mathbf{v}$. Consider an orthonormal matrix $\mathbf{M} \in \mathbb{R}^{n \times (n-1)}$ such that $\mathbf{M}^T \mathbf{v} = \mathbf{0}_{n-1}$ and $\mathbf{M}^T \mathbf{M} = \mathbf{I}_{n-1}$.

467   By denoting $\mathbf{u} := \boldsymbol{\epsilon}_t^{(\mathbf{n})} - \mathbf{M}\mathbf{M}^T \boldsymbol{\epsilon}_t^{(\mathbf{n})}$ we have $\mathbf{M}^T \mathbf{u} = \mathbf{0}_{n-1}$, thus we obtain

$$
\begin{aligned}
\|\mathbf{n}_t\|^2 &= \left\| \mathbf{v} + \sqrt{\delta_t} \boldsymbol{\epsilon}_t^{(\mathbf{n})} \right\|^2 \\
&= \left\| \mathbf{v} + \sqrt{\delta_t} \mathbf{u} + \sqrt{\delta_t} \mathbf{M}\mathbf{M}^T \boldsymbol{\epsilon}_t^{(\mathbf{n})} \right\|^2 \\
&= \left\| \mathbf{v} + \sqrt{\delta_t} \mathbf{u} \right\|^2 + \left\| \sqrt{\delta_t} \mathbf{M}\mathbf{M}^T \boldsymbol{\epsilon}_t^{(\mathbf{n})} \right\|^2 \\
&\geq \left\| \sqrt{\delta_t} \mathbf{M}\mathbf{M}^T \boldsymbol{\epsilon}_t^{(\mathbf{n})} \right\|^2 \\
&\geq \nu_0^2 \left\| \mathbf{M}^T \boldsymbol{\epsilon}_t^{(\mathbf{n})} \right\|^2 .
\end{aligned}
$$

468   Since $\boldsymbol{\epsilon}_t^{(\mathbf{n})} \sim \mathcal{N}(\mathbf{0}_n, \mathbf{I}_n)$ and $\mathbf{M}^T \mathbf{M} = \mathbf{I}_{n-1}$, we obtain $\mathbf{M}^T \boldsymbol{\epsilon}_t^{(\mathbf{n})} \sim \mathcal{N}(\mathbf{0}_{n-1}, \mathbf{I}_{n-1})$. Therefore, by

469   Lemma 1 we can bound

$$
\begin{aligned}
\mathbb{P}\left( \|\mathbf{n}_t\|^2 \leq \frac{3\nu_0^2 + \nu_{\max}^2}{4} d \right) &\leq \mathbb{P}\left( \left\| \mathbf{M}^T \boldsymbol{\epsilon}_t^{(\mathbf{n})} \right\|^2 \leq \frac{3\nu_0^2 + \nu_{\max}^2}{4\nu_0^2} d \right) \\
&= \mathbb{P}\left( \left\| \mathbf{M}^T \boldsymbol{\epsilon}_t^{(\mathbf{n})} \right\|^2 \leq d - 2\sqrt{d \cdot \left( \frac{\nu_0^2 - \nu_{\max}^2}{8\nu_0^2} \right)^2 d} \right) \\
&\leq \mathbb{P}\left( \left\| \mathbf{M}^T \boldsymbol{\epsilon}_t^{(\mathbf{n})} \right\|^2 \leq (n-1) - 2\sqrt{(n-1)\left( \frac{\nu_0^2 - \nu_{\max}^2}{8\nu_0^2} \right)^2 \frac{d}{2}} \right) \\
&\leq \exp\left( -\left( \frac{\nu_0^2 - \nu_{\max}^2}{8\nu_0^2} \right)^2 \frac{d}{2} \right),
\end{aligned}
$$

470   where the second last step follows from the assumption $d - n = r = o(d)$. Hence we complete the

471   proof of Lemma 3. $\qquad\square$

472   We then consider the case when $\delta_t \leq \nu_0^2$. Let $\mathbf{r} := \mathbf{R}^T \mathbf{x}$ and $\mathbf{n} := \mathbf{N}^T \mathbf{x}$, then $\mathbf{x} = \mathbf{R}\mathbf{r} + \mathbf{N}\mathbf{n}$. We

473   first show that when $\|\mathbf{n}\|^2 \geq \frac{\nu_0^2 + \nu_{\max}^2}{2} d$, $P^{(i)}(\mathbf{x})$ is exponentially smaller than $P^{(0)}(\mathbf{x})$ for all $i \in [k]$

474   in the following lemma.

475   **Lemma 4.** *Given that* $\|\mathbf{n}\|^2 \geq \frac{\nu_0^2 + \nu_{\max}^2}{2} d$ *and* $\|\boldsymbol{\mu}_i\|^2 \leq \frac{\nu_0^2 - \nu_i^2}{2} \left( \log\left( \frac{\nu_i^2}{\nu_0^2} \right) - \frac{\nu_i^2}{2\nu_0^2} + \frac{\nu_0^2}{2\nu_i^2} \right) d$ *for all*

476   $i \in [k]$, *we have* $\frac{P^{(i)}(\mathbf{x})}{P^{(0)}(\mathbf{x})} \leq \exp(-\Omega(d))$ *for all* $i \in [k]$.

477   *Proof of Lemma 4.* For all $i \in [k]$, define $\rho_i(\mathbf{x}) := \frac{P^{(i)}(\mathbf{x})}{P^{(0)}(\mathbf{x})}$, then

$$
\begin{aligned}
\rho_i(\mathbf{x}) = \frac{P^{(i)}(\mathbf{x})}{P^{(0)}(\mathbf{x})} &= \frac{(2\pi\nu_i^2)^{-d/2} \exp\left( -\frac{1}{2\nu_i^2} \|\mathbf{x} - \boldsymbol{\mu}_i\|^2 \right)}{(2\pi\nu_0^2)^{-d/2} \exp\left( -\frac{1}{2\nu_0^2} \|\mathbf{x}\|^2 \right)} \\
&= \left( \frac{\nu_0^2}{\nu_i^2} \right)^{d/2} \exp\left( \frac{1}{2\nu_0^2} \|\mathbf{x}\|^2 - \frac{1}{2\nu_i^2} \|\mathbf{x} - \boldsymbol{\mu}_i\|^2 \right) \\
&= \left( \frac{\nu_0^2}{\nu_i^2} \right)^{d/2} \exp\left( \left( \frac{1}{2\nu_0^2} - \frac{1}{2\nu_i^2} \right) \|\mathbf{N}\mathbf{n}\|^2 + \left( \frac{\|\mathbf{R}\mathbf{r}\|^2}{2\nu_0^2} - \frac{\|\mathbf{R}\mathbf{r} - \boldsymbol{\mu}_i\|^2}{2\nu_i^2} \right) \right) \\
&= \left( \frac{\nu_0^2}{\nu_i^2} \right)^{d/2} \exp\left( \left( \frac{1}{2\nu_0^2} - \frac{1}{2\nu_i^2} \right) \|\mathbf{n}\|^2 + \left( \frac{\|\mathbf{r}\|^2}{2\nu_0^2} - \frac{\left\| \mathbf{r} - \mathbf{R}^T \boldsymbol{\mu}_i \right\|^2}{2\nu_i^2} \right) \right),
\end{aligned}
$$

478   where the last step follows from the definition that $\mathbf{R} \in \mathbb{R}^{d \times r}$ an orthonormal basis of the vector space

479   $\{\boldsymbol{\mu}_i\}_{i \in [k]}$ and $\mathbf{N}^T \mathbf{N} = \mathbf{I}_n$. Since $\nu_0^2 > \nu_i^2$, the quadratic term $\frac{\|\mathbf{r}\|^2}{2\nu_0^2} - \frac{\left\| \mathbf{r} - \mathbf{R}^T \boldsymbol{\mu}_i \right\|^2}{2\nu_i^2}$ is maximized at

480    $\mathbf{r} = \frac{\nu_0^2 \mathbf{R}^T \boldsymbol{\mu}_i}{\nu_0^2 - \nu_i^2}$. Therefore,

$$\frac{\|\mathbf{r}\|^2}{2\nu_0^2} - \frac{\|\mathbf{r} - \mathbf{R}^T \boldsymbol{\mu}_i\|^2}{2\nu_i^2} \leq \frac{\nu_0^4 \|\mathbf{R}^T \boldsymbol{\mu}_i\|^2}{2\nu_0^2(\nu_0^2 - \nu_i^2)^2} - \frac{1}{2\nu_i^2}\left(\frac{\nu_0^2}{\nu_0^2 - \nu_i^2} - 1\right)^2 \|\mathbf{R}^T \boldsymbol{\mu}_i\|^2 = \frac{\|\boldsymbol{\mu}_i\|^2}{2(\nu_0^2 - \nu_i^2)}.$$

481    Hence, for $\|\mathbf{n}\|^2 \geq \frac{\nu_0^2 + \nu_{\max}^2}{2} d$ and $\|\boldsymbol{\mu}_i\|^2 \leq \frac{\nu_0^2 - \nu_i^2}{2}\left(\log\left(\frac{\nu_i^2}{\nu_0^2}\right) - \frac{\nu_i^2}{2\nu_0^2} + \frac{\nu_0^2}{2\nu_i^2}\right) d$, we have

$$\rho_i(\mathbf{x}) = \left(\frac{\nu_0^2}{\nu_i^2}\right)^{d/2} \exp\left(\left(\frac{1}{2\nu_0^2} - \frac{1}{2\nu_i^2}\right)\|\mathbf{n}\|^2 + \left(\frac{\|\mathbf{r}\|^2}{2\nu_0^2} - \frac{\|\mathbf{r} - \mathbf{R}^T \boldsymbol{\mu}_i\|^2}{2\nu_i^2}\right)\right)$$

$$\leq \left(\frac{\nu_0^2}{\nu_i^2}\right)^{d/2} \exp\left(\left(\frac{1}{2\nu_0^2} - \frac{1}{2\nu_i^2}\right)\frac{\nu_0^2 + \nu_i^2}{2}d + \frac{\|\boldsymbol{\mu}_i\|^2}{2(\nu_0^2 - \nu_i^2)}\right)$$

$$= \exp\left(-\left(\log\left(\frac{\nu_i^2}{\nu_0^2}\right) - \frac{\nu_i^2}{2\nu_0^2} + \frac{\nu_0^2}{2\nu_i^2}\right)\frac{d}{2} + \frac{\|\boldsymbol{\mu}_i\|^2}{2(\nu_0^2 - \nu_i^2)}\right)$$

$$\leq \exp\left(-\left(\log\left(\frac{\nu_i^2}{\nu_0^2}\right) - \frac{\nu_i^2}{2\nu_0^2} + \frac{\nu_0^2}{2\nu_i^2}\right)\frac{d}{4}\right).$$

482    Notice that for function $f(z) = \log z - \frac{z}{2} + \frac{1}{2z}$, we have $f(1) = 0$ and $\frac{\mathrm{d}}{\mathrm{d}z} f(z) = \frac{1}{z} - \frac{1}{2} - \frac{1}{2z^2} =$
483    $-\frac{1}{2}\left(\frac{1}{z} - 1\right)^2 < 0$ when $z \in (0, 1)$. Thus, $\log\left(\frac{\nu_i^2}{\nu_0^2}\right) - \frac{\nu_i^2}{2\nu_0^2} + \frac{\nu_0^2}{2\nu_i^2}$ is a positive constant for $\nu_i < \nu_0$,
484    i.e., $\rho_i(\mathbf{x}) = \exp(-\Omega(d))$. Therefore we finish the proof of Lemma 4. $\qquad\square$

485    Lemma 4 implies that when $\|\mathbf{n}\|$ is large, the Gaussian mode $P^{(0)}$ dominates other modes $P^{(i)}$. To
486    bound $\|\mathbf{n}_t\|$, we first consider a simpler case that $\|\mathbf{n}_{t-1}\|$ is large. Intuitively, the following lemma
487    proves that when the previous state $\mathbf{n}_{t-1}$ is far from a mode, a single step of Langevin dynamics with
488    bounded step size is not enough to find the mode.

489    **Lemma 5.** *Suppose* $\delta_t \leq \nu_0^2$ *and* $\|\mathbf{n}_{t-1}\|^2 > 36\nu_0^2 d$, *then for* $\mathbf{n}_t$ *following from equation 5, we have*
490    $\|\mathbf{n}_t\|^2 \geq \nu_0^2 d$ *with probability at least* $1 - \exp(-\Omega(d))$.

491    *Proof of Lemma 5.* From the recursion of $\mathbf{n}_t$ in equation 5 we have

$$\mathbf{n}_t = \mathbf{n}_{t-1} + \frac{\delta_t}{2}\mathbf{N}^T \nabla_{\mathbf{x}} \log P(\mathbf{x}_{t-1}) + \sqrt{\delta_t}\boldsymbol{\epsilon}_t^{(\mathbf{n})}$$

$$= \mathbf{n}_{t-1} - \frac{\delta_t}{2}\sum_{i=0}^{k}\frac{P^{(i)}(\mathbf{x}_{t-1})}{P(\mathbf{x}_{t-1})} \cdot \frac{\mathbf{N}^T(\mathbf{x}_{t-1} - \boldsymbol{\mu}_i)}{\nu_i^2} + \sqrt{\delta_t}\boldsymbol{\epsilon}_t^{(\mathbf{n})}$$

$$= \left(1 - \frac{\delta_t}{2}\sum_{i=0}^{k}\frac{P^{(i)}(\mathbf{x}_{t-1})}{P(\mathbf{x}_{t-1})} \cdot \frac{1}{\nu_i^2}\right)\mathbf{n}_{t-1} + \sqrt{\delta_t}\boldsymbol{\epsilon}_t^{(\mathbf{n})}. \tag{6}$$

492    By Lemma 4, we have $\frac{P^{(i)}(\mathbf{x}_{j-1})}{P^{(0)}(\mathbf{x}_{j-1})} \leq \exp(-\Omega(d))$ for all $i \in [k]$, therefore

$$1 - \frac{\delta_t}{2}\sum_{i=0}^{k}\frac{P^{(i)}(\mathbf{x}_{t-1})}{P(\mathbf{x}_{t-1})} \cdot \frac{1}{\nu_i^2} \geq 1 - \frac{\delta_t}{2} \cdot \frac{1}{\nu_0^2} - \frac{\delta_t}{2}\sum_{i \in [k]}\frac{w_i P^{(i)}(\mathbf{x}_{t-1})}{w_0 P^{(0)}(\mathbf{x}_{t-1})} \cdot \frac{1}{\nu_i^2} \geq 1 - \frac{1}{2} - \exp(-\Omega(d)) > \frac{1}{3}. \tag{7}$$

493    On the other hand, from $\boldsymbol{\epsilon}_t^{(\mathbf{n})} \sim \mathcal{N}(\mathbf{0}_n, \boldsymbol{I}_n)$ we know $\frac{\langle \mathbf{n}_{t-1}, \boldsymbol{\epsilon}_t^{(\mathbf{n})}\rangle}{\|\mathbf{n}_{t-1}\|} \sim \mathcal{N}(0, 1)$ for any fixed $\mathbf{n}_{t-1} \neq \mathbf{0}_n$,
494    hence by Lemma 2 we have

$$\mathbb{P}\left(\frac{\langle \mathbf{n}_{t-1}, \boldsymbol{\epsilon}_t^{(\mathbf{n})}\rangle}{\|\mathbf{n}_{t-1}\|} \geq \frac{\sqrt{d}}{4}\right) = \mathbb{P}\left(\frac{\langle \mathbf{n}_{t-1}, \boldsymbol{\epsilon}_t^{(\mathbf{n})}\rangle}{\|\mathbf{n}_{t-1}\|} \leq -\frac{\sqrt{d}}{4}\right) \leq \frac{4}{\sqrt{2\pi d}}\exp\left(-\frac{d}{32}\right) \tag{8}$$

Combining equation 6, equation 7 and equation 8 gives that

$$\|\mathbf{n}_t\|^2 \geq \left(\frac{1}{3}\right)^2 \|\mathbf{n}_{t-1}\|^2 - 2\nu_0 |\langle \mathbf{n}_{t-1}, \boldsymbol{\epsilon}_t^{(\mathbf{n})}\rangle|$$

$$\geq \frac{1}{9}\|\mathbf{n}_{t-1}\|^2 - \frac{\nu_0\sqrt{d}}{2}\|\mathbf{n}_{t-1}\|$$

$$\geq \frac{1}{9} \cdot 36\nu_0^2 d - \frac{\nu_0\sqrt{d}}{2} \cdot 6\nu_0\sqrt{d}$$

$$= \nu_0^2 d$$

with probability at least $1 - \frac{8}{\sqrt{2\pi d}}\exp\left(-\frac{d}{32}\right) = 1 - \exp(-\Omega(d))$. This proves Lemma 5. $\qquad\square$

We then proceed to bound $\|\mathbf{n}_t\|$ iteratively for $\|\mathbf{n}_{t-1}\|^2 \leq 36\nu_0^2 d$. Recall that equation 5 gives

$$\mathbf{n}_t = \mathbf{n}_{t-1} + \frac{\delta_t}{2}\mathbf{N}^T \nabla_{\mathbf{x}} \log P(\mathbf{x}_{t-1}) + \sqrt{\delta_t}\boldsymbol{\epsilon}_t^{(\mathbf{n})}.$$

We notice that the difficulty of solving $\mathbf{n}_t$ exhibits in the dependence of $\log P(\mathbf{x}_{t-1})$ on $\mathbf{r}_{t-1}$. Since $P = \sum_{i=0}^k w_i P^{(i)} = \sum_{i=0}^k w_i \mathcal{N}(\boldsymbol{\mu}_i, \nu_i^2 \mathbf{I}_d)$, we can rewrite the score function as

$$\nabla_{\mathbf{x}} \log P(\mathbf{x}) = \frac{\nabla_{\mathbf{x}} P(\mathbf{x})}{P(\mathbf{x})} = -\sum_{i=0}^k \frac{P^{(i)}(\mathbf{x})}{P(\mathbf{x})} \cdot \frac{\mathbf{x} - \boldsymbol{\mu}_i}{\nu_i^2} = -\frac{\mathbf{x}}{\nu_0^2} + \sum_{i \in [k]} \frac{P^{(i)}(\mathbf{x})}{P(\mathbf{x})}\left(\frac{\mathbf{x}}{\nu_0^2} - \frac{\mathbf{x} - \boldsymbol{\mu}_i}{\nu_i^2}\right). \tag{9}$$

Now, instead of directly working with $\mathbf{n}_t$, we consider a surrogate recursion $\hat{\mathbf{n}}_t$ such that $\hat{\mathbf{n}}_0 = \mathbf{n}_0$ and for all $t \geq 1$,

$$\hat{\mathbf{n}}_t = \hat{\mathbf{n}}_{t-1} - \frac{\delta_t}{2\nu_0^2}\hat{\mathbf{n}}_{t-1} + \sqrt{\delta_t}\boldsymbol{\epsilon}_t^{(\mathbf{n})}. \tag{10}$$

The advantage of the surrogate recursion is that $\hat{\mathbf{n}}_t$ is independent of $\mathbf{r}$, thus we can obtain the closed-form solution to $\hat{\mathbf{n}}_t$. Before we proceed to bound $\hat{\mathbf{n}}_t$, we first show that $\hat{\mathbf{n}}_t$ is sufficiently close to the original recursion $\mathbf{n}_t$ in the following lemma.

**Lemma 6.** *For any $t \geq 1$, given that $\delta_j \leq \nu_0^2$ and $\frac{\nu_0^2 + \nu_{\max}^2}{2}d \leq \|\mathbf{n}_{j-1}\|^2 \leq 36\nu_0^2 d$ for all $j \in [t]$ and $\|\boldsymbol{\mu}_i\|^2 \leq \frac{\nu_0^2 - \nu_i^2}{2}\left(\log\left(\frac{\nu_i^2}{\nu_0^2}\right) - \frac{\nu_i^2}{2\nu_0^2} + \frac{\nu_0^2}{2\nu_i^2}\right)d$ for all $i \in [k]$, we have $\|\hat{\mathbf{n}}_t - \mathbf{n}_t\| \leq \frac{t}{\exp(\Omega(d))}\sqrt{d}$.*

*Proof of Lemma 6.* Upon comparing equation 5 and equation 10, by equation 9 we have that for all $j \in [t]$,

$$\|\hat{\mathbf{n}}_j - \mathbf{n}_j\| = \left\|\hat{\mathbf{n}}_{j-1} - \frac{\delta_j}{2\nu_0^2}\hat{\mathbf{n}}_{j-1} - \mathbf{n}_{j-1} - \frac{\delta_j}{2}\mathbf{N}^T\nabla_{\mathbf{x}}\log P(\mathbf{x}_{j-1})\right\|$$

$$= \left\|\left(1 - \frac{\delta_j}{2\nu_0^2}\right)(\hat{\mathbf{n}}_{j-1} - \mathbf{n}_{j-1}) + \frac{\delta_j}{2}\sum_{i \in [k]}\frac{P^{(i)}(\mathbf{x}_{j-1})}{P(\mathbf{x}_{j-1})}\left(\frac{1}{\nu_i^2} - \frac{1}{\nu_0^2}\right)\mathbf{n}_{j-1}\right\|$$

$$\leq \left(1 - \frac{\delta_j}{2\nu_0^2}\right)\|\hat{\mathbf{n}}_{j-1} - \mathbf{n}_{j-1}\| + \sum_{i \in [k]}\frac{\delta_j}{2}\frac{P^{(i)}(\mathbf{x}_{j-1})}{P(\mathbf{x}_{j-1})}\left(\frac{1}{\nu_i^2} - \frac{1}{\nu_0^2}\right)\|\mathbf{n}_{j-1}\|$$

$$\leq \|\hat{\mathbf{n}}_{j-1} - \mathbf{n}_{j-1}\| + \sum_{i \in [k]}\frac{\delta_j}{2}\frac{P^{(i)}(\mathbf{x}_{j-1})}{P^{(0)}(\mathbf{x}_{j-1})}\left(\frac{1}{\nu_i^2} - \frac{1}{\nu_0^2}\right)6\nu_0\sqrt{d}.$$

By Lemma 4, we have $\frac{P^{(i)}(\mathbf{x}_{j-1})}{P^{(0)}(\mathbf{x}_{j-1})} \leq \exp(-\Omega(d))$ for all $i \in [k]$, hence we obtain a recursive bound

$$\|\hat{\mathbf{n}}_j - \mathbf{n}_j\| \leq \|\hat{\mathbf{n}}_{j-1} - \mathbf{n}_{j-1}\| + \frac{1}{\exp(\Omega(d))}\sqrt{d}.$$

510   Finally, by $\hat{\mathbf{n}}_0 = \mathbf{n}_0$, we have

$$\|\hat{\mathbf{n}}_t - \mathbf{n}_t\| = \sum_{j \in [t]} (\|\hat{\mathbf{n}}_j - \mathbf{n}_j\| - \|\hat{\mathbf{n}}_{j-1} - \mathbf{n}_{j-1}\|) \leq \frac{t}{\exp(\Omega(d))} \sqrt{d}.$$

511   Hence we obtain Lemma 6. $\qquad\qquad\qquad\qquad\qquad\qquad\qquad\qquad\qquad\qquad\qquad\quad\square$

512   We then proceed to analyze $\hat{\mathbf{n}}_t$, The following lemma gives us the closed-form solution of $\hat{\mathbf{n}}_t$. We
513   slightly abuse the notations here, e.g., $\prod_{i=c_1}^{c_2} \left(1 - \frac{\delta_i}{2\nu_0^2}\right) = 1$ and $\sum_{j=c_1}^{c_2} \delta_j = 0$ for $c_1 > c_2$.

514   **Lemma 7.** *For all $t \geq 0$, $\hat{\mathbf{n}}_t \sim \mathcal{N}\left(\prod_{i=1}^t \left(1 - \frac{\delta_i}{2\nu_0^2}\right) \mathbf{n}_0, \ \sum_{j=1}^t \prod_{i=j+1}^t \left(1 - \frac{\delta_i}{2\nu_0^2}\right)^2 \delta_j \mathbf{I}_n\right)$, where*

515   *the mean and covariance satisfy $\prod_{i=1}^t \left(1 - \frac{\delta_i}{2\nu_0^2}\right)^2 + \frac{1}{\nu_0^2} \sum_{j=1}^t \prod_{i=j+1}^t \left(1 - \frac{\delta_i}{2\nu_0^2}\right)^2 \delta_j \geq 1$.*

516   *Proof of Lemma 7.* We prove the two properties by induction. When $t = 0$, they are trivial. Suppose
517   they hold for $t - 1$, then for the distribution of $\hat{\mathbf{n}}_t$, we have

$$\hat{\mathbf{n}}_t = \hat{\mathbf{n}}_{t-1} - \frac{\delta_t}{2\nu_0^2} \hat{\mathbf{n}}_{t-1} + \sqrt{\delta_t} \boldsymbol{\epsilon}_t^{(\mathbf{n})}$$

$$\sim \mathcal{N}\left(\left(1 - \frac{\delta_t}{2\nu_0^2}\right) \prod_{i=1}^{t-1} \left(1 - \frac{\delta_i}{2\nu_0^2}\right) \mathbf{n}_0, \ \left(1 - \frac{\delta_t}{2\nu_0^2}\right)^2 \sum_{j=1}^{t-1} \prod_{i=j+1}^{t-1} \left(1 - \frac{\delta_i}{2\nu_0^2}\right)^2 \delta_j \mathbf{I}_n + \delta_t \mathbf{I}_n\right)$$

$$= \mathcal{N}\left(\prod_{i=1}^t \left(1 - \frac{\delta_i}{2\nu_0^2}\right) \mathbf{n}_0, \ \sum_{j=1}^t \prod_{i=j+1}^t \left(1 - \frac{\delta_i}{2\nu_0^2}\right)^2 \delta_j \mathbf{I}_n\right).$$

518   For the second property,

$$\prod_{i=1}^t \left(1 - \frac{\delta_i}{2\nu_0^2}\right)^2 + \frac{1}{\nu_0^2} \sum_{j=1}^t \prod_{i=j+1}^t \left(1 - \frac{\delta_i}{2\nu_0^2}\right)^2 \delta_j$$

$$= \left(1 - \frac{\delta_t}{2\nu_0^2}\right)^2 \left(\prod_{i=1}^{t-1} \left(1 - \frac{\delta_i}{2\nu_0^2}\right)^2 + \frac{1}{\nu_0^2} \sum_{j=1}^{t-1} \prod_{i=j+1}^{t-1} \left(1 - \frac{\delta_i}{2\nu_0^2}\right)^2 \delta_j\right) + \frac{1}{\nu_0^2} \delta_t$$

$$\geq \left(1 - \frac{\delta_t}{2\nu_0^2}\right)^2 + \frac{1}{\nu_0^2} \delta_t = 1 + \frac{\delta_t^2}{4\nu_0^4} \geq 1.$$

519   Hence we finish the proof of Lemma 7. $\qquad\qquad\qquad\qquad\qquad\qquad\qquad\qquad\qquad\quad\square$

520   Armed with Lemma 7, we are now ready to establish the lower bound on $\|\hat{\mathbf{n}}_t\|$. For simplicity,
521   denote $\alpha := \prod_{i=1}^t \left(1 - \frac{\delta_i}{2\nu_0^2}\right)^2$ and $\beta := \frac{1}{\nu_0^2} \sum_{j=1}^t \prod_{i=j+1}^t \left(1 - \frac{\delta_i}{2\nu_0^2}\right)^2 \delta_j$. By Lemma 7 we know
522   $\hat{\mathbf{n}}_t \sim \mathcal{N}(\alpha \mathbf{n}_0, \beta \nu_0^2 \mathbf{I}_n)$, so we can write $\hat{\mathbf{n}}_t = \alpha \mathbf{n}_0 + \sqrt{\beta} \nu_0 \boldsymbol{\epsilon}$, where $\boldsymbol{\epsilon} \sim \mathcal{N}(\mathbf{0}_n, \mathbf{I}_n)$.

523   **Lemma 8.** *Given that $\|\hat{\mathbf{n}}_0\|^2 \geq \frac{3\nu_0^2 + \nu_{\max}^2}{4} d$, we have $\|\hat{\mathbf{n}}_t\|^2 \geq \frac{5\nu_0^2 + 3\nu_{\max}^2}{8} d$ with probability at least*
524   $1 - \exp(-\Omega(d))$.

525   *Proof of Lemma 8.* By $\hat{\mathbf{n}}_t = \alpha \mathbf{n}_0 + \sqrt{\beta} \nu_0 \boldsymbol{\epsilon}$ we have

$$\|\hat{\mathbf{n}}_t\|^2 = \alpha^2 \|\mathbf{n}_0\|^2 + \beta \nu_0^2 \|\boldsymbol{\epsilon}\|^2 + 2\alpha \sqrt{\beta} \nu_0 \langle \mathbf{n}_0, \boldsymbol{\epsilon} \rangle$$

By Lemma 1 we can bound

$$\mathbb{P}\left(\|\boldsymbol{\epsilon}\|^2 \leq \frac{3\nu_0^2 + \nu_{\max}^2}{4\nu_0^2}d\right) = \mathbb{P}\left(\|\boldsymbol{\epsilon}\|^2 \leq d - 2\sqrt{d \cdot \left(\frac{\nu_0^2 - \nu_{\max}^2}{8\nu_0^2}\right)^2 d}\right)$$

$$\leq \mathbb{P}\left(\|\boldsymbol{\epsilon}\|^2 \leq (n-1) - 2\sqrt{(n-1)\left(\frac{\nu_0^2 - \nu_{\max}^2}{8\nu_0^2}\right)^2 \frac{d}{2}}\right)$$

$$\leq \exp\left(-\left(\frac{\nu_0^2 - \nu_{\max}^2}{8\nu_0^2}\right)^2 \frac{d}{2}\right),$$

where the second last step follows from the assumption $d - n = r = o(d)$. Since $\boldsymbol{\epsilon} \sim \mathcal{N}(\mathbf{0}_n, \boldsymbol{I}_n)$, we know $\frac{\langle \mathbf{n}_0, \boldsymbol{\epsilon} \rangle}{\|\mathbf{n}_0\|} \sim \mathcal{N}(0, 1)$. Therefore by Lemma 2,

$$\mathbb{P}\left(\frac{\langle \mathbf{n}_0, \boldsymbol{\epsilon} \rangle}{\|\mathbf{n}_0\|} \leq -\frac{\nu_0^2 - \nu_{\max}^2}{4\nu_0\sqrt{3\nu_0^2 + \nu_{\max}^2}}\sqrt{d}\right) \leq \frac{4\nu_0\sqrt{3\nu_0^2 + \nu_{\max}^2}}{\sqrt{2\pi}(\nu_0^2 - \nu_{\max}^2)\sqrt{d}}\exp\left(-\frac{(\nu_0^2 - \nu_{\max}^2)^2 d}{32\nu_0^2(3\nu_0^2 + \nu_{\max}^2)}\right)$$

Conditioned on $\|\hat{\mathbf{n}}_0\|^2 \geq \frac{3\nu_0^2 + \nu_{\max}^2}{4}d$, $\|\boldsymbol{\epsilon}\|^2 > \frac{3\nu_0^2 + \nu_{\max}^2}{4\nu_0^2}d$ and $\frac{1}{\|\mathbf{n}_0\|}\langle \mathbf{n}_0, \boldsymbol{\epsilon} \rangle > -\frac{\nu_0^2 - \nu_{\max}^2}{4\nu_0\sqrt{3\nu_0^2 + \nu_{\max}^2}}\sqrt{d}$, since Lemma 7 gives $\alpha^2 + \beta \geq 1$ we have

$$\|\hat{\mathbf{n}}_t\|^2 = \alpha^2\|\mathbf{n}_0\|^2 + \beta\nu_0^2\|\boldsymbol{\epsilon}\|^2 + 2\alpha\sqrt{\beta}\nu_0\langle \mathbf{n}_0, \boldsymbol{\epsilon} \rangle$$

$$\geq \alpha^2\|\mathbf{n}_0\|^2 + \beta\nu_0^2\|\boldsymbol{\epsilon}\|^2 - 2\alpha\sqrt{\beta}\nu_0\|\mathbf{n}_0\|\frac{\nu_0^2 - \nu_{\max}^2}{4\nu_0\sqrt{3\nu_0^2 + \nu_{\max}^2}}\sqrt{d}$$

$$\geq \alpha^2\|\mathbf{n}_0\|^2 + \beta\nu_0^2\|\boldsymbol{\epsilon}\|^2 - 2\alpha\sqrt{\beta}\nu_0\|\mathbf{n}_0\|\|\boldsymbol{\epsilon}\| \cdot \frac{\nu_0^2 - \nu_{\max}^2}{6\nu_0^2 + 2\nu_{\max}^2}$$

$$\geq \left(1 - \frac{\nu_0^2 - \nu_{\max}^2}{6\nu_0^2 + 2\nu_{\max}^2}\right)\left(\alpha^2\|\mathbf{n}_0\|^2 + \beta\nu_0^2\|\boldsymbol{\epsilon}\|^2\right)$$

$$\geq \frac{5\nu_0^2 + 3\nu_{\max}^2}{6\nu_0^2 + 2\nu_{\max}^2}\left(\alpha^2 + \beta\right) \cdot \frac{3\nu_0^2 + \nu_{\max}^2}{4}d$$

$$\geq \frac{5\nu_0^2 + 3\nu_{\max}^2}{8}d.$$

Hence by union bound, we complete the proof of Lemma 8. $\qquad\square$

Upon having all the above lemmas, we are now ready to establish Proposition 3 by induction. Suppose the theorem holds for all $T$ values of $1, \cdots, T-1$. We consider the following 3 cases:

- If there exists some $t \in [T]$ such that $\delta_t > \nu_0^2$, by Lemma 3 we know that with probability at least $1 - \exp(-\Omega(d))$, we have $\|\mathbf{n}_t\|^2 \geq \frac{3\nu_0^2 + \nu_{\max}^2}{4}d$, thus the problem reduces to the two sub-arrays $\mathbf{n}_0, \cdots, \mathbf{n}_{t-1}$ and $\mathbf{n}_t, \cdots, \mathbf{n}_T$, which can be solved by induction.

- Suppose $\delta_t \leq \nu_0^2$ for all $t \in [T]$. If there exists some $t \in [T]$ such that $\|\mathbf{n}_{t-1}\|^2 > 36\nu_0^2 d$, by Lemma 5 we know that with probability at least $1 - \exp(-\Omega(d))$, we have $\|\mathbf{n}_t\|^2 \geq \nu_0^2 d > \frac{3\nu_0^2 + \nu_{\max}^2}{4}d$, thus the problem similarly reduces to the two sub-arrays $\mathbf{n}_0, \cdots, \mathbf{n}_{t-1}$ and $\mathbf{n}_t, \cdots, \mathbf{n}_T$, which can be solved by induction.

- Suppose $\delta_t \leq \nu_0^2$ and $\|\mathbf{n}_{t-1}\|^2 \leq 36\nu_0^2 d$ for all $t \in [T]$. Conditioned on $\|\mathbf{n}_{t-1}\|^2 > \frac{\nu_0^2 + \nu_{\max}^2}{2}d$ for all $t \in [T]$, by Lemma 6 we have that for $T = \exp(\mathcal{O}(d))$,

$$\|\hat{\mathbf{n}}_T - \mathbf{n}_T\| < \left(\sqrt{\frac{5\nu_0^2 + 3\nu_{\max}^2}{8}} - \sqrt{\frac{\nu_0^2 + \nu_{\max}^2}{2}}\right)\sqrt{d}.$$

By Lemma 8 we have that with probability at least $1 - \exp(-\Omega(d))$,

$$\|\hat{\mathbf{n}}_T\|^2 \geq \frac{5\nu_0^2 + 3\nu_{\max}^2}{8}d.$$

Combining the two inequalities implies the desired bound

$$\|\mathbf{n}_T\| \geq \|\hat{\mathbf{n}}_T\| - \|\hat{\mathbf{n}}_T - \mathbf{n}_T\| > \sqrt{\frac{\nu_0^2 + \nu_{\max}^2}{2}} d.$$

Hence by induction we obtain $\|\mathbf{n}_t\|^2 > \frac{\nu_0^2 + \nu_{\max}^2}{2} d$ for all $t \in [T]$ with probability at least

$$(1 - (T - 1)\exp(-\Omega(d))) \cdot (1 - \exp(-\Omega(d))) \geq 1 - T\exp(-\Omega(d)).$$

Therefore we complete the proof of Proposition 3. $\qquad\square$

Finally, combining Propositions 2 and 3 finishes the proof of Theorem 1.

## A.2 Proof of Theorem 2: Annealed Langevin Dynamics under Gaussian Mixtures

To establish Theorem 2, we first note from Proposition 1 that perturbing a Gaussian distribution $\mathcal{N}(\boldsymbol{\mu}, \nu^2 \mathbf{I}_d)$ with noise level $\sigma$ results in a Gaussian distribution $\mathcal{N}(\boldsymbol{\mu}, (\nu^2 + \sigma^2)\mathbf{I}_d)$. Therefore, for a Gaussian mixture $P = \sum_{i=0}^{k} w_i P^{(i)} = \sum_{i=0}^{k} w_i \mathcal{N}(\boldsymbol{\mu}_i, \nu_i^2 \mathbf{I}_d)$, the perturbed distribution of noise level $\sigma$ is

$$P_\sigma = \sum_{i=0}^{k} w_i \mathcal{N}(\boldsymbol{\mu}_i, (\nu_i^2 + \sigma^2)\mathbf{I}_d).$$

Similar to the proof of Theorem 1, we decompose

$$\mathbf{x}_t = \mathbf{R}\mathbf{r}_t + \mathbf{N}\mathbf{n}_t, \text{ and } \boldsymbol{\epsilon}_t = \mathbf{R}\boldsymbol{\epsilon}_t^{(\mathbf{r})} + \mathbf{N}\boldsymbol{\epsilon}_t^{(\mathbf{n})},$$

where $\mathbf{R} \in \mathbb{R}^{d \times r}$ an orthonormal basis of the vector space $\{\boldsymbol{\mu}_i\}_{i \in [k]}$ and $\mathbf{N} \in \mathbb{R}^{d \times n}$ an orthonormal basis of the null space of $\{\boldsymbol{\mu}_i\}_{i \in [k]}$. Now, we prove Theorem 2 by applying the techniques developed in Appendix A.1 via substituting $\nu^2$ with $\nu^2 + \sigma_t^2$ at time step $t$.

First, by Proposition 2, suppose that the sample is initialized in the distribution $P_{\sigma_0}^{(0)}$, then with probability at least $1 - \exp(-\Omega(d))$, we have

$$\|\mathbf{n}_0\|^2 \geq \frac{3(\nu_0^2 + \sigma_0^2) + (\nu_{\max}^2 + \sigma_0^2)}{4} d = \frac{3\nu_0^2 + \nu_{\max}^2 + 4\sigma_0^2}{4} d. \tag{11}$$

Then, with the assumption that the initialization satisfies $\|\mathbf{n}_0\|^2 \geq \frac{3\nu_0^2 + \nu_{\max}^2 + 4\sigma_0^2}{4} d$, the following proposition similar to Proposition 3 shows that $\|\mathbf{n}_t\|$ remains large with high probability.

**Proposition 4.** *Consider a data distribution $P$ satisfies the constraints specified in Theorem 2. We follow annealed Langevin dynamics for $T = \exp(\mathcal{O}(d))$ steps with noise level $c_\sigma \geq \sigma_0 \geq \sigma_1 \geq \sigma_2 \geq \cdots \geq \sigma_T \geq 0$ for some constant $c_\sigma > 0$. Suppose that the initial sample satisfies $\|\mathbf{n}_0\|^2 \geq \frac{3\nu_0^2 + \nu_{\max}^2 + 4\sigma_0^2}{4} d$, then with probability at least $1 - T \cdot \exp(-\Omega(d))$, we have that $\|\mathbf{n}_t\|^2 > \frac{\nu_0^2 + \nu_{\max}^2 + 2\sigma_t^2}{2} d$ for all $t \in \{0\} \cup [T]$.*

*Proof of Proposition 4.* We prove Proposition 4 by induction. Suppose the theorem holds for all $T$ values of $1, \cdots, T - 1$. We consider the following 3 cases:

- If there exists some $t \in [T]$ such that $\delta_t > \nu_0^2 + \sigma_t^2$, by Lemma 3 we know that with probability at least $1 - \exp(-\Omega(d))$, we have $\|\mathbf{n}_t\|^2 \geq \frac{3(\nu_0^2 + \sigma_t^2) + (\nu_{\max}^2 + \sigma_t^2)}{4} d = \frac{3\nu_0^2 + \nu_{\max}^2 + 4\sigma_t^2}{4} d$, thus the problem reduces to the two sub-arrays $\mathbf{n}_0, \cdots, \mathbf{n}_{t-1}$ and $\mathbf{n}_t, \cdots, \mathbf{n}_T$, which can be solved by induction.

- Suppose $\delta_t \leq \nu_0^2 + \sigma_t^2$ for all $t \in [T]$. If there exists some $t \in [T]$ such that $\|\mathbf{n}_{t-1}\|^2 > 36(\nu_0^2 + \sigma_{t-1}^2)d \geq 36(\nu_0^2 + \sigma_t^2)d$, by Lemma 5 we know that with probability at least $1 - \exp(-\Omega(d))$, we have $\|\mathbf{n}_t\|^2 \geq (\nu_0^2 + \sigma_t^2)d > \frac{3\nu_0^2 + \nu_{\max}^2 + 4\sigma_t^2}{4} d$, thus the problem similarly reduces to the two sub-arrays $\mathbf{n}_0, \cdots, \mathbf{n}_{t-1}$ and $\mathbf{n}_t, \cdots, \mathbf{n}_T$, which can be solved by induction.

- Suppose $\delta_t \le \nu_0^2 + \sigma_t^2$ and $\|\mathbf{n}_{t-1}\|^2 \le 36(\nu_0^2 + \sigma_{t-1}^2)d$ for all $t \in [T]$. Consider a surrogate sequence $\hat{\mathbf{n}}_t$ such that $\hat{\mathbf{n}}_0 = \mathbf{n}_0$ and for all $t \ge 1$,

$$\hat{\mathbf{n}}_t = \hat{\mathbf{n}}_{t-1} - \frac{\delta_t}{2\nu_0^2 + 2\sigma_t^2}\hat{\mathbf{n}}_{t-1} + \sqrt{\delta_t}\boldsymbol{\epsilon}_t^{(\mathbf{n})}.$$

Since $\nu_0 > \nu_i$ and $c_\sigma \ge \sigma_t$ for all $t \in \{0\} \cup [T]$, we have $\frac{\nu_i^2 + c_\sigma^2}{\nu_0^2 + c_\sigma^2} \ge \frac{\nu_i^2 + \sigma_t^2}{\nu_0^2 + \sigma_t^2}$. Notice that for function $f(z) = \log z - \frac{z}{2} + \frac{1}{2z}$, we have $\frac{\mathrm{d}}{\mathrm{d}z}f(z) = \frac{1}{z} - \frac{1}{2} - \frac{1}{2z^2} = -\frac{1}{2}\left(\frac{1}{z} - 1\right)^2 \le 0$. Thus, by the assumption

$$\|\boldsymbol{\mu}_i - \boldsymbol{\mu}_0\|^2 \le \frac{\nu_0^2 - \nu_i^2}{2}\left(\log\left(\frac{\nu_i^2 + c_\sigma^2}{\nu_0^2 + c_\sigma^2}\right) - \frac{\nu_i^2 + c_\sigma^2}{2\nu_0^2 + c_\sigma^2} + \frac{\nu_0^2 + c_\sigma^2}{2\nu_i^2 + c_\sigma^2}\right)d,$$

we have that for all $t \in [T]$,

$$\|\boldsymbol{\mu}_i - \boldsymbol{\mu}_0\|^2 \le \frac{\nu_0^2 - \nu_i^2}{2}\left(\log\left(\frac{\nu_i^2 + \sigma_t^2}{\nu_0^2 + \sigma_t^2}\right) - \frac{\nu_i^2 + \sigma_t^2}{2\nu_0^2 + \sigma_t^2} + \frac{\nu_0^2 + \sigma_t^2}{2\nu_i^2 + \sigma_t^2}\right)d.$$

Conditioned on $\|\mathbf{n}_{t-1}\|^2 > \frac{\nu_0^2 + \nu_{\max}^2 + 2\sigma_{t-1}^2}{2}d$ for all $t \in [T]$, by Lemma 6 we have that for $T = \exp(\mathcal{O}(d))$,

$$\|\hat{\mathbf{n}}_T - \mathbf{n}_T\| < \left(\sqrt{\frac{5\nu_0^2 + 3\nu_{\max}^2 + 8\sigma_T^2}{8}} - \sqrt{\frac{\nu_0^2 + \nu_{\max}^2 + 2\sigma_T^2}{2}}\right)\sqrt{d}.$$

By Lemma 8 we have that with probability at least $1 - \exp(-\Omega(d))$,

$$\|\hat{\mathbf{n}}_T\|^2 \ge \frac{5\nu_0^2 + 3\nu_{\max}^2 + 8\sigma_T^2}{8}d.$$

Combining the two inequalities implies the desired bound

$$\|\mathbf{n}_T\| \ge \|\hat{\mathbf{n}}_T\| - \|\hat{\mathbf{n}}_T - \mathbf{n}_T\| > \sqrt{\frac{\nu_0^2 + \nu_{\max}^2 + 2\sigma_T^2}{2}d}.$$

Hence by induction we obtain $\|\mathbf{n}_t\|^2 > \frac{\nu_0^2 + \nu_{\max}^2 + 2\sigma_t^2}{2}d$ for all $t \in \{0\} \cup [T]$ with probability at least

$$(1 - (T-1)\exp(-\Omega(d))) \cdot (1 - \exp(-\Omega(d))) \ge 1 - T\exp(-\Omega(d)).$$

Therefore we complete the proof of Proposition 4. $\qquad\square$

Finally, combining equation 11 and Proposition 4 finishes the proof of Theorem 2.

## A.3 Proof of Theorem 3: Langevin Dynamics under Sub-Gaussian Mixtures

The proof framework is similar to the proof of Theorem 1. To begin with, we validate Assumption 2.v. in the following lemma:

**Lemma 9.** *For constants $\nu_0, \nu_i, c_\nu, c_L$ satisfying Assumptions 2.iii. and 2.iv., we have $\frac{(1-c_\nu)\nu_0^2 - \nu_i^2}{2(1-c_\nu)} >$ 0 and $\log\frac{c_\nu \nu_i^2}{(c_L^2 + c_\nu c_L)\nu_0^2} - \frac{\nu_i^2}{2(1-c_\nu)\nu_0^2} + \frac{(1-c_\nu)\nu_0^2}{2\nu_i^2} > 0$ are both positive constants.*

*Proof of Lemma 9.* From Assumption 2.iv. that $\nu_0^2 > \frac{\nu_{\max}^2}{1-c_\nu} \ge \frac{\nu_i^2}{1-c_\nu}$, we easily obtain $\frac{(1-c_\nu)\nu_0^2 - \nu_i^2}{2(1-c_\nu)} >$ 0 is a positive constant. For the second property, let $f(z) := \log\frac{c_\nu \nu_i^2}{(c_L^2 + c_\nu c_L)z} - \frac{\nu_i^2}{2(1-c_\nu)z} + \frac{(1-c_\nu)z}{2\nu_i^2}$. For any $z > \frac{\nu_i^2}{1-c_\nu}$, the derivative of $f(z)$ satisfies

$$\frac{\mathrm{d}}{\mathrm{d}z}f(z) = -\frac{1}{z} + \frac{\nu_i^2}{2(1-c_\nu)z^2} + \frac{1-c_\nu}{2\nu_i^2} = \frac{\nu_i^2}{2(1-c_\nu)}\left(\frac{1-c_\nu}{\nu_i^2} - \frac{1}{z}\right)^2 > 0.$$

Therefore, when $\frac{4(c_L^2 + c_\nu c_L)}{c_\nu (1 - c_\nu)} \leq 1$, we have

$$f(\nu_0^2) > f\left(\frac{\nu_i^2}{1 - c_\nu}\right) = \log \frac{c_\nu(1 - c_\nu)}{c_L^2 + c_\nu c_L} \geq \log 4 > 0.$$

When $\frac{4(c_L^2 + c_\nu c_L)}{c_\nu (1 - c_\nu)} > 1$, we have

$$f(\nu_0^2) > f\left(\frac{4(c_L^2 + c_\nu c_L)}{c_\nu(1 - c_\nu)} \frac{\nu_i^2}{1 - c_\nu}\right) = 2\log \frac{c_\nu(1 - c_\nu)}{2(c_L^2 + c_\nu c_L)} - \frac{c_\nu(1 - c_\nu)}{8(c_L^2 + c_\nu c_L)} + \frac{2(c_L^2 + c_\nu c_L)}{c_\nu(1 - c_\nu)}$$

$$\geq 2 - 2\log 2 - \frac{2(c_L^2 + c_\nu c_L)}{c_\nu(1 - c_\nu)} - \frac{c_\nu(1 - c_\nu)}{8(c_L^2 + c_\nu c_L)} + \frac{2(c_L^2 + c_\nu c_L)}{c_\nu(1 - c_\nu)} > 2 - 2\log 2 - \frac{1}{2} > 0.$$

Thus we obtain Lemma 9. □

Without loss of generality, we assume $\boldsymbol{\mu}_0 = \mathbf{0}_d$. Similar to the proof of Theorem 1, we decompose

$$\mathbf{x}_t = \mathbf{R}\mathbf{r}_t + \mathbf{N}\mathbf{n}_t, \text{ and } \boldsymbol{\epsilon}_t = \mathbf{R}\boldsymbol{\epsilon}_t^{(\mathbf{r})} + \mathbf{N}\boldsymbol{\epsilon}_t^{(\mathbf{n})},$$

where $\mathbf{R} \in \mathbb{R}^{d \times r}$ an orthonormal basis of the vector space $\{\boldsymbol{\mu}_i\}_{i \in [k]}$ and $\mathbf{N} \in \mathbb{R}^{d \times n}$ an orthonormal basis of the null space of $\{\boldsymbol{\mu}_i\}_{i \in [k]}$. To show $\|\mathbf{x}_t - \boldsymbol{\mu}_i\|^2 > \left(\frac{\nu_0^2}{2} + \frac{\nu_{\max}^2}{2(1 - c_\nu)}\right)d$, it suffices to prove $\|\mathbf{n}_t\|^2 > \left(\frac{\nu_0^2}{2} + \frac{\nu_{\max}^2}{2(1 - c_\nu)}\right)d$. By Proposition 2, if $\mathbf{x}_0$ is initialized in the distribution $P^{(0)}$, i.e., $\mathbf{x}_0 \sim P^{(0)}$, since $\nu_0^2 > \frac{1}{1 - c_\nu}\nu_{\max}^2$, with probability at least $1 - \exp(-\Omega(d))$ we have

$$\|\mathbf{n}_0\|^2 \geq \left(\frac{3\nu_0^2}{4} + \frac{\nu_{\max}^2}{4(1 - c_\nu)}\right)d. \tag{12}$$

Then, conditioned on $\|\mathbf{n}_0\|^2 \geq \left(\frac{3\nu_0^2}{4} + \frac{\nu_{\max}^2}{4(1 - c_\nu)}\right)d$, the following proposition shows that $\|\mathbf{n}_t\|$ remains large with high probability.

**Proposition 5.** *Consider a distribution $P$ satisfying Assumption 2. We follow the Langevin dynamics for $T = \exp(\mathcal{O}(d))$ steps. Suppose that the initial sample satisfies $\|\mathbf{n}_0\|^2 \geq \left(\frac{3\nu_0^2}{4} + \frac{\nu_{\max}^2}{4(1 - c_\nu)}\right)d$, then with probability at least $1 - T \cdot \exp(-\Omega(d))$, we have that $\|\mathbf{n}_t\|^2 > \left(\frac{\nu_0^2}{2} + \frac{\nu_{\max}^2}{2(1 - c_\nu)}\right)d$ for all $t \in \{0\} \cup [T]$.*

*Proof of Proposition 5.* Firstly, by Lemma 3, if $\delta_t > \nu_0^2$, since $\nu_0^2 > \frac{\nu_{\max}^2}{1 - c_\nu}$, we similarly have that $\|\mathbf{n}_t\|^2 \geq \left(\frac{3\nu_0^2}{4} + \frac{\nu_{\max}^2}{4(1 - c_\nu)}\right)d$ with probability at least $1 - \exp(-\Omega(d))$ regardless of the previous state $\mathbf{x}_{t-1}$. We then consider the case when $\delta_t \leq \nu_0^2$. Intuitively, we aim to prove that the score function is close to $-\frac{\mathbf{x}}{\nu_0^2}$ when $\|\mathbf{n}\|^2 \geq \left(\frac{\nu_0^2}{2} + \frac{\nu_{\max}^2}{2(1 - c_\nu)}\right)d$. Towards this goal, we first show that $P^{(0)}(\mathbf{x})$ is exponentially larger than $P^{(i)}(\mathbf{x})$ for all $i \in [k]$ in the following lemma:

**Lemma 10.** *Suppose $P$ satisfies Assumption 2. Then for any $\|\mathbf{n}\|^2 \geq \left(\frac{\nu_0^2}{2} + \frac{\nu_{\max}^2}{2(1 - c_\nu)}\right)d$, we have $\frac{P^{(i)}(\mathbf{x})}{P^{(0)}(\mathbf{x})} \leq \exp(-\Omega(d))$ and $\frac{\|\nabla_\mathbf{x} P^{(i)}(\mathbf{x})\|}{P(\mathbf{x})} \leq \exp(-\Omega(d))$ for all $i \in [k]$.*

*Proof of Lemma 10.* We first give an upper bound on the sub-Gaussian probability density. For any vector $\mathbf{v} \in \mathbb{R}^d$, by considering some vector $\mathbf{m} \in \mathbb{R}^d$, from Markov's inequality and the definition in equation 4 we can bound

$$\mathbb{P}_{\mathbf{z} \sim P^{(i)}}\left(\mathbf{m}^T(\mathbf{z} - \boldsymbol{\mu}_i) \geq \mathbf{m}^T(\mathbf{v} - \boldsymbol{\mu}_i)\right) \leq \frac{\mathbb{E}_{\mathbf{z} \sim P^{(i)}}\left[\exp\left(\mathbf{m}^T(\mathbf{z} - \boldsymbol{\mu}_i)\right)\right]}{\exp\left(\mathbf{m}^T(\mathbf{v} - \boldsymbol{\mu}_i)\right)}$$

$$\leq \exp\left(\frac{\nu_i^2 \|\mathbf{m}\|^2}{2} - \mathbf{m}^T(\mathbf{v} - \boldsymbol{\mu}_i)\right).$$

Upon optimizing the last term at $\mathbf{m} = \frac{\mathbf{v} - \boldsymbol{\mu}_i}{\nu_i^2}$, we obtain

$$\mathbb{P}_{\mathbf{z} \sim P^{(i)}} \left( (\mathbf{v} - \boldsymbol{\mu}_i)^T (\mathbf{v} - \mathbf{z}) \leq 0 \right) \leq \exp \left( -\frac{\|\mathbf{v} - \boldsymbol{\mu}_i\|^2}{2\nu_i^2} \right). \tag{13}$$

Denote $\mathbb{B} := \left\{ \mathbf{z} : (\mathbf{v} - \boldsymbol{\mu}_i)^T (\mathbf{v} - \mathbf{z}) \leq 0 \right\}$. To bound $\mathbb{P}_{\mathbf{z} \sim P^{(i)}}(\mathbf{z} \in \mathbb{B})$, we first note that

$$
\begin{aligned}
&\log P^{(i)}(\mathbf{v}) - \log P^{(i)}(\mathbf{z}) \\
&= \int_0^1 \langle \mathbf{v} - \mathbf{z}, \nabla \log P^{(i)}(\mathbf{v} + \lambda(\mathbf{z} - \mathbf{v})) \rangle \, \mathrm{d}\lambda \\
&= \langle \mathbf{v} - \mathbf{z}, \nabla \log P^{(i)}(\mathbf{v}) \rangle + \int_0^1 \langle \mathbf{v} - \mathbf{z}, \nabla \log P^{(i)}(\mathbf{v} + \lambda(\mathbf{z} - \mathbf{v})) - \nabla \log P^{(i)}(\mathbf{v}) \rangle \, \mathrm{d}\lambda \\
&\leq \|\mathbf{v} - \mathbf{z}\| \left\| \nabla \log P^{(i)}(\mathbf{v}) \right\| + \int_0^1 \|\mathbf{v} - \mathbf{z}\| \left\| \nabla \log P^{(i)}(\mathbf{v} + \lambda(\mathbf{z} - \mathbf{v})) - \nabla \log P^{(i)}(\mathbf{v}) \right\| \, \mathrm{d}\lambda \\
&\leq \|\mathbf{v} - \mathbf{z}\| \cdot L_i \|\mathbf{v} - \boldsymbol{\mu}_i\| + \int_0^1 \|\mathbf{v} - \mathbf{z}\| \cdot L_i \|\lambda(\mathbf{z} - \mathbf{v})\| \, \mathrm{d}\lambda \tag{14} \\
&\leq \frac{L_i c_\nu}{2c_L} \|\mathbf{v} - \boldsymbol{\mu}_i\|^2 + \left( \frac{c_L + c_\nu}{2c_\nu} \right) L_i \|\mathbf{v} - \mathbf{z}\|^2,
\end{aligned}
$$

where equation 14 follows from Assumption 2.ii. that $\nabla \log P^{(i)}(\boldsymbol{\mu}_i) = \mathbf{0}_d$ and Assumption 2.iii. that the score function $\nabla \log P^{(i)}$ is $L_i$-Lipschitz. Therefore we obtain

$$
\begin{aligned}
\mathbb{P}_{\mathbf{z} \sim P^{(i)}}(\mathbf{z} \in \mathbb{B}) &= \int_{\mathbf{z} \in \mathbb{B}} P^{(i)}(\mathbf{z}) \, \mathrm{d}\mathbf{z} \\
&\geq \int_{\mathbf{z} \in \mathbb{B}} P^{(i)}(\mathbf{v}) \exp \left( -\frac{L_i c_\nu}{2c_L} \|\mathbf{v} - \boldsymbol{\mu}_i\|^2 - \frac{c_L + c_\nu}{2c_\nu} L_i \|\mathbf{v} - \mathbf{z}\|^2 \right) \, \mathrm{d}\mathbf{z} \\
&= P^{(i)}(\mathbf{v}) \exp \left( -\frac{L_i c_\nu}{2c_L} \|\mathbf{v} - \boldsymbol{\mu}_i\|^2 \right) \int_{\mathbf{z} \in \mathbb{B}} \exp \left( -\frac{c_L + c_\nu}{2c_\nu} L_i \|\mathbf{v} - \mathbf{z}\|^2 \right) \, \mathrm{d}\mathbf{z}. \tag{15}
\end{aligned}
$$

By observing that $g : \mathbb{B} \to \left\{ \mathbf{z} : (\mathbf{v} - \boldsymbol{\mu}_i)^T (\mathbf{v} - \mathbf{z}) \geq 0 \right\}$ with $g(\mathbf{z}) = 2\mathbf{v} - \mathbf{z}$ is a bijection such that $\|\mathbf{v} - \mathbf{z}\| = \|\mathbf{v} - g(\mathbf{z})\|$ for any $\mathbf{z} \in \mathbb{B}$, we have

$$
\begin{aligned}
\int_{\mathbf{z} \in \mathbb{B}} \exp \left( -\frac{c_L + c_\nu}{2c_\nu} L_i \|\mathbf{v} - \mathbf{z}\|^2 \right) \, \mathrm{d}\mathbf{z} &= \frac{1}{2} \int_{\mathbf{z} \in \mathbb{R}^d} \exp \left( -\frac{c_L + c_\nu}{2c_\nu} L_i \|\mathbf{v} - \mathbf{z}\|^2 \right) \, \mathrm{d}\mathbf{z} \\
&= \frac{1}{2} \left( \frac{2\pi c_\nu}{(c_L + c_\nu) L_i} \right)^{\frac{d}{2}}. \tag{16}
\end{aligned}
$$

Hence, by combining equation 13, equation 15, and equation 16, we obtain

$$
\begin{aligned}
\exp \left( -\frac{\|\mathbf{v} - \boldsymbol{\mu}_i\|^2}{2\nu_i^2} \right) &\geq \mathbb{P}_{\mathbf{z} \sim P^{(i)}} \left( (\mathbf{v} - \boldsymbol{\mu}_i)^T (\mathbf{v} - \mathbf{z}) \leq 0 \right) \\
&\geq P^{(i)}(\mathbf{v}) \exp \left( -\frac{L_i c_\nu}{2c_L} \|\mathbf{v} - \boldsymbol{\mu}_i\|^2 \right) \cdot \frac{1}{2} \left( \frac{2\pi c_\nu}{(c_L + c_\nu) L_i} \right)^{\frac{d}{2}}.
\end{aligned}
$$

By Assumption 2.iii. that $L_i \leq \frac{c_L}{\nu_i^2}$ we obtain the following bound on the probability density:

$$P^{(i)}(\mathbf{v}) \leq 2 \left( \frac{2\pi c_\nu \nu_i^2}{(c_L + c_\nu) c_L} \right)^{-\frac{d}{2}} \exp \left( -\frac{1 - c_\nu}{2\nu_i^2} \|\mathbf{v} - \boldsymbol{\mu}_i\|^2 \right). \tag{17}$$

Then we can bound the ratio of $P^{(i)}$ and $P^{(0)}$. For all $i \in [k]$, define $\rho_i(\mathbf{x}) := \frac{P^{(i)}(\mathbf{x})}{P^{(0)}(\mathbf{x})}$, then we have

$$
\rho_i(\mathbf{x}) = \frac{P^{(i)}(\mathbf{x})}{P^{(0)}(\mathbf{x})} \leq \frac{2(2\pi c_\nu \nu_i^2/(c_L^2 + c_\nu c_L))^{-d/2} \exp\left(-(1-c_\nu)\|\mathbf{x} - \boldsymbol{\mu}_i\|^2 / 2\nu_i^2\right)}{(2\pi\nu_0^2)^{-d/2} \exp\left(-\|\mathbf{x}\|^2 / 2\nu_0^2\right)}
$$

$$
= 2\left(\frac{(c_L^2 + c_\nu c_L)\nu_0^2}{c_\nu \nu_i^2}\right)^{\frac{d}{2}} \exp\left(\frac{\|\mathbf{x}\|^2}{2\nu_0^2} - \frac{(1-c_\nu)\|\mathbf{x} - \boldsymbol{\mu}_i\|^2}{2\nu_i^2}\right)
$$

$$
= 2\left(\frac{(c_L^2 + c_\nu c_L)\nu_0^2}{c_\nu \nu_i^2}\right)^{\frac{d}{2}} \exp\left(\left(\frac{1}{2\nu_0^2} - \frac{1-c_\nu}{2\nu_i^2}\right)\|\mathbf{N}\mathbf{n}\|^2 + \left(\frac{\|\mathbf{R}\mathbf{r}\|^2}{2\nu_0^2} - \frac{(1-c_\nu)\|\mathbf{R}\mathbf{r} - \boldsymbol{\mu}_i\|^2}{2\nu_i^2}\right)\right)
$$

$$
= 2\left(\frac{(c_L^2 + c_\nu c_L)\nu_0^2}{c_\nu \nu_i^2}\right)^{\frac{d}{2}} \exp\left(\left(\frac{1}{2\nu_0^2} - \frac{1-c_\nu}{2\nu_i^2}\right)\|\mathbf{n}\|^2 + \left(\frac{\|\mathbf{r}\|^2}{2\nu_0^2} - \frac{(1-c_\nu)\|\mathbf{r} - \mathbf{R}^T\boldsymbol{\mu}_i\|^2}{2\nu_i^2}\right)\right),
$$

where the last step follows from the definition that $\mathbf{R} \in \mathbb{R}^{d \times r}$ an orthogonal basis of the vector space $\{\boldsymbol{\mu}_i\}_{i \in [k]}$ and $\mathbf{N}^T\mathbf{N} = \mathbf{I}_n$. Since $\nu_i^2 < (1-c_\nu)\nu_0^2$, the quadratic term $\frac{\|\mathbf{r}\|^2}{2\nu_0^2} - \frac{(1-c_\nu)\|\mathbf{r} - \mathbf{R}^T\boldsymbol{\mu}_i\|^2}{2\nu_i^2}$ is maximized at $\mathbf{r} = \frac{(1-c_\nu)\nu_0^2 \mathbf{R}^T\boldsymbol{\mu}_i}{(1-c_\nu)\nu_0^2 - \nu_i^2}$. Therefore, we obtain

$$
\frac{\|\mathbf{r}\|^2}{2\nu_0^2} - \frac{(1-c_\nu)\|\mathbf{r} - \mathbf{R}^T\boldsymbol{\mu}_i\|^2}{2\nu_i^2} \leq \frac{(1-c_\nu)\|\boldsymbol{\mu}_i\|^2}{2((1-c_\nu)\nu_0^2 - \nu_i^2)}.
$$

Hence, for $\|\boldsymbol{\mu}_i - \boldsymbol{\mu}_0\|^2 \leq \frac{(1-c_\nu)\nu_0^2 - \nu_i^2}{2(1-c_\nu)}\left(\log\frac{c_\nu \nu_i^2}{(c_L^2 + c_\nu c_L)\nu_0^2} - \frac{\nu_i^2}{2(1-c_\nu)\nu_0^2} + \frac{(1-c_\nu)\nu_0^2}{2\nu_i^2}\right)d$ and $\|\mathbf{n}\|^2 \geq \left(\frac{\nu_0^2}{2} + \frac{\nu_{\max}^2}{2(1-c_\nu)}\right)d$, we have

$$
\rho_i(\mathbf{x}) \leq 2\left(\frac{(c_L^2 + c_\nu c_L)\nu_0^2}{c_\nu \nu_i^2}\right)^{\frac{d}{2}} \exp\left(\left(\frac{1}{2\nu_0^2} - \frac{1-c_\nu}{2\nu_i^2}\right)\|\mathbf{n}\|^2 + \frac{(1-c_\nu)\|\boldsymbol{\mu}_i\|^2}{2((1-c_\nu)\nu_0^2 - \nu_i^2)}\right)
$$

$$
\leq 2\left(\frac{(c_L^2 + c_\nu c_L)\nu_0^2}{c_\nu \nu_i^2}\right)^{\frac{d}{2}} \exp\left(\left(\frac{1}{2\nu_0^2} - \frac{1-c_\nu}{2\nu_i^2}\right)\left(\frac{\nu_0^2}{2} + \frac{\nu_i^2}{2(1-c_\nu)}\right)d + \frac{(1-c_\nu)\|\boldsymbol{\mu}_i\|^2}{2((1-c_\nu)\nu_0^2 - \nu_i^2)}\right)
$$

$$
= 2\exp\left(-\left(\log\frac{c_\nu \nu_i^2}{(c_L^2 + c_\nu c_L)\nu_0^2} - \frac{\nu_i^2}{2(1-c_\nu)\nu_0^2} + \frac{(1-c_\nu)\nu_0^2}{2\nu_i^2}\right)\frac{d}{2} + \frac{(1-c_\nu)\|\boldsymbol{\mu}_i\|^2}{2((1-c_\nu)\nu_0^2 - \nu_i^2)}\right)
$$

$$
\leq 2\exp\left(-\left(\log\frac{c_\nu \nu_i^2}{(c_L^2 + c_\nu c_L)\nu_0^2} - \frac{\nu_i^2}{2(1-c_\nu)\nu_0^2} + \frac{(1-c_\nu)\nu_0^2}{2\nu_i^2}\right)\frac{d}{4}\right).
$$

From Lemma 9, we obtain $\rho_i(\mathbf{x}) \leq \exp(-\Omega(d))$.

To show $\frac{\|\nabla_{\mathbf{x}} P^{(i)}(\mathbf{x})\|}{P(\mathbf{x})} \leq \exp(-\Omega(d))$, from Assumptions 2.ii. and 2.iii. we have

$$
\left\|\frac{\nabla_{\mathbf{x}} P^{(i)}(\mathbf{x})}{P^{(i)}(\mathbf{x})}\right\| = \left\|\frac{\nabla_{\mathbf{x}} P^{(i)}(\mathbf{x})}{P^{(i)}(\mathbf{x})} - \frac{\nabla_{\mathbf{x}} P^{(i)}(\boldsymbol{\mu}_i)}{P^{(i)}(\boldsymbol{\mu}_i)}\right\| = \left\|\nabla_{\mathbf{x}} \log P^{(i)}(\mathbf{x}) - \nabla_{\mathbf{x}} \log P^{(i)}(\boldsymbol{\mu}_i)\right\|
$$

$$
\leq L_i \|\mathbf{x} - \boldsymbol{\mu}_i\| \leq \frac{c_L}{\nu_i^2}\|\mathbf{x} - \boldsymbol{\mu}_i\|.
$$

Therefore, we can bound $\frac{\|\nabla_{\mathbf{x}} P^{(i)}(\mathbf{x})\|}{P(\mathbf{x})} \leq \frac{c_L}{\nu_i^2}\rho_i(\mathbf{x})\|\mathbf{x} - \boldsymbol{\mu}_i\|$. When $\|\mathbf{x} - \boldsymbol{\mu}_i\| = \exp(o(d))$ is small, by $\rho_i(\mathbf{x}) \leq \exp(-\Omega(d))$ we directly have $\frac{\|\nabla_{\mathbf{x}} P^{(i)}(\mathbf{x})\|}{P(\mathbf{x})} \leq \exp(-\Omega(d))$. When $\|\mathbf{x} - \boldsymbol{\mu}_i\| = \exp(\Omega(d))$ is exceedingly large, from equation 17 we have

$$
\frac{\|\nabla_{\mathbf{x}} P^{(i)}(\mathbf{x})\|}{P(\mathbf{x})} \leq \frac{2c_L}{\nu_i^2}\left(\frac{(c_L^2 + c_\nu c_L)\nu_0^2}{c_\nu \nu_i^2}\right)^{\frac{d}{2}} \exp\left(\frac{\|\mathbf{x}\|^2}{2\nu_0^2} - \frac{(1-c_\nu)\|\mathbf{x} - \boldsymbol{\mu}_i\|^2}{2\nu_i^2}\right)\|\mathbf{x} - \boldsymbol{\mu}_i\|.
$$

Since $\nu_0^2 > \frac{\nu_i^2}{1-c_\nu}$, when $\|\mathbf{x} - \boldsymbol{\mu}_i\| = \exp(\Omega(d)) \gg \|\boldsymbol{\mu}_i\|$ we have

$$\exp\left(\frac{\|\mathbf{x}\|^2}{2\nu_0^2} - \frac{(1-c_\nu)\|\mathbf{x}-\boldsymbol{\mu}_i\|^2}{2\nu_i^2}\right) = \exp(-\Omega(\|\mathbf{x}-\boldsymbol{\mu}_i\|^2)).$$

Therefore $\frac{\|\nabla_\mathbf{x} P^{(i)}(\mathbf{x})\|}{P(\mathbf{x})} \leq \exp(-\Omega(d))$. Thus we complete the proof of Lemma 10. $\qquad\square$

Similar to Lemma 5, the following lemma proves that when the previous state $\mathbf{n}_{t-1}$ is far from a mode, a single step of Langevin dynamics with bounded step size is not enough to find the mode.

**Lemma 11.** *Suppose $\delta_t \leq \nu_0^2$ and $\|\mathbf{n}_{t-1}\|^2 > 36\nu_0^2 d$, then we have $\|\mathbf{n}_t\|^2 \geq \nu_0^2 d$ with probability at least $1 - \exp(-\Omega(d))$.*

*Proof of Lemma 11.* For simplicity, denote $\mathbf{v} := \mathbf{n}_{t-1} + \frac{\delta_t}{2}\mathbf{N}^T \nabla_\mathbf{x} \log P(\mathbf{x}_{t-1})$. Since $P = \sum_{i=0}^{k} w_i P^{(i)}$ and $P^{(0)} = \mathcal{N}(\boldsymbol{\mu}_0, \nu_0^2 \boldsymbol{I}_d)$, the score function can be written as

$$
\begin{aligned}
\nabla_\mathbf{x} \log P(\mathbf{x}) = \frac{\nabla_\mathbf{x} P(\mathbf{x})}{P(\mathbf{x})} &= \frac{\nabla_\mathbf{x} w_0 P^{(0)}(\mathbf{x})}{P(\mathbf{x})} + \sum_{i\in[k]} \frac{\nabla_\mathbf{x} w_i P^{(i)}(\mathbf{x})}{P(\mathbf{x})} \\
&= -\frac{w_0 P^{(0)}(\mathbf{x})}{P(\mathbf{x})} \cdot \frac{\mathbf{x}}{\nu_0^2} + \sum_{i\in[k]} \frac{w_i \nabla_\mathbf{x} P^{(i)}(\mathbf{x})}{P(\mathbf{x})} \\
&= -\frac{\mathbf{x}}{\nu_0^2} + \sum_{i\in[k]} \frac{w_i P^{(i)}(\mathbf{x})}{P(\mathbf{x})} \cdot \frac{\mathbf{x}}{\nu_0^2} + \sum_{i\in[k]} \frac{w_i \nabla_\mathbf{x} P^{(i)}(\mathbf{x})}{P(\mathbf{x})}.
\end{aligned}
\tag{18}
$$

For $\|\mathbf{n}_{t-1}\|^2 > 36\nu_0^2 d$ by Lemma 10 we have $\frac{\|\nabla_\mathbf{x} P^{(i)}(\mathbf{x}_{t-1})\|}{P(\mathbf{x}_{t-1})} \leq \exp(-\Omega(d))$. Since $\delta_t \leq \nu_0^2$, we can bound the norm of $\mathbf{v}$ by

$$
\begin{aligned}
\|\mathbf{v}\| &= \left\| \mathbf{n}_{t-1} + \frac{\delta_t}{2}\mathbf{N}^T \nabla_\mathbf{x} \log P(\mathbf{x}_{t-1}) \right\| \\
&= \left\| \mathbf{n}_{t-1} - \frac{\delta_t}{2\nu_0^2}\mathbf{n}_{t-1} + \sum_{i\in[k]} \frac{w_i \delta_t}{2\nu_0^2} \frac{P^{(i)}(\mathbf{x}_{t-1})}{P(\mathbf{x}_{t-1})}\mathbf{n}_{t-1} + \sum_{i\in[k]} \frac{w_i \delta_t}{2}\frac{\mathbf{N}^T \nabla_\mathbf{x} P^{(i)}(\mathbf{x}_{t-1})}{P(\mathbf{x}_{t-1})} \right\| \\
&\geq \left\| \left(1 - \frac{\delta_t}{2\nu_0^2} + \sum_{i\in[k]} \frac{w_i \delta_t}{2\nu_0^2}\frac{P^{(i)}(\mathbf{x}_{t-1})}{P(\mathbf{x}_{t-1})}\right)\mathbf{n}_{t-1} \right\| - \sum_{i\in[k]} \frac{w_i \delta_t}{2}\frac{\|\nabla_\mathbf{x} P^{(i)}(\mathbf{x}_{t-1})\|}{P(\mathbf{x}_{t-1})} \\
&\geq \frac{1}{2}\|\mathbf{n}_{t-1}\| - \sum_{i\in[k]} \frac{w_i \delta_t}{2}\exp(-\Omega(d)) \\
&> 2\nu_0 \sqrt{d}.
\end{aligned}
$$

On the other hand, from $\boldsymbol{\epsilon}_t^{(\mathbf{n})} \sim \mathcal{N}(\mathbf{0}_n, \boldsymbol{I}_n)$ we know $\frac{\langle \mathbf{v}, \boldsymbol{\epsilon}_t^{(\mathbf{n})}\rangle}{\|\mathbf{v}\|} \sim \mathcal{N}(0,1)$ for any fixed $\mathbf{v} \neq \mathbf{0}_n$, hence by Lemma 2 we have

$$\mathbb{P}\left(\frac{\langle \mathbf{v}, \boldsymbol{\epsilon}_t^{(\mathbf{n})}\rangle}{\|\mathbf{v}\|} \geq \frac{\sqrt{d}}{4}\right) = \mathbb{P}\left(\frac{\langle \mathbf{v}, \boldsymbol{\epsilon}_t^{(\mathbf{n})}\rangle}{\|\mathbf{v}\|} \leq -\frac{\sqrt{d}}{4}\right) \leq \frac{4}{\sqrt{2\pi d}}\exp\left(-\frac{d}{32}\right)$$

Combining the above inequalities gives

$$\|\mathbf{n}_t\|^2 = \left\| \mathbf{v} + \sqrt{\delta_t}\boldsymbol{\epsilon}_t^{(\mathbf{n})} \right\|^2 \geq \|\mathbf{v}\|^2 - 2\nu_0 |\langle \mathbf{v}, \boldsymbol{\epsilon}_t^{(\mathbf{n})}\rangle| \geq \|\mathbf{v}\|^2 - \frac{\nu_0 \sqrt{d}}{2}\|\mathbf{v}\| > \nu_0^2 d$$

with probability at least $1 - \frac{8}{\sqrt{2\pi d}}\exp\left(-\frac{d}{32}\right) = 1 - \exp(-\Omega(d))$. This proves Lemma 11. $\qquad\square$

When $\|\mathbf{n}_{t-1}\|^2 \leq 36\nu_0^2 d$, similar to Theorem 1, we consider a surrogate recursion $\hat{\mathbf{n}}_t$ such that $\hat{\mathbf{n}}_0 = \mathbf{n}_0$ and for all $t \geq 1$,

$$\hat{\mathbf{n}}_t = \hat{\mathbf{n}}_{t-1} - \frac{\delta_t}{2\nu_0^2}\hat{\mathbf{n}}_{t-1} + \sqrt{\delta_t}\boldsymbol{\epsilon}_t^{(\mathbf{n})}. \tag{19}$$

The following Lemma shows that $\hat{\mathbf{n}}_t$ is sufficiently close to the original recursion $\mathbf{n}_t$.

**Lemma 12.** *For any $t \geq 1$, given that for all $j \in [t]$, $\delta_j \leq \nu_0^2$ and $\left(\frac{\nu_0^2}{2} + \frac{\nu_{\max}^2}{2(1-c_\nu)}\right) d \leq \|\mathbf{n}_{j-1}\|^2 \leq 36\nu_0^2 d$, if $\boldsymbol{\mu}_i$ satisfies Assumption 2.v. for all $i \in [k]$, we have $\|\hat{\mathbf{n}}_t - \mathbf{n}_t\| \leq \frac{t}{\exp(\Omega(d))}\sqrt{d}$.*

*Proof of Lemma 12.* By equation 18 we have that for all $j \in [t]$,

$$\|\hat{\mathbf{n}}_j - \mathbf{n}_j\| = \left\|\hat{\mathbf{n}}_{j-1} - \mathbf{n}_{j-1} - \frac{\delta_j}{2\nu_0^2}\hat{\mathbf{n}}_{j-1} - \frac{\delta_j}{2}\mathbf{N}^T\nabla_{\mathbf{x}}\log P(\mathbf{x}_{j-1})\right\|$$

$$= \left\|\hat{\mathbf{n}}_{j-1} - \mathbf{n}_{j-1} - \sum_{i\in[k]}\frac{w_i P^{(i)}(\mathbf{x}_{j-1})}{\nu_0^2 P(\mathbf{x}_{j-1})}\mathbf{n}_{j-1} - \sum_{i\in[k]}\frac{w_i \mathbf{N}^T\nabla_{\mathbf{x}}P^{(i)}(\mathbf{x}_{j-1})}{P(\mathbf{x}_{j-1})}\right\|$$

$$\leq \|\hat{\mathbf{n}}_{j-1} - \mathbf{n}_{j-1}\| + \sum_{i\in[k]}\frac{w_i P^{(i)}(\mathbf{x}_{j-1})}{\nu_0^2 P(\mathbf{x}_{j-1})}\|\mathbf{n}_{j-1}\| + \sum_{i\in[k]}\frac{w_i \left\|\nabla_{\mathbf{x}}P^{(i)}(\mathbf{x}_{j-1})\right\|}{P(\mathbf{x}_{j-1})}.$$

By Lemma 10, we have $\frac{P^{(i)}(\mathbf{x}_{j-1})}{P^{(0)}(\mathbf{x}_{j-1})} \leq \exp(-\Omega(d))$ and $\frac{\|\nabla_{\mathbf{x}}P^{(i)}(\mathbf{x}_{j-1})\|}{P(\mathbf{x}_{j-1})} \leq \exp(-\Omega(d))$ for all $i \in [k]$, hence from $\|\mathbf{n}_{j-1}\| \leq 6\nu_0\sqrt{d}$ we obtain a recursive bound

$$\|\hat{\mathbf{n}}_j - \mathbf{n}_j\| \leq \|\hat{\mathbf{n}}_{j-1} - \mathbf{n}_{j-1}\| + \frac{1}{\exp(\Omega(d))}\sqrt{d}.$$

Finally, by $\hat{\mathbf{n}}_0 = \mathbf{n}_0$, we have

$$\|\hat{\mathbf{n}}_t - \mathbf{n}_t\| = \sum_{j\in[t]}\left(\|\hat{\mathbf{n}}_j - \mathbf{n}_j\| - \|\hat{\mathbf{n}}_{j-1} - \mathbf{n}_{j-1}\|\right) \leq \frac{t}{\exp(\Omega(d))}\sqrt{d}.$$

Hence we obtain Lemma 12. $\qquad\square$

Armed with the above lemmas, we are now ready to establish Proposition 5 by induction. Please note that we also apply some lemmas from the proof of Theorem 1 by substituting $\nu_{\max}^2$ with $\frac{\nu_{\max}^2}{1-c_\nu}$. Suppose the theorem holds for all $T$ values of $1, \cdots, T-1$. We consider the following 3 cases:

- If there exists some $t \in [T]$ such that $\delta_t > \nu_0^2$, by Lemma 3 we know that with probability at least $1 - \exp(-\Omega(d))$, we have $\|\mathbf{n}_t\|^2 \geq \left(\frac{3\nu_0^2}{4} + \frac{\nu_{\max}^2}{4(1-c_\nu)}\right) d$, thus the problem reduces to the two sub-arrays $\mathbf{n}_0, \cdots, \mathbf{n}_{t-1}$ and $\mathbf{n}_t, \cdots, \mathbf{n}_T$, which can be solved by induction.

- Suppose $\delta_t \leq \nu_0^2$ for all $t \in [T]$. If there exists some $t \in [T]$ such that $\|\mathbf{n}_{t-1}\|^2 > 36\nu_0^2 d$, by Lemma 11 we know that with probability at least $1 - \exp(-\Omega(d))$, we have $\|\mathbf{n}_t\|^2 \geq \nu_0^2 d > \left(\frac{3\nu_0^2}{4} + \frac{\nu_{\max}^2}{4(1-c_\nu)}\right) d$, thus the problem similarly reduces to the two sub-arrays $\mathbf{n}_0, \cdots, \mathbf{n}_{t-1}$ and $\mathbf{n}_t, \cdots, \mathbf{n}_T$, which can be solved by induction.

- Suppose $\delta_t \leq \nu_0^2$ and $\|\mathbf{n}_{t-1}\|^2 \leq 36\nu_0^2 d$ for all $t \in [T]$. Conditioned on $\|\mathbf{n}_{t-1}\|^2 > \left(\frac{\nu_0^2}{2} + \frac{\nu_{\max}^2}{2(1-c_\nu)}\right) d$ for all $t \in [T]$, by Lemma 12 we have that for $T = \exp(\mathcal{O}(d))$,

$$\|\hat{\mathbf{n}}_T - \mathbf{n}_T\| < \left(\sqrt{\frac{5\nu_0^2}{8} + \frac{3\nu_{\max}^2}{8(1-c_\nu)}} - \sqrt{\frac{\nu_0^2}{2} + \frac{\nu_{\max}^2}{2(1-c_\nu)}}\right)\sqrt{d}.$$

By Lemma 8 we have that with probability at least $1 - \exp(-\Omega(d))$,

$$\|\hat{\mathbf{n}}_T\|^2 \geq \left(\frac{5\nu_0^2}{8} + \frac{3\nu_{\max}^2}{8(1-c_\nu)}\right) d.$$

Combining the two inequalities implies the desired bound

$$\|\mathbf{n}_T\| \geq \|\hat{\mathbf{n}}_T\| - \|\hat{\mathbf{n}}_T - \mathbf{n}_T\| > \sqrt{\left(\frac{\nu_0^2}{2} + \frac{\nu_{\max}^2}{2(1-c_\nu)}\right) d}.$$

Hence by induction we obtain $\|\mathbf{n}_t\|^2 > \left(\frac{\nu_0^2}{2} + \frac{\nu_{\max}^2}{2(1-c_\nu)}\right) d$ for all $t \in [T]$ with probability at least

$$(1 - (T-1)\exp(-\Omega(d))) \cdot (1 - \exp(-\Omega(d))) \geq 1 - T\exp(-\Omega(d)).$$

Therefore we complete the proof of Proposition 5. □

Finally, combining equation 12 and Proposition 5 finishes the proof of Theorem 3.

### A.4 Proof of Theorem 4: Annealed Langevin Dynamics under Sub-Gaussian Mixtures

**Assumption 3.** *Consider a data distribution $P := \sum_{i=0}^{k} w_i P^{(i)}$ as a mixture of sub-Gaussian distributions, where $1 \leq k = o(d)$ and $w_i > 0$ is a positive constant such that $\sum_{i=0}^{k} w_i = 1$. Suppose that $P^{(0)} = \mathcal{N}(\boldsymbol{\mu}_0, \nu_0^2 \mathbf{I}_d)$ is Gaussian and for all $i \in [k]$, $P^{(i)}$ satisfies*

*i. $P^{(i)}$ is a sub-Gaussian distribution of mean $\boldsymbol{\mu}_i$ with parameter $\nu_i^2$,*

*ii. $P^{(i)}$ is differentiable and $\nabla P_{\sigma_t}^{(i)}(\boldsymbol{\mu}_i) = \mathbf{0}_d$ for all $t \in \{0\} \cup [T]$,*

*iii. for all $t \in \{0\} \cup [T]$, the score function of $P_{\sigma_t}^{(i)}$ is $L_{i,t}$-Lipschitz such that $L_{i,t} \leq \frac{c_L}{\nu_i^2 + \sigma_t^2}$ for some constant $c_L > 0$,*

*iv. $\nu_0^2 > \max\left\{1, \frac{4(c_L^2 + c_\nu c_L)}{c_\nu(1-c_\nu)}\right\} \frac{\nu_{\max}^2 + c_\sigma^2}{1-c_\nu} - c_\sigma^2$ for constant $c_\nu \in (0,1)$, where $\nu_{\max} := \max_{i\in[k]} \nu_i$,*

*v. $\|\boldsymbol{\mu}_i - \boldsymbol{\mu}_0\|^2 \leq \frac{(1-c_\nu)\nu_0^2 - \nu_i^2 - c_\nu c_\sigma^2}{2(1-c_\nu)} \left(\log \frac{c_\nu(\nu_i^2 + c_\sigma^2)}{(c_L^2 + c_\nu c_L)(\nu_0^2 + c_\sigma^2)} - \frac{(\nu_i^2 + c_\sigma^2)}{2(1-c_\nu)(\nu_0^2 + c_\sigma^2)} + \frac{(1-c_\nu)(\nu_0^2 + c_\sigma^2)}{2(\nu_i^2 + c_\sigma^2)}\right) d.$*

The feasibility of Assumption 3.v. can be validated by substituting $\nu^2$ in Lemma 9 with $\nu^2 + c_\sigma^2$. To establish Theorem 4, we first note from Proposition 1 that for a sub-Gaussian mixture $P = \sum_{i=0}^{k} w_i P^{(i)}$, the perturbed distribution of noise level $\sigma$ is $P_\sigma = \sum_{i=0}^{k} w_i P_\sigma^{(i)}$, where $P^{(0)} = \mathcal{N}(\boldsymbol{\mu}_0, (\nu_i^2 + \sigma^2)\mathbf{I}_d)$ and $P^{(i)}$ is a sub-Gaussian distribution with mean $\boldsymbol{\mu}_i$ and sub-Gaussian parameter $(\nu_i^2 + \sigma^2)$. Similar to the proof of Theorem 1, we decompose

$$\mathbf{x}_t = \mathbf{R}\mathbf{r}_t + \mathbf{N}\mathbf{n}_t, \text{ and } \boldsymbol{\epsilon}_t = \mathbf{R}\boldsymbol{\epsilon}_t^{(\mathbf{r})} + \mathbf{N}\boldsymbol{\epsilon}_t^{(\mathbf{n})},$$

where $\mathbf{R} \in \mathbb{R}^{d\times r}$ an orthonormal basis of the vector space $\{\boldsymbol{\mu}_i\}_{i\in[k]}$ and $\mathbf{N} \in \mathbb{R}^{d\times n}$ an orthonormal basis of the null space of $\{\boldsymbol{\mu}_i\}_{i\in[k]}$. Now, we prove Theorem 4 by applying the techniques developed in Appendix A.1 and A.3 via substituting $\nu^2$ and $\frac{\nu^2}{1-c_\nu}$ with $\frac{\nu^2 + \sigma_t^2}{1-c_\nu}$ at time step $t$. Note that for all $t \in \{0\} \cup [T]$, Assumption 3.iv. implies $\nu_0^2 + \sigma_t^2 > \max\left\{1, \frac{4(c_L^2 + c_\nu c_L)}{c_\nu(1-c_\nu)}\right\} \frac{\nu_{\max}^2 + \sigma_t^2}{1-c_\nu}$ because $c_\sigma \geq \sigma_t$.

First, by Proposition 2, suppose that the sample is initialized in the distribution $P_{\sigma_0}^{(0)}$, then with probability at least $1 - \exp(-\Omega(d))$, we have

$$\|\mathbf{n}_0\|^2 \geq \left(\frac{3(\nu_0^2 + \sigma_0^2)}{4} + \frac{\nu_{\max}^2 + \sigma_0^2}{4(1-c_\nu)}\right) d. \tag{20}$$

Then, with the assumption that the initialization satisfies $\|\mathbf{n}_0\|^2 \geq \left(\frac{3(\nu_0^2 + \sigma_0^2)}{4} + \frac{\nu_{\max}^2 + \sigma_0^2}{4(1-c_\nu)}\right) d$, the following proposition similar to Proposition 5 shows that $\|\mathbf{n}_t\|$ remains large with high probability.

**Proposition 6.** *Consider a distribution $P$ satisfying Assumption 3. We follow annealed Langevin dynamics for $T = \exp(\mathcal{O}(d))$ steps with noise level $c_\sigma \geq \sigma_0 \geq \sigma_1 \geq \cdots \geq \sigma_T \geq 0$ for some constant $c_\sigma > 0$. Suppose that the initial sample satisfies $\|\mathbf{n}_0\|^2 \geq \left(\frac{3(\nu_0^2 + \sigma_0^2)}{4} + \frac{\nu_{\max}^2 + \sigma_0^2}{4(1-c_\nu)}\right) d$, then with probability at least $1 - T \cdot \exp(-\Omega(d))$, we have that $\|\mathbf{n}_t\|^2 > \left(\frac{\nu_0^2 + \sigma_t^2}{2} + \frac{\nu_{\max}^2 + \sigma_t^2}{2(1-c_\nu)}\right) d$ for all $t \in \{0\} \cup [T]$.*

*Proof of Proposition 6.* We prove Proposition 6 by induction. Suppose the theorem holds for all $T$ values of $1, \cdots, T-1$. We consider the following 3 cases:

- If there exists some $t \in [T]$ such that $\delta_t > \nu_0^2 + \sigma_t^2$, by Lemma 3 we know that with probability at least $1 - \exp(-\Omega(d))$, we have $\|\mathbf{n}_t\|^2 \geq \left(\frac{3(\nu_0^2 + \sigma_t^2)}{4} + \frac{\nu_{\max}^2 + \sigma_t^2}{4(1-c_\nu)}\right) d$, thus the problem reduces to the two sub-arrays $\mathbf{n}_0, \cdots, \mathbf{n}_{t-1}$ and $\mathbf{n}_t, \cdots, \mathbf{n}_T$, which can be solved by induction.

- Suppose $\delta_t \leq \nu_0^2 + \sigma_t^2$ for all $t \in [T]$. If there exists some $t \in [T]$ such that $\|\mathbf{n}_{t-1}\|^2 > 36(\nu_0^2 + \sigma_{t-1}^2)d \geq 36(\nu_0^2 + \sigma_t^2)d$, by Lemma 11 we know that with probability at least $1 - \exp(-\Omega(d))$, we have $\|\mathbf{n}_t\|^2 \geq (\nu_0^2 + \sigma_t^2)d > \left(\frac{3(\nu_0^2 + \sigma_t^2)}{4} + \frac{\nu_{\max}^2 + \sigma_t^2}{4(1-c_\nu)}\right) d$, thus the problem similarly reduces to the two sub-arrays $\mathbf{n}_0, \cdots, \mathbf{n}_{t-1}$ and $\mathbf{n}_t, \cdots, \mathbf{n}_T$, which can be solved by induction.

- Suppose $\delta_t \leq \nu_0^2 + \sigma_t^2$ and $\|\mathbf{n}_{t-1}\|^2 \leq 36(\nu_0^2 + \sigma_{t-1}^2)d$ for all $t \in [T]$. Consider a surrogate sequence $\hat{\mathbf{n}}_t$ such that $\hat{\mathbf{n}}_0 = \mathbf{n}_0$ and for all $t \geq 1$,

$$\hat{\mathbf{n}}_t = \hat{\mathbf{n}}_{t-1} - \frac{\delta_t}{2\nu_0^2 + 2\sigma_t^2}\hat{\mathbf{n}}_{t-1} + \sqrt{\delta_t}\boldsymbol{\epsilon}_t^{(\mathbf{n})}.$$

Since $\nu_0 > \nu_i$ and $c_\sigma \geq \sigma_t$ for all $t \in \{0\} \cup [T]$, we have $\frac{\nu_i^2 + c_\sigma^2}{\nu_0^2 + c_\sigma^2} > \frac{\nu_i^2 + \sigma_t^2}{\nu_0^2 + \sigma_t^2}$. Notice that for function $f(z) = \log z - \frac{z}{2} + \frac{1}{2z}$, we have $\frac{\mathrm{d}}{\mathrm{d}z} f(z) = \frac{1}{z} - \frac{1}{2} - \frac{1}{2z^2} = -\frac{1}{2}\left(\frac{1}{z} - 1\right)^2 \leq 0$.

Thus, by Assumption 3.v. we have that for all $t \in [T]$,

$$\|\boldsymbol{\mu}_i - \boldsymbol{\mu}_0\|^2 \leq \frac{(1-c_\nu)\nu_0^2 - \nu_i^2 - c_\nu c_\sigma^2}{2(1-c_\nu)}\left(\log \frac{c_\nu(\nu_i^2 + c_\sigma^2)}{(c_L^2 + c_\nu c_L)(\nu_0^2 + c_\sigma^2)}\right.$$
$$\left. - \frac{(\nu_i^2 + c_\sigma^2)}{2(1-c_\nu)(\nu_0^2 + c_\sigma^2)} + \frac{(1-c_\nu)(\nu_0^2 + c_\sigma^2)}{2(\nu_i^2 + c_\sigma^2)}\right) d$$
$$\leq \frac{(1-c_\nu)\nu_0^2 - \nu_i^2 - c_\nu \sigma_t^2}{2(1-c_\nu)}\left(\log \frac{c_\nu(\nu_i^2 + \sigma_t^2)}{(c_L^2 + c_\nu c_L)(\nu_0^2 + \sigma_t^2)}\right.$$
$$\left. - \frac{(\nu_i^2 + \sigma_t^2)}{2(1-c_\nu)(\nu_0^2 + \sigma_t^2)} + \frac{(1-c_\nu)(\nu_0^2 + \sigma_t^2)}{2(\nu_i^2 + \sigma_t^2)}\right) d$$

Conditioned on $\|\mathbf{n}_{t-1}\|^2 > \left(\frac{\nu_0^2 + \sigma_{t-1}^2}{2} + \frac{\nu_{\max}^2 + \sigma_{t-1}^2}{2(1-c_\nu)}\right) d$ for all $t \in [T]$, by Lemma 12 we have that for $T = \exp(\mathcal{O}(d))$,

$$\|\hat{\mathbf{n}}_T - \mathbf{n}_T\| < \left(\sqrt{\frac{5(\nu_0^2 + \sigma_T^2)}{8} + \frac{3(\nu_{\max}^2 + \sigma_T^2)}{8(1-c_\nu)}} - \sqrt{\frac{\nu_0^2 + \sigma_T^2}{2} + \frac{\nu_{\max}^2 + \sigma_T^2}{2(1-c_\nu)}}\right) \sqrt{d}.$$

By Lemma 8 we have that with probability at least $1 - \exp(-\Omega(d))$,

$$\|\hat{\mathbf{n}}_T\|^2 \geq \left(\frac{5(\nu_0^2 + \sigma_T^2)}{8} + \frac{3(\nu_{\max}^2 + \sigma_T^2)}{8(1-c_\nu)}\right) d.$$

Combining the two inequalities implies the desired bound

$$\|\mathbf{n}_T\| \geq \|\hat{\mathbf{n}}_T\| - \|\hat{\mathbf{n}}_T - \mathbf{n}_T\| > \sqrt{\left(\frac{\nu_0^2 + \sigma_T^2}{2} + \frac{\nu_{\max}^2 + \sigma_T^2}{2(1-c_\nu)}\right) d}.$$

Hence by induction we obtain $\|\mathbf{n}_t\|^2 > \left(\frac{\nu_0^2 + \sigma_T^2}{2} + \frac{\nu_{\max}^2 + \sigma_T^2}{2(1-c_\nu)}\right) d$ for all $t \in [T]$ with probability at least

$$(1 - (T-1)\exp(-\Omega(d))) \cdot (1 - \exp(-\Omega(d))) \geq 1 - T\exp(-\Omega(d)).$$

Therefore we complete the proof of Proposition 6. $\qquad\square$

Finally, combining equation 20 and Proposition 6 finishes the proof of Theorem 4.

 **A.5   Proof of Theorem 5: Convergence Analysis of Chained Langevin Dynamics**

738  For simplicity, denote $\mathbf{x}^{[q]} = \left\{ \mathbf{x}^{(1)}, \cdots, \mathbf{x}^{(q)} \right\}$. By the definition of total variation distance, for all
739  $q \in [d/Q]$ we have

$$
\begin{aligned}
&\mathrm{TV}\left( \hat{P}\left( \mathbf{x}^{[q]} \right), P\left( \mathbf{x}^{[q]} \right) \right) \\
&= \frac{1}{2} \int \left| \hat{P}\left( \mathbf{x}^{[q]} \right) - P\left( \mathbf{x}^{[q]} \right) \right| \mathrm{d}\mathbf{x}^{[q]} \\
&= \frac{1}{2} \int \left| \hat{P}\left( \mathbf{x}^{(q)} \mid \mathbf{x}^{[q-1]} \right) \hat{P}\left( \mathbf{x}^{[q-1]} \right) - P\left( \mathbf{x}^{(q)} \mid \mathbf{x}^{[q-1]} \right) P\left( \mathbf{x}^{[q-1]} \right) \right| \mathrm{d}\mathbf{x}^{[q]} \\
&\leq \frac{1}{2} \int \left| \hat{P}\left( \mathbf{x}^{(q)} \mid \mathbf{x}^{[q-1]} \right) \hat{P}\left( \mathbf{x}^{[q-1]} \right) - \hat{P}\left( \mathbf{x}^{(q)} \mid \mathbf{x}^{[q-1]} \right) P\left( \mathbf{x}^{[q-1]} \right) \right| \mathrm{d}\mathbf{x}^{[q]} \\
&\quad + \frac{1}{2} \int \left| \hat{P}\left( \mathbf{x}^{(q)} \mid \mathbf{x}^{[q-1]} \right) P\left( \mathbf{x}^{[q-1]} \right) - P\left( \mathbf{x}^{(q)} \mid \mathbf{x}^{[q-1]} \right) P\left( \mathbf{x}^{[q-1]} \right) \right| \mathrm{d}\mathbf{x}^{[q]} \\
&= \frac{1}{2} \int \hat{P}\left( \mathbf{x}^{(q)} \mid \mathbf{x}^{[q-1]} \right) \mathrm{d}\mathbf{x}^{(q)} \int \left| \hat{P}\left( \mathbf{x}^{[q-1]} \right) - P\left( \mathbf{x}^{[q-1]} \right) \right| \mathrm{d}\mathbf{x}^{[q-1]} \\
&\quad + \frac{1}{2} \int \left| \hat{P}\left( \mathbf{x}^{(q)} \mid \mathbf{x}^{[q-1]} \right) - P\left( \mathbf{x}^{(q)} \mid \mathbf{x}^{[q-1]} \right) \right| \mathrm{d}\mathbf{x}^{(q)} \int P\left( \mathbf{x}^{[q-1]} \right) \mathrm{d}\mathbf{x}^{[q-1]} \\
&= \mathrm{TV}\left( \hat{P}\left( \mathbf{x}^{[q-1]} \right), P\left( \mathbf{x}^{[q-1]} \right) \right) + \mathrm{TV}\left( \hat{P}\left( \mathbf{x}^{(q)} \mid \mathbf{x}^{[q-1]} \right), P\left( \mathbf{x}^{(q)} \mid \mathbf{x}^{[q-1]} \right) \right) \\
&\leq \mathrm{TV}\left( \hat{P}\left( \mathbf{x}^{[q-1]} \right), P\left( \mathbf{x}^{[q-1]} \right) \right) + \varepsilon \cdot \frac{Q}{d}.
\end{aligned}
$$

740  Upon summing up the above inequality for all $q \in [d/Q]$, we obtain

$$
\begin{aligned}
\mathrm{TV}\left( \hat{P}(\mathbf{x}), P(\mathbf{x}) \right) &= \sum_{q=1}^{d/Q} \left( \mathrm{TV}\left( \hat{P}\left( \mathbf{x}^{[q]} \right), P\left( \mathbf{x}^{[q]} \right) \right) - \mathrm{TV}\left( \hat{P}\left( \mathbf{x}^{[q-1]} \right), P\left( \mathbf{x}^{[q-1]} \right) \right) \right) \\
&\leq \sum_{q=1}^{d/Q} \varepsilon \cdot \frac{Q}{d} = \varepsilon
\end{aligned}
$$

741  Thus we finish the proof of Theorem 5.

# B   Additional Experiments

743  **Algorithm Setup:** Our choices of algorithm hyperparameters are based on Song and Ermon (2019).
744  We consider $L = 10$ different standard deviations such that $\{\lambda_i\}_{i \in [L]}$ is a geometric sequence with
745  $\lambda_1 = 1$ and $\lambda_{10} = 0.01$. For annealed Langevin dynamics with $T$ iterations, we choose the noise
746  levels $\{\sigma_t\}_{t \in [T]}$ by repeating every element of $\{\lambda_i\}_{i \in [L]}$ for $T/L$ times and we set the step size as
747  $\delta_t = 2 \times 10^{-5} \cdot \sigma_t^2 / \sigma_T^2$ for every $t \in [T]$. For vanilla Langevin dynamics with $T$ iterations, we use the
748  same step size as annealed Langevin dynamics. For chained Langevin dynamics with $T$ iterations, the
749  patch size $Q$ is chosen depending on different tasks. For every patch of chained Langevin dynamics,
750  we choose the noise levels $\{\sigma_t\}_{t \in [TQ/d]}$ by repeating every element of $\{\lambda_i\}_{i \in [L]}$ for $TQ/dL$ times
751  and we set the step size as $\delta_t = 2 \times 10^{-5} \cdot \sigma_t^2 / \sigma_{TQ/d}^2$ for every $t \in [TQ/d]$.

## B.1   Synthetic Gaussian Mixture Model

753  We choose the data distribution $P$ as a mixture of three Gaussian components in dimension $d = 100$:

$$
P = 0.2 P^{(0)} + 0.4 P^{(1)} + 0.4 P^{(2)} = 0.2 \mathcal{N}(\mathbf{0}_d, 3\boldsymbol{I}_d) + 0.4 \mathcal{N}(\mathbf{1}_d, \boldsymbol{I}_d) + 0.4 \mathcal{N}(-\mathbf{1}_d, \boldsymbol{I}_d).
$$

754  Since the distribution is given, we assume that the sampling algorithms have access to the ground-truth
755  score function. We set the batch size as 1000 and patch size $Q = 10$ for chained Langevin dynamics.
756  We use $T \in \left\{ 10^3, 10^4, 10^5, 10^6 \right\}$ iterations for vanilla, annealed, and chained Langevin dynamics.
757  The initial samples are i.i.d. chosen from $P^{(0)}$, $P^{(1)}$, or $P^{(2)}$, and the results are presented in Figures

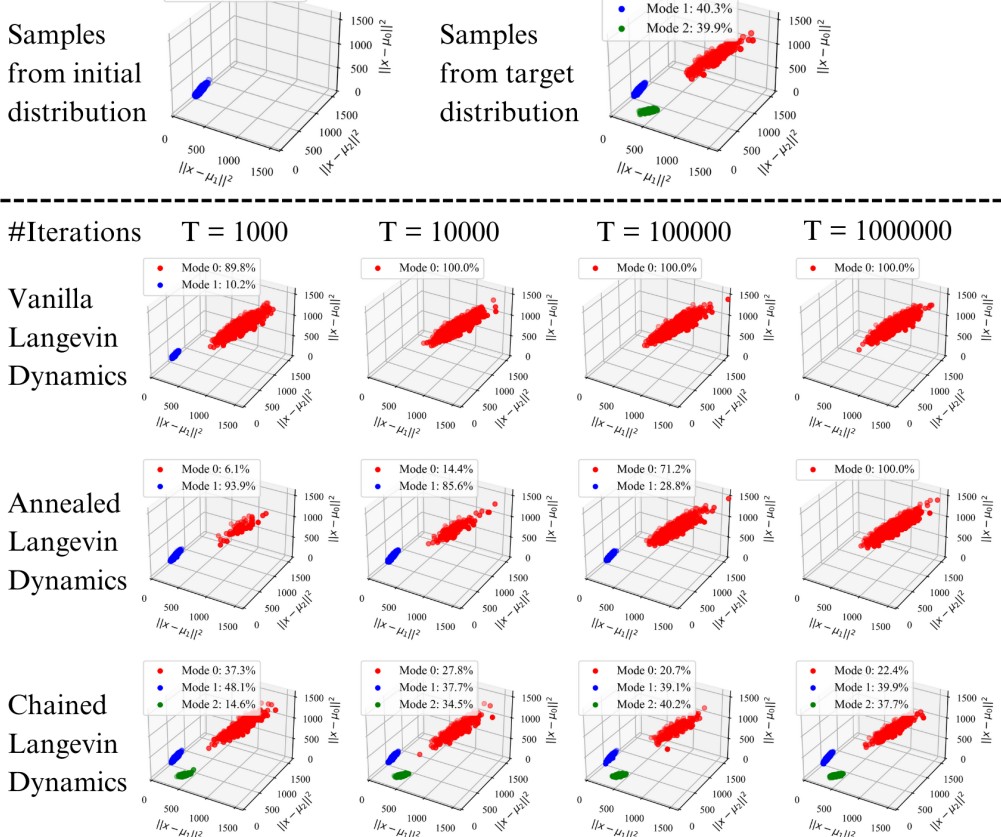

Figure 4: Samples from a mixture of three Gaussian modes generated by vanilla, annealed, and chained Langevin dynamics. Three axes are $\ell_2$ distance from samples to the mean of the three modes. The samples are initialized in mode 1.

1, 4, and 5 respectively. The two subfigures above the dashed line illustrate the samples from the initial distribution and target distribution, and the subfigures below the dashed line are the samples generated by different algorithms. A sample $\mathbf{x}$ is clustered in mode 1 if it satisfies $\|\mathbf{x} - \boldsymbol{\mu}_1\|^2 \leq 5d$ and $\|\mathbf{x} - \boldsymbol{\mu}_1\|^2 \leq \|\mathbf{x} - \boldsymbol{\mu}_2\|^2$; in mode 2 if $\|\mathbf{x} - \boldsymbol{\mu}_2\|^2 \leq 5d$ and $\|\mathbf{x} - \boldsymbol{\mu}_1\|^2 > \|\mathbf{x} - \boldsymbol{\mu}_2\|^2$; and in mode 0 otherwise. The experiments were run on an Intel Xeon CPU with 2.90GHz.

## B.2 Image Datasets

Our implementation and hyperparameter selection are based on Song and Ermon (2019). During training, we i.i.d. randomly flip an image with probability 0.5 to construct the two modes (i.e., original and flipped images). All models are optimized by Adam with learning rate 0.001 and batch size 128 for a total of 200000 training steps, and we use the model at the last iteration to generate the samples. We perform experiments on MNIST (LeCun, 1998) (CC BY-SA 3.0 License) and Fashion-MNIST (Xiao et al., 2017) (MIT License) datasets and we set the patch size as $Q = 14$.

For the score networks of vanilla and annealed Langevin dynamics, following from Song and Ermon (2019), we use the 4-cascaded RefineNet (Lin et al., 2017), a modern variant of U-Net (Ronneberger et al., 2015) with residual design. For the score networks of chained Langevin dynamics, we use the official PyTorch implementation of an LSTM network (Sak et al., 2014) followed by a linear layer. For MNIST and Fashion-MNIST datasets, we set the input size of the LSTM as $Q = 14$, the number of features in the hidden state as 1024, and the number of recurrent layers as 2. The inputs of LSTM include inputting tensor, hidden state, and cell state, and the outputs of LSTM include the next hidden state and cell state, which can be fed to the next input. To estimate the noisy score function, we first input the noise level $\sigma$ (repeated for $Q$ times to match the input size of LSTM) and all-0 hidden and

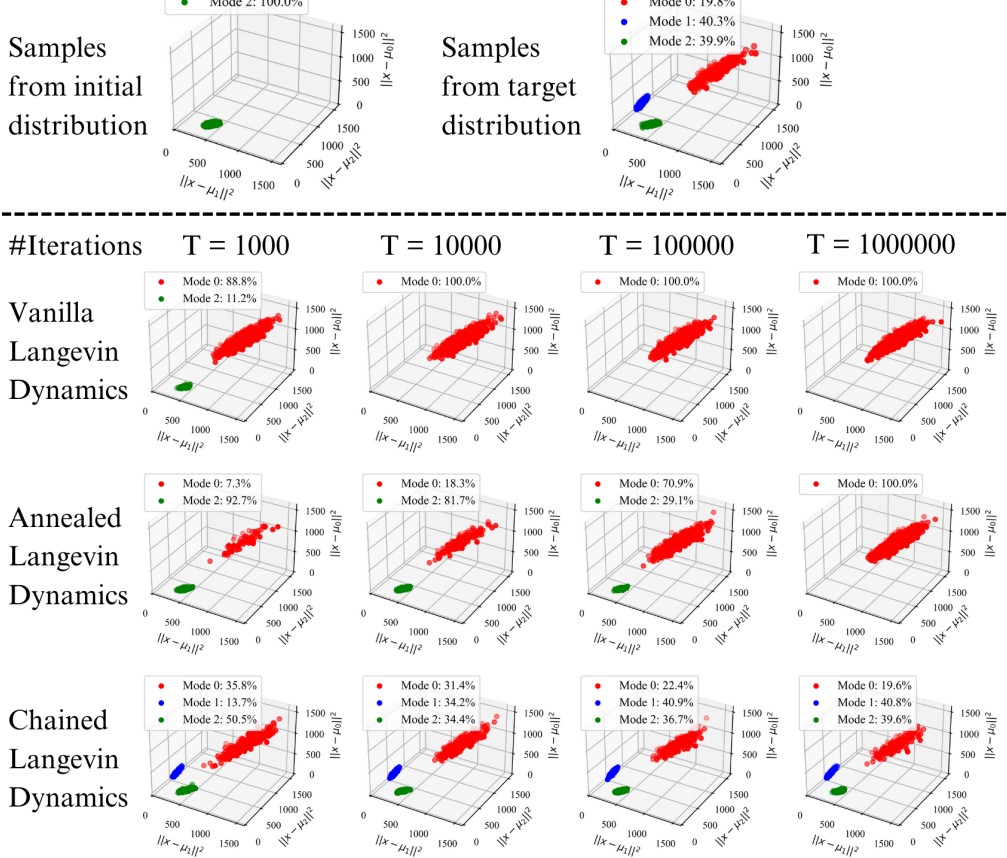

Figure 5: Samples from a mixture of three Gaussian modes generated by vanilla, annealed, and chained Langevin dynamics. Three axes are $\ell_2$ distance from samples to the mean of the three modes. The samples are initialized in mode 2.

cell states to obtain an initialization of the hidden and cell states. Then, we divide a sample into $d/Q$ patches and input the sequence of patches to the LSTM. For every output hidden state corresponding to one patch, we apply a linear layer of size $1024 \times Q$ to estimate the noisy score function of the patch.

To generate samples, we use $T \in \{3000, 10000, 30000, 100000\}$ iterations for vanilla, annealed, and chained Langevin dynamics. The initial samples are chosen as either original or flipped images from the dataset, and the results for MNIST and Fashion-MNIST datasets are presented in Figures 2, 6, 3, and 7 respectively. The two subfigures above the dashed line illustrate the samples from the initial distribution and target distribution, and the subfigures below the dashed line are the samples generated by different algorithms. High-quality figures generated by annealed and chained Langevin dynamics for $T = 100000$ iterations are presented in Figures 8 and 9.

All experiments were run with one RTX3090 GPU. It is worth noting that the training and inference time of chained Langevin dynamics using LSTM is considerably faster than vanilla/annealed Langevin dynamics using RefineNet. For a course of 200000 training steps on MNIST/Fashion-MNIST, due to the different network architectures, LSTM takes around 2.3 hours while RefineNet takes around 9.2 hours. Concerning image generation, chained Langevin dynamics is significantly faster than vanilla/annealed Langevin dynamics since every iteration of chained Langevin dynamics only updates a patch of constant size, while every iteration of vanilla/annealed Langevin dynamics requires computing all coordinates of the sample. One iteration of chained Langevin dynamics using LSTM takes around 1.97 ms, while one iteration of vanilla/annealed Langevin dynamics using RefineNet takes around 43.7 ms.

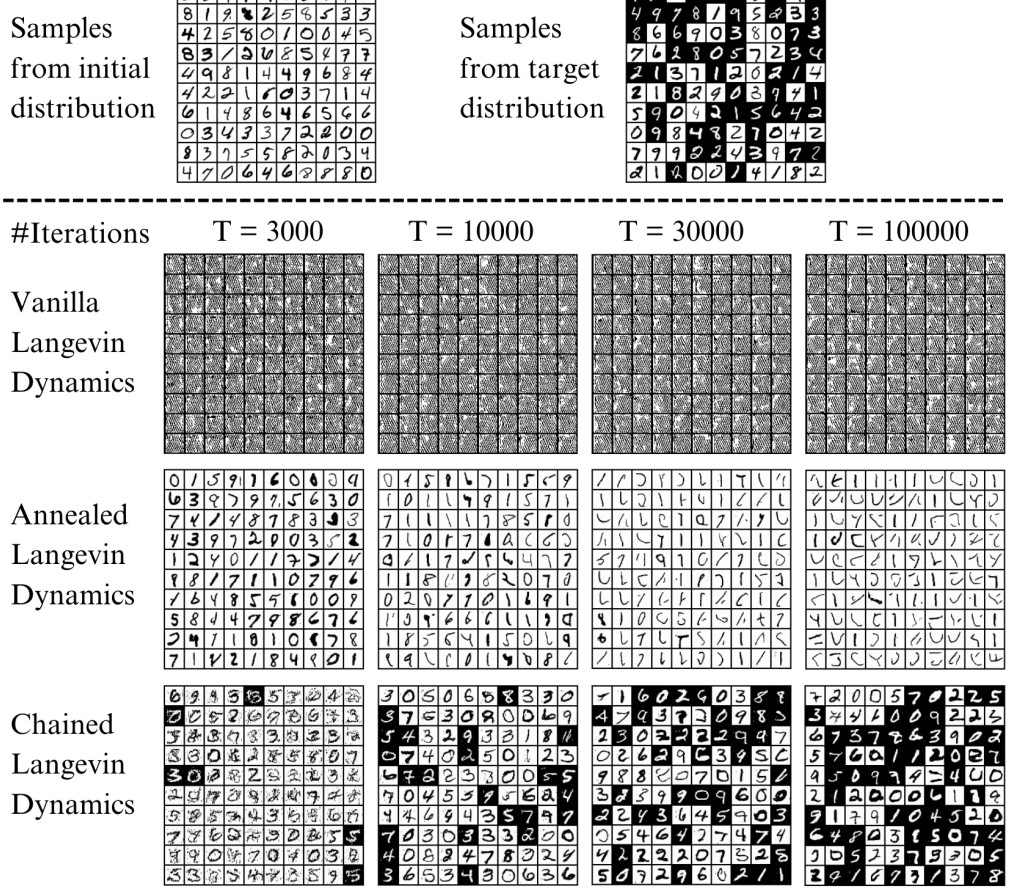

Figure 6: Samples from a mixture distribution of the original and flipped images from the MNIST dataset generated by vanilla, annealed, and chained Langevin dynamics. The samples are initialized as flipped images from MNIST.

## C  Boarder Impacts

This paper presents work whose goal is to advance the field of machine learning. No potential societal consequence of this work needs to be highlighted here.

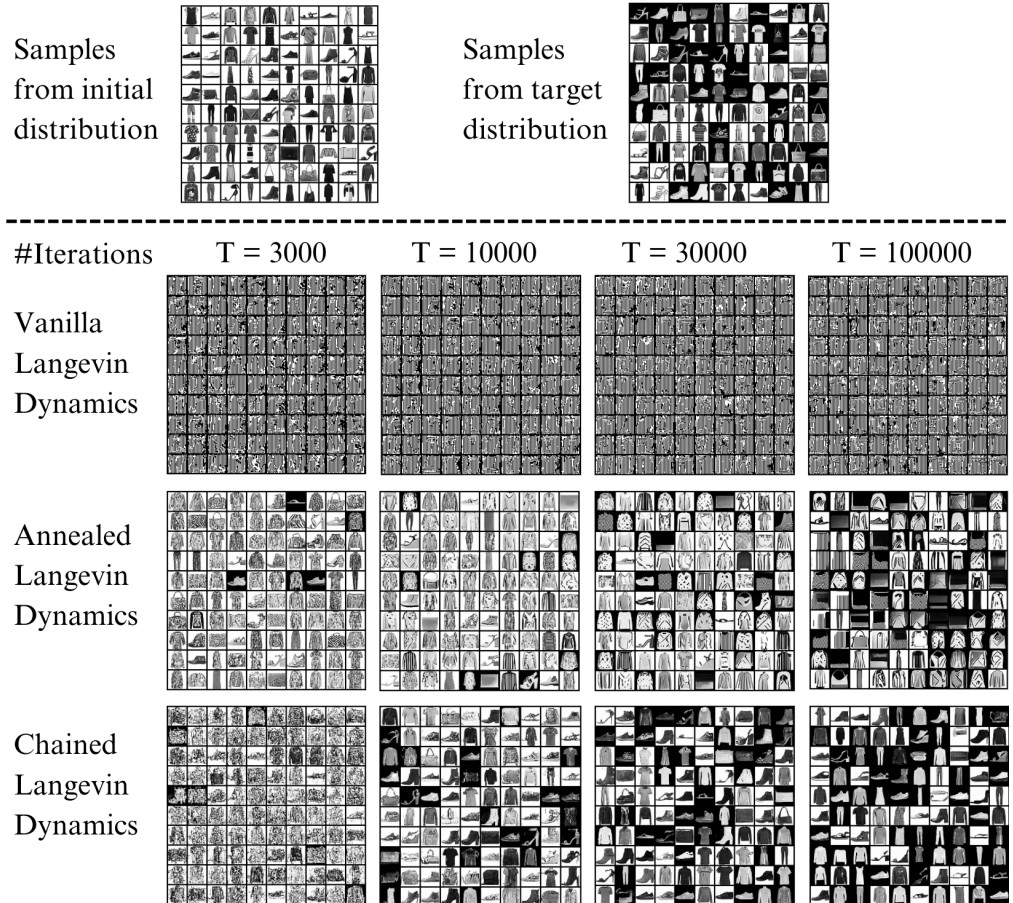

Figure 7: Samples from a mixture distribution of the original and flipped images from the Fashion-MNIST dataset generated by vanilla, annealed, and chained Langevin dynamics. The samples are initialized as flipped images from Fashion-MNIST.

Initialization          Original Images                    Flipped Images

Annealed
Langevin
Dynamics

Chained
Langevin
Dynamics

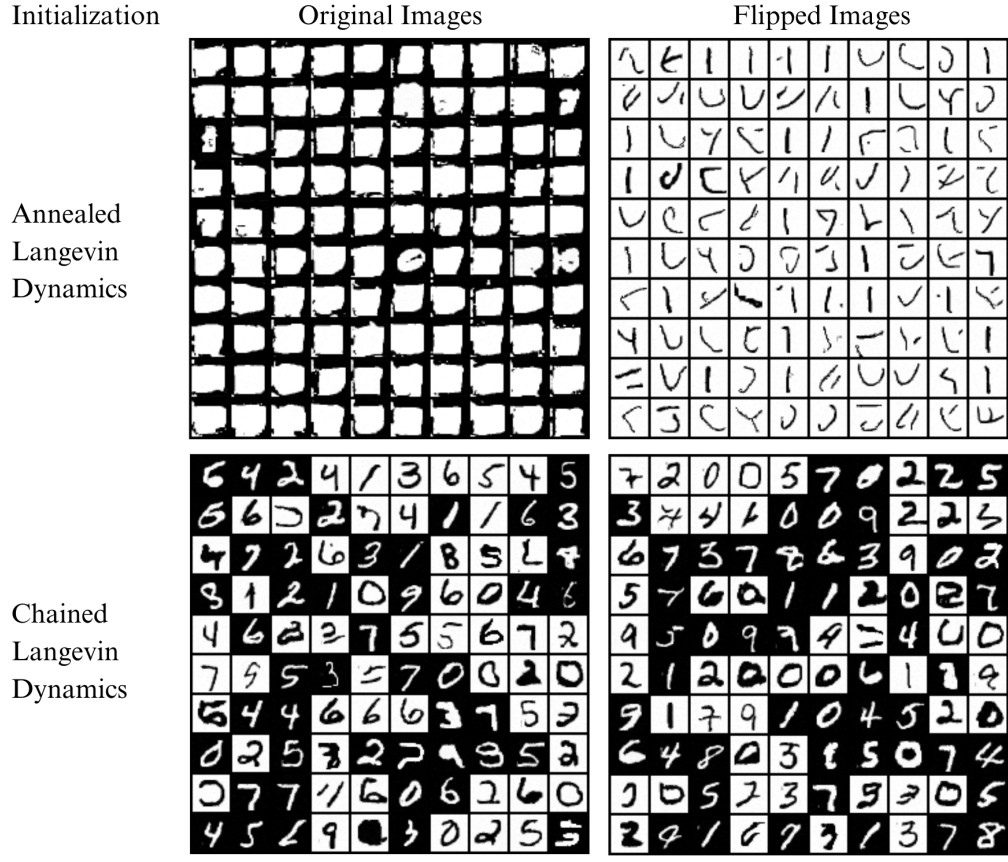

Figure 8: Samples from a mixture distribution of the original and flipped images from the MNIST dataset generated by annealed and chained Langevin dynamics for $T = 100000$ iterations. The samples are initialized as the original or flipped images from MNIST.

Initialization            Original Images                    Flipped Images

Annealed
Langevin
Dynamics

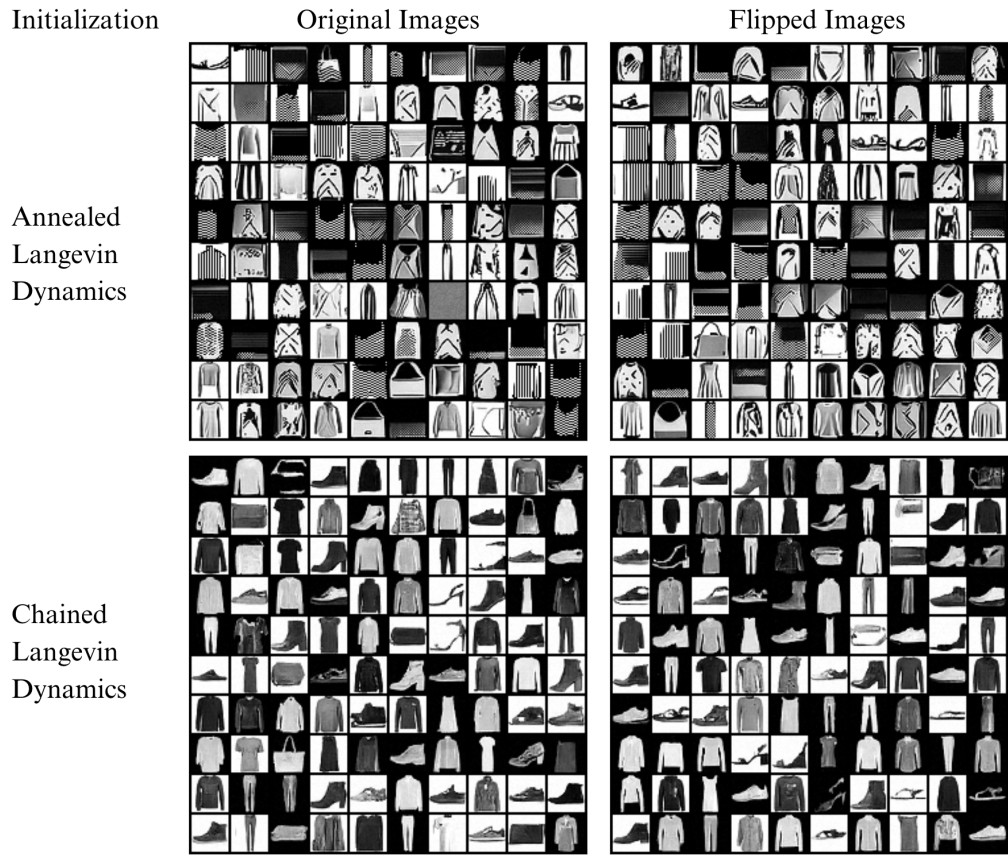

Chained
Langevin
Dynamics

Figure 9: Samples from a mixture distribution of the original and flipped images from the Fashion-MNIST dataset generated by annealed and chained Langevin dynamics for $T = 100000$ iterations. The samples are initialized as the original or flipped images from Fashion-MNIST.

