# OpenReview forum: "On the Mode-Seeking Properties of Langevin Dynamics"
_NeurIPS.cc/2024/Conference — Submitted to NeurIPS 2024_

### Official Review · Reviewer_nEdB · 2024-07-06

**Soundness:** 3
**Presentation:** 3
**Contribution:** 3
**Rating:** 7
**Confidence:** 3

**Summary:**

The authors consider the Langevin process for sampling from a target distribution $\pi$. This process is known to be slow-converging for multimodal targets: in practice, it has been observed that the process gets "stuck" in some modes of the target, and do not "reach" other modes of the target. The authors provide theoretical results for this behavior. In Theorem 1, they prove that by evolving a particle with the Langevin process during exponential time (in the dimension), the particle will still be far away (in probability) from some modes. They also prove, in Theorem 2, that this negative result holds even when using the popular heuristic of "annealing" the Langevin process using intermediate distributions, obtained by adding different levels of Gaussian noise to the target samples.

Instead, the authors propose running an alternative sampling process which they call "Chained Langevin dynamics". This consists in running "annealed" Langevin processes for each component of the target distribution, that is, for each $\pi(x_i | x_{-i})$. The authors estimate the score of each of these conditional targets using a score-matching loss, and empirically demonstrate the ability of their process to reach the different target modes in a limited time. Theoretically, they prove their process approximates the target (in TV divergence) in linear time (in the dimension).

**Strengths:**

The authors provide an interesting perspective on popular sampling processes, Langevin and its annealed counterpart. The paper is clearly written and the results are an interesting contribution to the sampling community.

**Weaknesses:**

See questions.

**Questions:**

**Q1**: Theorem 5 is stated at the distribution-level (the result is for $p_t$), while Theorems 1 and 2 are stated at the particle-level (the result is for $x_t \sim p_t$ ). The authors hint that Theorems 1 and 2 could also be formulated at the distribution-level: "this notion can also easily be translated into a lower bound in terms of other distance measures such as total variation distance and Wasserstein 2-distance". If this is an easy result to add, could the authors add it to their paper? Lower-bounds are notoriously hard(er than upper bounds) on the convergence. Namely, stating Theorem 2 as a lower bound on Annealed Langevin Dynamics would be an important contribution.

**Q2**: Theorem 2 is surprising to me. It seems to convey that Langevin Dynamics are *worse* when annealing, in the sense that the particle $x_t$ is *further* away by the additive constant $2 \sigma_t^2$ from the non-dominant modes $\mu_i$. This seems to be in contradiction with theoretical and empirical results that show that annealing helps reach all the modes of the target distribution, as conveyed in Figure 1 of [1] or Figure 3 of [2] or even Figure [3] of the authors' submission where the Annealed Langevin process produces more accurate samples than the Vanilla Langevin process. Can the authors explain why that is?

**Q3**: When introducing the Chained Langevin Dynamics, the authors seem to motivate this process by saying that by sampling one component at a time from $\pi(x_i | x_{-i})$, this reduces the complexity in the dimension. Could the authors elaborate on this? My understand is that if each of the one-dimensional, conditional targets $\pi(x_i | x_{-i})$ were heavily multimodal, then Annealed Langevin dynamics would still struggle to cover all the modes, although it would struggle less than in higher dimensions in light of Theorem 2 taking $d >> 1$.

[1] Dynamical Regimes of Diffusion Models. Biroli et al. Arxiv, 2024.

[2] Generative Modeling by Estimating Gradients of the Data Distribution. Song et al. NeurIPS 2019.

**Limitations:**

Yes.

---

> ### Author Rebuttal · Authors · 2024-08-07
>
> We would like to thank Reviewer nEdB for his/her time and constructive feedback on our work. Below is our response to the questions and comments in the review.
>
>
> **1- Guarantees in terms of standard distance measures between probability models**
>
> **Re:** Please refer to global rebuttal #1.
>
>
>
> **2- Comparison between vanilla and annealed Langevin dynamics**
>
> **Re:** We would like to clarify that our theoretical results do not imply that annealed Langevin dynamics is worse than the vanilla Langevin dynamics, since the hidden constant in the notation $\Omega(d)$ in Theorem 2 is smaller than Theorem 1. Note that the hidden constant $c_1$ in $\Omega(d)$ for Theorem 1 is
>
> $$
> c_1 = \min \left\\{ \frac{1}{2} \left(\frac{\nu_0^2-\nu_{\max}^2}{8\nu_0^2}\right)^2 , \frac{1}{8}  \left( \log \left( \frac{\nu_{\max}^2}{\nu_0^2} \right) - \frac{\nu_{\max}^2}{2\nu_0^2} + \frac{\nu_0^2}{2\nu_{\max}^2} \right), \frac{1}{32},    \frac{(\nu_0^2-\nu_{\max}^2)^2}{32\nu_0^2(\nu_0^2+\nu_{\max}^2)}  \right\\},
> $$
>
> while the constant $c_2$ for Theorem 2 is
>
> $$
> c_2 = \min \left\\{ \frac{1}{2} \left(\frac{\nu_0^2-\nu_{\max}^2}{8(\nu_0^2+c_{\sigma}^2)}\right)^2 , \frac{1}{8}  \left( \log \left( \frac{\nu_{\max}^2+c_{\sigma}^2}{\nu_0^2+c_{\sigma}^2} \right) - \frac{\nu_{\max}^2+c_{\sigma}^2}{2(\nu_0^2+c_{\sigma}^2)} + \frac{\nu_0^2+c_{\sigma}^2}{2(\nu_{\max}^2+c_{\sigma}^2)} \right), \frac{1}{32},    \frac{(\nu_0^2-\nu_{\max}^2)^2}{32(\nu_0^2+c_{\sigma}^2)(\nu_0^2+\nu_{\max}^2+2c_{\sigma}^2)}  \right\\}.
> $$
>
>
>
>
> **3- Effect of dimension reduction on mode covering hardness**
>
> **Re:** As mentioned in Theorem 5, we prove a linear reduction from learning the high-dimensional variable to a constant-dimensional variable. We define $\tau(\varepsilon/d)$ as the iteration complexity for Langevin dynamics to learn a $Q$-dimensional distribution (for constant $Q$) within $Q \cdot \varepsilon/d$ total variation distance. We note that Theorem 1 of [1] implies that for a $Q$-dimensional distribution $P(\mathbf{x}^{(q)} \mid \mathbf{x}^{(1)}, \cdots, \mathbf{x}^{(q-1)})$ with smoothness and local nonconvexity assumptions on the log-pdf (specified in Appendix A of [1]), we have
>
> $$
> \tau(\varepsilon/d) \le c \cdot \frac{d^2}{\varepsilon^2} \log \left( \frac{d^2}{\varepsilon^2} \right)
> $$
>
> for some constant $c > 0$. Therefore, Theorem 5 shows that Chained Langevin dynamics can achieve $TV(\hat P(\mathbf{x}), P(\mathbf{x})) \le \varepsilon$ in $\frac{c}{Q} \cdot \frac{d^3}{\varepsilon^2} \log \left( \frac{d^2}{\varepsilon^2} \right)$ iterations.
>
>
>
>
> [1] Ma, Y. A., Chen, Y., Jin, C., Flammarion, N., & Jordan, M. I. (2019). Sampling can be faster than optimization. Proceedings of the National Academy of Sciences, 116(42), 20881-20885.

---

> ### Comment · Reviewer_nEdB · 2024-08-11
> **Answer to the authors**
>
> I thank the authors for their response. However, two points are still quite unclear to me.
>
> **Explaining the limitation of Annealed Langevin dynamics**. I thank the authors for their clarification on the bounds for Vanilla and Annealed Langevin Dynamics. I understand these bounds have hidden constants and therefore one cannot easily claim that Annealed Langevin Dynamics performs worse. That being said, the authors seem to claim a serious limitation of Annealed Langevin Dynamics in Theorem 2, which I interpret to be that particles following this process will always be "far enough" from some mode and this hinders convergence. As I stated above: this is surprising, given the widespread success of Annealed Langevin Dynamics for efficiently sampling from multimodal target distributions. Reviewer Ave7 also picked up on this, saying "unlike what was believed, annealed Langevin also fails." Because this is surprising, Theorem 2 deserves some commentary, some high-level argument explaining why this popular method may fail. Pointing to the proof is not enough. As it stands, Theorem 2 is formally stated at the end of section 4.1 with no explanation.
>
> **Explaining the benefits of Chained Langevin dynamics**. While I understand the authors' result, I still find it unclear why using the conditionals of the target $p(x_i | x_{-i})$ makes the sampling problem easier than sampling from the joint target $p(x)$. The authors mention Theorem 1 of [1] which they apply to each conditional $p(x_i | x_{-i})$, but couldn't this Theorem be applied directly to the joint target directly $p(x)$? The authors also mention that the joint target has a multi-dimensional input, while the conditionals have a one-dimensional input: still, that is not a satisfying explanation. Sampling from a one-dimensional distribution is not necessarily easier. Again, could the authors provide some high-level arguments on why sampling from the conditional distributions helps, versus sampling from the joint distribution?
>
> [1] Ma, Y. A., Chen, Y., Jin, C., Flammarion, N., & Jordan, M. I. (2019). Sampling can be faster than optimization. Proceedings of the National Academy of Sciences, 116(42), 20881-20885.

---

> > ### Author Response · Authors · 2024-08-12
> > **Thanks for your feedback**
> >
> > We thank Reviewer nEdB for his/her time and feedback on our response. Regarding the raised points:
> >
> > **1- Theoretical results on annealed Langevin dynamics**
> >
> > **Re:** Thank you for letting us know about the unclarity in interpreting Theorem 2 on annealed Langevin dynamics (ALD). We would like to clarify that Theorem 2 does not aim to highlight a serious limitation of the ALD approach. Please note that our analysis focuses on the diversity of generated samples under a multi-modal distribution. Especially, Theorem 2 in our work suggests that if we consider an upper-bound $c_\sigma$ on the noise levels $\sigma = O(1)$ in ALD that remains a constant in dimension $d$, then the generated samples over a sub-exponential iterations (in dimension $d$) could miss low-variance modes separated from the initialization mode $P^{(0)}$. Of course, we assume a constant bound (dimension-independent) on the noise level $c_\sigma=O(1)$ in ALD, which does not imply the ALD method would suffer from mode dropping in the general case.
> >
> > We note that the proper selection of noise level in ALD has been acknowledged in the literature. For example, Song and Ermon (2020) [2] explain in their paper: "it is necessary for $\sigma_0$ to be numerically comparable to the maximum pairwise distances of data to facilitate transitioning of Langevin dynamics and hence improving sample diversity" (on page 4 of [2]).
> >
> > In the revision, we will be clear about the constant noise level assumption in our writing, and use the term “annealed Langevin dynamics with *bounded noise level*” to ensure the result will not be misinterpreted as a general limitation of ALD.
> >
> > **2- Iteration complexity of Chained Langevin dynamics**
> >
> > **Re:** On a high level, Langevin dynamics performs a *noisy local search* to generate samples around the peaks of the likelihood function. When sampling from $P(\mathbf{x})$ for a high-dimensional $\mathbf{x}\in\mathbb{R}^d$, the algorithm has to randomly explore a large volume growing exponentially in $d$ to find the high-probability yet low-variance modes $P^{(1)},\ldots , P^{(k)}$. On the other hand, when sampling from the conditional distribution $P(\mathbf{x}^{(q)} \mid \mathbf{x}^{(1)}, \cdots, \mathbf{x}^{(q-1)})$, the algorithms needs to only search over a $Q$-dimensional space ($Q$ being the patch size where $Q\ll d$) to find the peaks of the resulting multi-modal conditional density. Therefore, one can expect a faster convergence to the support set of target modes $P^{(1)},\ldots , P^{(k)}$.
> >
> > Regarding the reviewer’s question on applying Theorem 1 in [1], we want to clarify that this theorem states
> > $$
> > \tau(\varepsilon) \le \mathcal{O} \left( \exp(32LR^2) \kappa^2 \frac{d}{\varepsilon^2} \ln \left( \frac{d}{\varepsilon^2} \right) \right).
> > $$
> >
> > The above bound is exponential in $R^2$ ($R$ is the radius of strong convexity of $\log(P)$ where $P$ is the density function), which in the Gaussian mixture case of our theorems will scale linearly with dimension $d$. For example, if we directly apply the theorem to the joint target of the Gaussian mixture case in our synthetic experiments (Section 6), we have $R^2 \ge d$, which means Langevin dynamics is expected to require $\mathcal{O}(\exp(32Ld))$ iterations (Theorem 1 in [1]). On the other hand, chained Langevin dynamics breaks the sample into patches of $Q$-dimension, for which case $R^2 = Q$ (for constant $Q$) and thus making the term $\mathcal{O}(\exp(32LQ))$ to be independent of $d$.
> >
> >
> >
> > [1] Ma, Y. A., Chen, Y., Jin, C., Flammarion, N., & Jordan, M. I. (2019). Sampling can be faster than optimization. Proceedings of the National Academy of Sciences, 116(42), 20881-20885.
> >
> > [2] Song, Y. and Ermon, S. (2020). Improved techniques for training score-based generative models. Advances in neural information processing systems, 33:12438–12448.

---

> > > ### Comment · Reviewer_nEdB · 2024-08-12
> > > **Answer to authors**
> > >
> > > Thank you authors for these clarifications: they are quite helpful and would be useful to add to the paper.
> > >
> > > As it is currently written, the paper suggests that:
> > > - **Theorem 1: limitation**. Langevin on the full target can fail to find some modes.
> > > - **Theorem 2: limitation**. Annealed Langevin on the full target can fail to find some modes.
> > > - **Theorem 5: solution**. Annealed Langevin on the target conditionals may solve the problem. This is suggested in writing, given that this method is proposed "to reduce the mode-seeking tendencies of vanilla and annealed Langevin dynamics".
> > >
> > > **Comments on Theorems 1 and 2**. As you have said in your comment above, Theorems 1 and 2 are more subtle than this. They point out, on the particle-level, that some target modes may be missed with high probability if the initial distribution is not "noisy enough". This is coherent with the observation of Song and Ermon (2020) who related "noisy enough" to the distance between data points. This result at the particle-level suggests that, at the distribution-level, convergence may be hindered. But that is not automatic. To formally claim that "mode-missing" by particles results in "slower convergence of the probability law", we will need the lower bounds in TV that you stated.
> > >
> > > Do you have these lower bounds for **both** Theorems 1 and 2? So for both Vanilla and Annealed Langevin?
> > >
> > > **Comment on Theorem 5**. As it is stated, it is not at all clear that Theorem 5 "reduce[s] the mode-seeking tendencies of vanilla and annealed Langevin dynamics". Theorem 5 is formally stated as: if you sample the conditionals with a given precision, then you sample the joint with a given precision. But the difficulty of sampling the conditionals is not clear. Again, this theorem is stated bluntly in the text and is not commented. Your argument above is useful: comparing $R^2$ of order $d$ vs. $Q$ and explaining why Langevin on the target conditionals has a nicer upper-bound than Annealed Langevin on the full target. It should be part of the explanation.

---

> > > > ### Author Response · Authors · 2024-08-13
> > > > **Thanks for your feedback**
> > > >
> > > > We sincerely thank Reviewer nEdB for the thorough summarization and constructive feedback on our response. The answer is affirmative to the raised question "Do you have these lower bounds for **both** Theorems 1 and 2?". For both vanilla and annealed Langevin dynamics (with bounded noise levels), Theorems 1 and 2 can be translated into lower bounds in total variation distance, as described in global rebuttal #1. We will clarify this point in the text.

---

### Official Review · Reviewer_Ave7 · 2024-07-11

**Soundness:** 4
**Presentation:** 3
**Contribution:** 3
**Rating:** 6
**Confidence:** 3

**Summary:**

The authors study Langevin dynamics (as well as its annealed counterpart) for gaussian mixtures and sub-gaussian mixtures. In Sec. 4, they prove that Langevin remains stuck in the "dominant mode" for an at least exponential time, a claim that is often made in the ML literature but which is never formally proved. In Sec. 5, they provide a sequential method to get rid of this dependence.

**Strengths:**

It is healthy to finally have a paper that explicitly prove the claims made in the ML literature and that were known in practice for a long time.  Furthermore, it shows that, unlike what was believed, annealed Langevin also fails.

**Weaknesses:**

I do not understand why it is sensible to say that initially, $p_0$ should follow $P_0$, one of the component of the mixture, isn't it a rather strong assumption?

**Questions:**

Do similar results hold when $p_0$ is initialized at $\mathcal{N}(0, Id)$ and this component is not in the target mixture?

**Limitations:**

It seems like the assumption $p_0 \sim P_0$ is not enough justified. Also, I would have like an insight of the proof of the Theorems in Sec. 4.

---

> ### Author Rebuttal · Authors · 2024-08-07
>
> We thank Reviewer Ave7 for his/her time and constructive feedback on our work. Below is our response to the questions and comments in the review.
>
>
> **1- Insights behind Theorems 1,2 and the role of mode $P^{(0)}$**
>
>
> **Re:** Please note that in Theorem 1, the mode $P^{(0)}$ plays the role of a large-variance Gaussian mode surrounding the other modes $P^{(i)}$’s with a significantly lower variance. Our intuition is that if the Langevin Dynamics gets initialized at a sample from $P^{(0)}$, then the score function will be dominated by the mode $P^{(0)}$, where the PDF of the other modes is expected to have a significantly less impact on the PDF of the mixture distribution (assuming a high dimension $d$). Therefore, the Langevin Dynamics is expected to randomly explore a large area in $\mathbb{R}^d$ (due to the high variance of $P^{(0)}$) which makes finding the remaining low-variance mode $P^{(i)}$’s overly expensive, requiring an exponential time (in terms of $d$) to find the missing modes. Theorems 1,2 formalize this intuition and prove the iteration complexity of finding the low-variance modes will indeed exponentially grow with the dimension.
>
> We note that the above result can be further extended to a hardness result under the mean separation.  Here, we again assume a high-variance mode $P^{(0)}$ filling in the space between the support sets of the low-variance mode $P^{(1)},\ldots , P^{(k)}$ (with bounded support sets with Euclidean radius $r$). This time, we suppose the Langevin Dynamics is initialized at a sample drawn from $P^{(1)}$ whose mean vector is sufficiently separated from the mean of the other modes. Then, there would be two possibilities for the Langevin Dynamics. Either the dynamics will remain in the support set of $P^{(1)}$ which cannot capture the other modes, or the dynamics would exit $P^{(1)}$‘s support set which will reduce to the case in Theorem 1, requiring the exploration of a large subset of $\mathbb{R}^d$ due to the high variance of the surrounding mode $P^{(0)}$. We will add the remark explaining the implications of Theorem 1 in such a setting where the support sets of $P^{(1)},\ldots , P^{(k)}$ are bounded with a small radius and have sufficiently distant means.
>
>
>
> **2- "Do similar results hold when $p_0$ is initialized at $\mathcal{N}(\mathbf{0}_d, \mathbf{1}_d)$?"**
>
>
> **Re:**  We note that our theoretical result holds as long as, with high probability, the initial sample $\mathbf{x}_0$ will be far from the vector space of $\left\\{ \mathbf{\mu}\_i \right\\}\_{i \in [k]}$. If we assume the sample is initialized as $\mathbf{x}_0 \sim \mathcal{N}(\mathbf{0}_d, \mathbf{1}_d)$ and $P^{(0)}$ satisfies $\nu_0 < 1$, similar to Proposition 2 we know $\left\\| \mathbf{n}\_0 \right\\|^2 \ge \frac{3\nu_0^2+\nu\_{\max}^2}{4} d$ with high probability. Therefore by Proposition 3, we obtain the same result as Theorem 1.

---

> > ### Comment · Reviewer_Ave7 · 2024-08-12
> > **Response to rebuttal**
> >
> > I thank the authors for their feedback.
> > Actually, the fact that similar results would hold if $P_0 \sim \mathcal{N}(0, 1)$ under the additional assumption that the initial sample will be far from the vector space of $\left\{ \mathbf{\mu}_i \right\}_{i \in [k]}$ worries me a bit about the significance of the work as such assumption cannot be expected to hold in practice. After having a closer look to Assumption 1, the lower bound feels somewhat odd as we could expect it to depend on the distance between the modes and not assume a priori that they are close.

---

> > > ### Author Response · Authors · 2024-08-13
> > > **Thanks for your feedback**
> > >
> > > We thank Reviewer Ave7 for his/her feedback on our response. First, we would like to clarify that our response was based on a multivariate $d$-dimensional $\mathbf{x}_0 \sim \mathcal{N}(\mathbf{0}_d,\mathbf{I}_d)$, where the complexity analysis is in terms of dimension $d$. Therefore, our analysis does not focus on the univariate case $\mathcal{N}(0,1)$ mentioned in the reviewer’s question.
> > >
> > > Next, we would like to clarify the assumption of the initial sample being far from the vector space of $\\left\\{ \mathbf{\mu}\_i \\right\\}_{i \in [k]}$. This assumption only requires that the initialized sample $\mathbf{x}_0\in\mathbb{R}^d$ will have a reasonably-large projection onto the $(d-k)$-dimensional subspace orthogonal to vectors $\boldsymbol{\mu}_1,\\ldots , \boldsymbol{\mu}_k$. By only assuming that $d-k$ is moderately large, this assumption is guaranteed to hold according to Proposition 2. This is following the intuition that  a $d$-dimensional Gaussian vector $\mathbf{x}_0 \sim \mathcal{N}(\mathbf{0}_d,\mathbf{I}_d)$ will (with high probability) possess a $\mathcal{O}(\sqrt{d-k})$-magnitude projection onto a fixed $(d-k)$-dimensional space.
> > >
> > >
> > > Regarding Assumption 1, we note that as discussed in rebuttal #1, $P^{(0)}$ is supposed to be a large-variance (yet low-probability) mode *surrounding the small-variance and high-probability modes $P^{(1)}, \cdots, P^{(k)}$*. Assumption 1 formalizes this intuition as the center of the high-variance mode $P^{(0)}$ should not be exceedingly far from the other low-variance modes, otherwise $P^{(0)}$ will not be capable of surrounding the extremely far modes and dominating the Langevin dynamics. Please note that as described in the second paragraph of rebuttal #1, the theoretical result can be further extended to a hardness result assuming the large-enough distance between the low-variance modes $P^{(1)}, \cdots, P^{(k)}$, which requires sufficiently large distance between the means of $P^{(1)}, \cdots, P^{(k)}$ (consistent with the reviewer’s intuition).

---

### Official Review · Reviewer_7G8Z · 2024-07-12

**Soundness:** 4
**Presentation:** 3
**Contribution:** 4
**Rating:** 8
**Confidence:** 4

**Summary:**

A new algorithm is proposed, called Chained Langevin Dynamics, to improve on the mode-seeking properties of Langevin Dynamics, after annleade Langevin Dynamics had been proposed but did not give significant improvements.
Results about the mode-seeking properties of the three algorithms are obtained.
The results of numerical experiments on synthetic and real image datasets are also shown.

**Strengths:**

Very inspiring idea on how to improve mode-search for multimodal distributions.
Very clear presentation of premises and of the old and new algorithms.
The new algorithm looks very powerful.

**Weaknesses:**

No evident connection has been established between experiments and mathematical results.
The description/comment of experiments could have been more accurate (see Questions).

**Questions:**

In structured data, does the order of patches matter?

How large is the selected size Q of the patches in the examples? Why was it selected that way? Smaller patches help convergence due to the reduced dimension. What happens for too small patches? Does the algorithm still work?

"Regarding the neural network architecture of the score function estimator, for vanilla and annealed Langevin dynamics we use U-Net (Ronneberger et al., 2015) following from Song and Ermon (2019). For chained Langevin dynamics, we proposed to use Recurrent Neural Network (RNN) architectures." The change in the set up seems a little arbitrary. What did it happen by using U-Net for chained Langevin dynamics or, viceversa, using the RNN for vanilla and annealed Langevin dynamics?

More on the general concepts of the paper, the authors might find interesting that the principle of chained Langevin dynamics seems based on "nucleation" of different modes and spreading of the information of the randomly selected mode through the conditional probability to the entire image, much like the mechanism described in this old paper that allowed reconstruction of images from very little sampling
Statistical-Physics-Based Reconstruction in Compressed Sensing
F. Krzakala, M. Mézard, F. Sausset, Y. F. Sun, and L. Zdeborová
Phys. Rev. X 2, 021005 – Published 11 May 2012

**Limitations:**

There is a section about limitations in the text.

---

> ### Author Rebuttal · Authors · 2024-08-07
>
> We would like to thank Reviewer 7G8Z for his/her time and constructive feedback and suggestions on our work. Below is our response to the questions and comments in the review.
>
>
> **1- The order of patches in Chained Langevin Dynamics**
>
> **Re:** In our analysis, the convergence rate of chained Langevin dynamics does not change with the patch ordering, as long as each patch can be accurately sampled according to the conditional distribution given the previous patches (an assumption that is supposed to hold for running Langevin Dynamics). To test this, we have performed experiments by (uniformly) randomly choosing the order of patches. As suggested in Figure 1 of rebuttal PDF, the numerical results looked similar to the results of our original implementation.
>
>
> **2- Selection of the patch size hyperparameter**
>
> **Re:** In the paper’s experiments on MNIST and Fashion-MNIST datasets, we chose patch size $Q=14$. To address the reviewer’s question, we tested different values of $Q \in \\{1,7,14,28\\}$. As suggested by Figure 2 in rebuttal PDF, the experimental results look insensitive to a moderate (not overly large) choice of patch size.
>
>
> **3- Selection of the neural network architecture**
>
> **Re:** We chose different architectures for vanilla Langevin dynamics and chained Langevin dynamics due to the difference in the learning objectives. In vanilla and annealed Langevin dynamics, we used a U-Net to jointly estimate the score function of every dimension of the sample due to its high capacity. In chained Langevin dynamics, we applied a Recurrent Neural Network (RNN) to memorize information about the previous inputs and estimate the conditional distribution of the next patch.
>
>
> **4- The reference on the reconstruction of images**
>
> **Re:** We thank the reviewer for introducing the related work. Our intuition of sequential sampling is echoed by the idea in the related work about reconstructing the true signal from its compression at a high level. We will discuss the work in the revised text.

---

### Official Review · Reviewer_3VKE · 2024-07-20

**Soundness:** 3
**Presentation:** 3
**Contribution:** 3
**Rating:** 5
**Confidence:** 3

**Summary:**

This paper studies Langevin-based algorithms for sampling from multimodal distributions, motivated by generative modeling. The main content of the paper are lower bounds on the convergence of both Langevin and annealed Langevin for mixtures of Gaussian and sub-Gaussian distributions, as well as a proposed modification of the annealed Langevin dynamics to operate on coordinate patches one-at-a-time.

**Strengths:**

- Sampling from multimodal distributions is an importnat problem both theoretically and practically.

- The Chained Langevin Dynamics algorithm that is proposed appears to be novel.

- The empirical results are promising, albeit in a rather contrived setting.

**Weaknesses:**

- The lower bounds hold only for the distance between the sample and the mean, rather than any standard notion of distance or divergence between probability measures. Moreover, I do not expect that these bounds imply such a quantity is large.

- Related to the above point, it is difficult to appreciate the significance of the lower bound since the lower bound does not depend on the separation between the means. In particular, it seems the lower bounds only show that the iterate remains roughly on the order of the larger variance which is, for example, not surprising in the case where the variances are all of the same order.

- The hidden constants in the $\Omega$ notation are important but difficult to find (as they are suppressed in the main text and some of the appendix). In particular, there should be dependence on the mixture weights but this can't be seen from their result.

- It is unclear if the upper bound in Theorem 5 can be instantiated for their algorithm (see question below).

**Questions:**

- In light of [1], there is arguably "no mystery" in terms of convergence of the reversed Langevin dynamics in diffusion models: as long as the score function can be accurately estimated, the reversed dynamics will converge to the target. At a high level, why do you then study annealed Langevin in the setting where the scores are known exactly?

- I suggest you stop using the $\Omega$ notation and make the hidden constants more clear as they are very important for understanding your result.

- With regard to Theorem 5, a remark that explains, even if conjecturally, how the run-time of the Langevin algorithm might scale in the dimension and how the bound in Theorem 5 does, or does not, imply that your method could be successful would be greatly helpful.

- On page 2 you write "Regarding discrete SGLD, Lee et al. (2018) constructed a probability distribution whose density is close to a mixture of two well-separated isotropic Gaussians, and proved that SGLD could not find one of the two modes within an exponential number of steps." However, I was not able to find this lower bound in Lee et al. (2018). Which result specifically are you referring to?

[1] Chen, S., Chewi, S., Li, J., Li, Y., Salim, A., and Zhang, A. R. (2023). Sampling is as easy as learning the score: theory for diffusion models with minimal data assumptions. In International Conference on Learning Representations

**Limitations:**

The limitations of the work have been adequately addressed.

---

> ### Author Rebuttal · Authors · 2024-08-07
>
> We thank Reviewer 3VKE for his/her time and detailed feedback on our work. Below is our response to the questions and comments in the review.
>
> **1- Guarantees in terms of standard distance measures between probability models**
>
>
> **Re:** Please refer to global rebuttal #1.
>
>
>
>
> **2- Insights of the lower bounds' dependence on the covariance difference**
>
>
> **Re:** We understand the reviewer’s comment on the impact of covariance separation on our results in Theorems 1,2. In the following, we first explain our intuition behind the setting in these theorems, and next we argue how this result can be extended to the case with mean separation.
>
> Please note that in Theorem 1, the mode $P^{(0)}$ plays the role of a large-variance Gaussian mode surrounding the other modes $P^{(i)}$’s with a significantly lower variance. Our intuition is that if the Langevin Dynamics gets initialized at a sample from $P^{(0)}$, then the score function will be dominated by the mode $P^{(0)}$, where the PDF of the other modes is expected to have a significantly less impact on the PDF of the mixture distribution (assuming a high dimension $d$). Therefore, the Langevin Dynamics is expected to randomly explore a large area in $\mathbb{R}^d$ (due to the high variance of $P^{(0)}$) which makes finding the remaining  low-variance mode $P^{(i)}$’s overly expensive, requiring an exponential time (in terms of $d$) to find the missing modes. Theorems 1,2 formalize this intuition and prove the iteration complexity of finding the low-variance modes will indeed exponentially grow with the dimension.
>
> We note that the above result can be further extended to a hardness result under the mean separation.  Here, we again assume a high-variance mode $P^{(0)}$ filling in the space between the support sets of the low-variance mode $P^{(1)},\ldots , P^{(k)}$ (with bounded support sets with Euclidean radius $r$). This time, we suppose the Langevin Dynamics is initialized at a sample drawn from $P^{(1)}$ whose mean vector is sufficiently separated from the mean of the other modes. Then, there would be two possibilities for the Langevin Dynamics. Either the dynamics will remain in the support set of $P^{(1)}$ which cannot capture the other modes, or the dynamics would exit $P^{(1)}$‘s support set which will reduce to the case in Theorem 1, requiring the exploration of a large subset of $\mathbb{R}^d$ due to the high variance of the surrounding mode $P^{(0)}$. We will add the remark explaining the implications of Theorem 1 in such a setting where the support sets of $P^{(1)},\ldots , P^{(k)}$ are bounded with a small radius and have sufficiently distant means.
>
>
>
> **3- Hidden constants in the $\Omega$ notation**
>
> **Re**: In Theorem 1, notation $\Omega(d)$ means $\Omega(d) \ge cd$,  for the following constant $c$
>
> $$
> c = \min \left\\{ \frac{1}{2} \left(\frac{\nu_0^2-\nu_{\max}^2}{8\nu_0^2}\right)^2 , \frac{1}{8}  \left( \log \left( \frac{\nu_{\max}^2}{\nu_0^2} \right) - \frac{\nu_{\max}^2}{2\nu_0^2} + \frac{\nu_0^2}{2\nu_{\max}^2} \right), \frac{1}{32},    \frac{(\nu_0^2-\nu_{\max}^2)^2}{32\nu_0^2(\nu_0^2+\nu_{\max}^2)}  \right\\},
> $$
>
> when $d$ is greater than
>
> $$
> \max \left\\{ 8  \left( \log \left( \frac{\nu_{\max}^2}{\nu_0^2} \right) - \frac{\nu_{\max}^2}{2\nu_0^2} + \frac{\nu_0^2}{2\nu_{\max}^2} \right)^{-1} \log \left(\frac{3\nu_0^3}{w_0 \min_{i\in[k]}\nu_i^2}\right), \frac{8\nu_0^2(3\nu_0^2+\nu_{\max}^2)}{\pi(\nu_0^2-\nu_{\max}^2)^2} \right\\}.
> $$
>
> For example in the Gaussian mixture in our synthetic experiments (Section 6), $\nu_0=\sqrt{3}$, $\nu_1=\nu_2=1$ and $w_0=0.2$, therefore $\Omega(d) \ge \frac{1}{288} d$ for any $d \ge 149$. The constant in Theorem 2 can be obtained by substituting $\nu_i^2$ with $\nu_i^2+c_{\sigma}^2$. We will include these constants in the main text.
>
>
>
> **4- Running time of Chained Langevin Dynamics**
>
> **Re:** In Theorem 5, we define $\tau(\varepsilon/d)$ as the iteration complexity for Langevin dynamics to learn a $Q$-dimensional distribution (for constant $Q$) within $Q \cdot \varepsilon/d$ total variation distance. We note that Theorem 1 of [1] implies that for a $Q$-dimensional distribution $P(\mathbf{x}^{(q)} \mid \mathbf{x}^{(1)}, \cdots, \mathbf{x}^{(q-1)})$ with smoothness and local nonconvexity assumptions on the log-pdf (specified in Appendix A of [1]), we have
>
> $$
> \tau(\varepsilon/d) \le c \cdot \frac{d^2}{\varepsilon^2} \log \left( \frac{d^2}{\varepsilon^2} \right)
> $$
>
> for some constant $c > 0$. Therefore, Theorem 5 shows that Chained Langevin dynamics can achieve $TV(\hat P(\mathbf{x}), P(\mathbf{x})) \le \varepsilon$ in $\frac{c}{Q} \cdot \frac{d^3}{\varepsilon^2} \log \left( \frac{d^2}{\varepsilon^2} \right)$ iterations.
>
>
> **5- Motivation for studying annealed Langevin dynamics with exact score function**
>
> **Re:** We note that the key difference between Langevin dynamics and denoising diffusion models (DDPM) is that the DDPM’s update rule scales the sample $\mathbf{x}_{i-1}$ with a factor of $\frac{1}{\sqrt{1-\beta_i}}$ at every iteration while the Langevin dynamics do not. The difference is referred to as the variance exploding property of Langevin dynamics and the variance preserving property of DDPM [2]. We think the scaling of samples is an important factor in analyzing the mode-seeking properties of Langevin dynamics.
>
>
> **6- Results of Lee et al. (2018) [3] regarding isotropic Gaussian mixtures**
>
> **Re:** Please refer to Theorem K.1 in Appendix K (Lower bound when Gaussians have different variance) of Lee et al. (2018) [3].
>
>
> [1] Ma, Y. A. et al (2019). Sampling can be faster than optimization. PNAS, 116(42), 20881-20885.
>
> [2] Song, Y. et al. (2020c). Score-based generative modeling through stochastic differential equations. In ICLR.
>
> [3] Lee, H. et al. (2018). Beyond log-concavity: Provable guarantees for sampling multi-modal distributions using simulated tempering langevin monte carlo. Advances in neural information processing systems, 31.

---

> > ### Comment · Reviewer_3VKE · 2024-08-12
> > **Thanks for your thorough response**
> >
> > Thanks for your thorough response, especially the clarification for the TV lower bound as well as the intuition for your setting.
> >
> > One final request:
> >
> > - I assume that in Theorem 1, the $T = \exp(O(t))$ condition means the result holds for _any_ $T$ such that $T \leqslant \exp(O(t))$.
> >
> > I have updated my score to take into account the respones.

---

> > > ### Author Response · Authors · 2024-08-13
> > > **Thanks for your feedback**
> > >
> > > We thank Reviewer 3VKE for his/her feedback on our response. We are pleased to hear the reviewer finds the rebuttal satisfactory. Regarding the raised point, we think the reviewer means whether Theorem 1 holds for any $T \le \exp(\mathcal{O}(d))$ ($t$ replaced by $d$). If so, we confirm the reviewer's interpretation of the statement, as the bounds in Theorems 1-4 hold for any $T \le \exp(\mathcal{O}(d))$. We will clarify this point in the revised text.

---

### Author Rebuttal · Authors · 2024-08-07

We would like to thank the reviewers for their constructive feedback. Here we respond to the common question of Reviewers 3VKE and nEdB. We provide our response to the other comments and questions under each review textbox.


**1- Guarantees in terms of standard distance measures between probability models**

**Re:** As pointed out by the reviewers, our current theoretical statement is in terms of the distance between the generated sample $\mathbf{x}_t$ and the missing mean vector $\boldsymbol{\mu}_i$. We note that this statement can translate into a lower bound guarantee in terms of total variation ($TV$) distance. Please note the definition of total variation distance:

$$
d_{\text{TV}} (\hat P_t, P) = \sup_A \left|\hat P_t(A) - P(A) \right|
$$

In the above, we only need to choose event $A$ as $\left\\{ \mathbf{x} : \forall i \in [k], \\, \left\\| \mathbf{x} - \mathbf{\mu}\_i \right\\|^2 \ge \frac{\nu_0^2 + \nu_{\max}^2}{2} d \right\\}$, which we prove in Theorems 1-2 to occur with high probability. Also, using the standard concentration bound of Gaussians, from Assumption 1 we can derive

$$
P(A) \le w_0 + (1-w_0)\exp\left(-\left( \frac{\nu_0^2 - \nu_{\max}^2}{8\nu_0^2}\right)^2 d\right).
$$

Using the above two equations, the following lower-bound on the total variation distance will follow:

$$
d_{\text{TV}} (\hat P_t, P) \ge \hat P_t(A) - P(A) \ge (1 - w_0) (1 - T \cdot \exp(-\Omega(d))).
$$

We will include the above remark in the revised paper.

---

### Decision · Program_Chairs · 2024-09-25

**Decision:**

Reject

**Comment:**

Although the reviewers have identified some positive aspects of this work, I have ultimately decided to recommend rejection. My main concern is that this work is primarily a theory paper, but it does not have sufficient technical novelty to be a good NeurIPS theory paper.

- I think from a generative modeling perspective, the chained Langevin dynamics is not a particularly novel idea; it's just adding an autoregressive layer to the generation process.
- From a theoretical standpoint, the only analysis given of the chained Langevin dynamics (theorem 5) is basically just the triangle inequality.
- In the main theoretical contribution, authors give some examples of multimodal settings where "Langevin Dynamics is unlikely to find all mixture components within a sub-exponential number of steps in the data dimension". At a high level, this is already a very well-known and well understood phenomena and the authors even cite some works on metastability which have clearly already demonstrated this high level point. I do not think the example given by the authors adds something truly new to the literature.
- The failure of simulated annealing in simple multimodal settings would probably be a more interesting phenomena and the authors made some efforts to extend in this direction. But we already know from the theoretical guarantees for diffusion models that if the setup is done properly then a natural annealed sampler does provably work [e.g. from CCLYZ '23]... (The authors should, I think, at least seriously discuss this in the next version of the work.)

I encourage the authors to revise the work and resubmit to a future venue. Perhaps a more serious investigation of the chained sampler could be done.